# mmT5: Modular Multilingual Pre-Training Solves Source Language Hallucinations

**Jonas Pfeiffer**   **Francesco Piccinno**   **Massimo Nicosia**
**Xinyi Wang**   **Machel Reid**   **Sebastian Ruder**
Google DeepMind

## Abstract

Multilingual sequence-to-sequence models perform poorly with increased language coverage and fail to consistently generate text in the correct target language in few-shot settings. To address these challenges, we propose mmT5, a modular multilingual sequence-to-sequence model. mmT5 utilizes language-specific modules during pre-training, which disentangle language-specific information from language-agnostic information. We identify representation drift during fine-tuning as a key limitation of modular generative models and develop strategies that enable effective zero-shot transfer. Our model outperforms mT5 at the same parameter sizes by a large margin on representative natural language understanding and generation tasks in 40+ languages. Compared to mT5, mmT5 raises the rate of generating text in the correct language under zero-shot settings from 7% to 99%, thereby greatly alleviating the source language hallucination problem.

## 1 Introduction

Multilingual pre-trained models (Conneau et al., 2020a; Xue et al., 2021) have demonstrated impressive performance on natural language understanding (NLU) tasks across different languages (Hu et al., 2020; Ruder et al., 2021). These models are typically trained on large amounts of unlabeled data in hundreds of languages. Recent large language models (Brown et al., 2020; Chowdhery et al., 2023) display surprising multilingual capabilities despite being pre-trained predominantly on English data. However, all of these models share a key limitation: representations of all languages compete for the model's limited capacity. As a result, models perform poorly with an increasing number of pre-training languages and on languages with less pre-training data. This is also known as the "**curse of multilinguality**" (Conneau et al., 2020a).

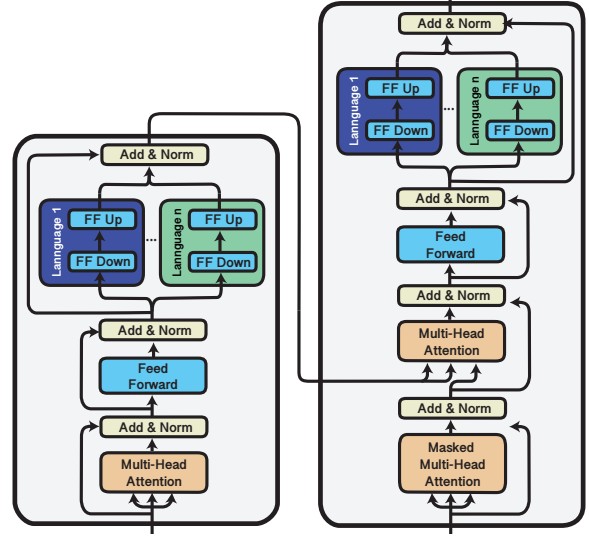

Figure 1: Architecture of mmT5. Language-specific bottleneck modules (dark blue and green components) are placed after the feed-forward component within each layer of the Transformer encoder-decoder model.

Natural language generation (NLG) tasks present another challenge for current multilingual models, which may overfit to the training languages and partially forget their generation ability in the target language (Vu et al., 2022), generating text with the *correct* meaning in the *wrong* language. We refer to this as the "**source language hallucination problem**".

To address these two limitations, we propose the modular multilingual T5 (mmT5, Figure 1), the first modular multilingual generative model. During pre-training, mmT5 allocates a small amount of language-specific parameters to increase capacity for multilingual modeling. At fine-tuning time, we freeze the language-specific modules while tuning the shared parameters, allowing direct adaptation to a target language by swapping to the corresponding language-specific module.

However, we observe an additional challenge for mmT5: the fine-tuned shared representations

may drift away from the frozen modular representations in the decoder. The modular model is thus susceptible to generating text in the incorrect language, similar to its non-modular counterparts. To ameliorate this, we propose to freeze a subset of shared decoder parameters, which shows large improvements in zero-shot cross-lingual generation for modular generative models.

In general, we find that mmT5 is an effective model that overcomes the two limitations of multilingual sequence-to-sequence models: **1)** mmT5 alleviates the curse of multilinguality by adding additional model capacity to different languages during pre-training. It outperforms both standard baselines as well as mT5 (Xue et al., 2021) at the same parameter sizes on a representative set of multilingual NLU and NLG tasks; **2)** mmT5 resolves the source language hallucination problem with impressive ability on zero-shot cross-lingual text generation. Our analysis (§6.4) shows that mT5 only generates text in the target language 7% of the time for a zero-shot multilingual summarization task, while mmT5 generates text in the correct language for 99% of examples.

## 2 Related work

**Modular language models**   Much work has focused on *post-hoc* modularity of pre-trained multilingual models, i.e., modular representations are added to existing dense models. The most commonly used modules are known as adapters (Rebuffi et al., 2017, 2018; Houlsby et al., 2019). They enable specialization to new data settings (Chen et al., 2019; Rücklé et al., 2020), combination of new and existing knowledge (Stickland and Murray, 2019; Wang et al., 2021a; Pfeiffer et al., 2021a; Lauscher et al., 2020a; Mahabadi et al., 2021b; Poth et al., 2021), and adaptation to new crosslingual (Pfeiffer et al., 2020, 2021b; Üstün et al., 2020; Vidoni et al., 2020; Ansell et al., 2021a, 2022; Wang et al., 2021b) and NMT scenarios (Bapna and Firat, 2019; Philip et al., 2020; Chronopoulou et al., 2020; Le et al., 2021; Üstün et al., 2021; Stickland et al., 2021; Garcia et al., 2021; Dua et al., 2022; Zhang et al., 2021; Pires et al., 2023).

Our approach, in contrast, uses modularity *a priori*, i.e., modularity is integrated into the module architecture as an inductive bias. Such modularity is similar to parameter sharing strategies commonly defined in multi-task learning (Ruder, 2017) as well as to mixture-of-experts approaches (MoE; Shazeer

et al., 2017), which have been used to scale models to trillion parameters (Fedus et al., 2022) and for domain-specific pre-training of LMs (Gururangan et al., 2022). The most related work to ours is X-Mod (Pfeiffer et al., 2022), which pre-trains an encoder-only BERT-style model in a modular fashion. Their model, however, cannot be used for natural language generation and underperforms our model on NLU tasks (see Section 4).

**Limitations of multilingual language models** State-of-the-art multilingual LMs are pre-trained on large amounts of multilingual data in around 100 languages. Prior work has demonstrated, however, that models' performance deteriorates with increasing language coverage given the same fixed capacity, known as the *curse of multilinguality* (Conneau et al., 2020b). Prior studies also found that models perform poorly on languages that are underrepresented in pre-training (Wu and Dredze, 2020; Hu et al., 2020; Lauscher et al., 2020b; Artetxe et al., 2020; Pfeiffer et al., 2020, 2021b; Chau et al., 2020; Ponti et al., 2020). For natural language generation, multilingual models have been observed to overfit to the source language and fail to generate text consistently in the correct target language (Vu et al., 2022).

## 3 mmT5

Standard multilingual models update the same model parameters for hundreds of languages during pre-training, resulting in the curse of multilinguality where different languages compete for the limited model capacity (Conneau et al., 2020a). We propose mmT5, the first modular sequence-to-sequence multilingual model that allocates language specific modules during pre-training. In this section, we discuss the architecture of mmT5, its training and fine-tuning methods, and our strategies to resolve the source language hallucination problem with mmT5.

### 3.1 Modeling

First, we describe the overall architecture of mmT5. We augment a standard Transformer encoder-decoder model with language-specific modules at every transformer layer (see Figure 1). The selection of modules (i.e., fixed routing; Pfeiffer et al., 2023) is performed via the language ID provided with each example[1]; all tokens of an example are

---

[1] Our pre-training data contains such metadata; alternatively, automatic language ID methods can be used (see §6.4).

passed through the same language-specific module.

We use bottleneck adapters as the language-specific module because they perform better at smaller model sizes compared to other modular methods such as continuous prompts (Mahabadi et al., 2021a; He et al., 2022). We place a module after the feed-forward component in each layer. In contrast to Pfeiffer et al. (2022) that only experimented with encoder-only models, we focus on a more general sequence-to-sequence model following the T5 architecture (Raffel et al., 2020).

We add $N \times L$ modular components to the T5 architecture where $L$ is the number of layers of the model and $N$ corresponds to the number of languages which the model is pre-trained on. The transformer weights are shared across languages while the modular component provides the model with language-specific capacity. During a forward pass, each input is first passed through the shared transformer weights and then routed through the corresponding language-specific module based on the language of the input. We follow this procedure for all transformer layers until the representations are passed to the shared prediction head.

### 3.2 Modular Pre-training, Fine-tuning, and Inference

We pre-train both language-specific modules and shared parameters jointly. During fine-tuning, we freeze all language-specific modules and only update the shared parameters. This paradigm allows us to more effectively adapt the fine-tuned model to any of the languages included in the pre-training data by simply switching to the corresponding language-specific module. At inference, the module corresponding to the target language is used together with the fine-tuned shared parameters.

### 3.3 Overcoming Modular Representation Drift

When fine-tuning the modular model for transfer settings in §5, we observe a scenario of *modular representation drift*: we find that the shared parameters that are updated during task-specific training drift away from the modular parameters and become thus less compatible with modules that are used for inference. In practice, this leads to a loss of compositional generalization where the modular model generates text in the incorrect language, similar to its non-modular counterparts (Vu et al., 2022); see §6.4.

In order to ameliorate this drift, we propose to freeze parts of the model, with a focus on the decoder. We find that freezing the decoder feed-forward parameters provides the biggest benefit (see §6.1 for the detailed ablation) and almost completely eliminates the source language hallucination problem in modular models.[2]

## 4 Experiments

**Pre-training Details** We pre-train mmT5 on data from 100 languages in mC4 (Xue et al., 2021) following the general pre-training setup of mT5 (Xue et al., 2021), if not specified otherwise. We pre-train mmT5 at two model sizes: small (300M parameters), and base (580M parameters). We train model variants with an input sequence length of 1024 and a target sequence length of 256 for 1M update steps with a batch size of 1024. The bottleneck size of each module is half of the hidden dimension of the transformer model. For instance, as the base variant has a hidden dimension of 768, we set the bottleneck size to 384.[3] We additionally pre-train a non-modular variant of our modular model, mT5$^S$, where all parameters are shared across all languages. The mT5$^S$ variant uses exactly the same hyper-parameters and pre-training setup as mmT5. To ensure that the models are directly comparable and have exactly the same number of parameters, we add shared bottleneck layers to mT5$^S$ in the same configuration as in mmT5.

**Experimental setting** We conduct experiments across datasets in *zero-shot cross-lingual transfer* and *multilingual training* scenarios. For *zero-shot cross-lingual transfer*, we train the model on a subset of languages (e.g. only English) and evaluate the model on held-out data of the same task in other languages. In *multilingual training*, we fine-tune the model on multiple languages of the same task, and evaluate the model on the same set of languages. As the language-specific modular components are replaced at inference time, we do not update the parameters of the modular components. We do the same for our shared model variants, in order for the number of trainable parameters to be equal for comparable scenarios.[4] For each dataset, we select the best model checkpoint based on performance on the validation set.

---

[2]We observe little benefit to freezing decoder parameters in non-modular models, however.

[3]We analyze the impact of bottleneck sizes in §6.2.

[4]Here, we follow the procedure of Pfeiffer et al. (2022).

| Model | Variant | Shared Params. | Mod. Params. per Lang. |
|---|---|---|---|
| mBERT | Base | 178M | – |
| X-Mod | Base | 270M | 7M |
| XLM-R | Base | 270M | – |
|  | Large | 550M | – |
| mT5 | Small | 300M | – |
|  | Base | 580M | – |
| mT5$^S$ | Small | 300M + 4M | – |
|  | Base | 580M + 14M | – |
| mmT5 | Small | 300M | 4M |
|  | Base | 580M | 14M |

Table 1: Number of shared and modular parameters of baselines and our models.

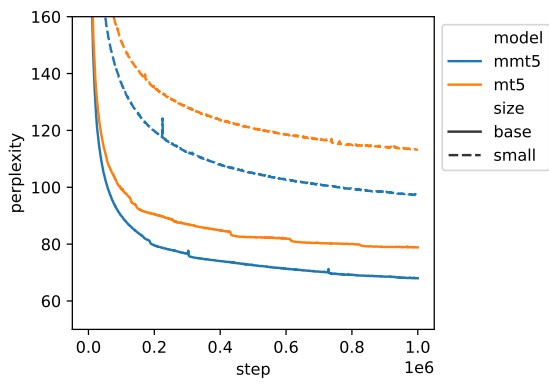

Figure 2: Perplexity (lower is better) of different model sizes during pre-training for mmT5 and mT5$^S$, averaged across languages.

**Evaluation Tasks** For *zero-shot cross-lingual transfer*, we evaluate on the XQuAD (Artetxe et al., 2020) and TyDi QA GoldP (Clark et al., 2020) question answering datasets; on the XNLI (Conneau et al., 2018) natural language inference dataset; on XL-Sum (Hasan et al., 2021) for summarization;[5] and MASSIVE (FitzGerald et al., 2023) for semantic parsing.[6] We mainly fine-tune the model on English training data and evaluate on the target languages (Hu et al., 2020). For XL-Sum, we additionally evaluate in a multi-source zero-shot transfer setting where we train jointly on data in Arabic, English, Japanese and Chinese (XL-Sum$^{ar,en,ja,zh}$).

For *multilingual training*, we evaluate on semantic parsing (MASSIVE) and summarization (XL-Sum) datasets. For each dataset, we fine-tune and evaluate the model on all languages jointly.

**Baselines** Our main comparison method is **mT5$^S$**, a shared model that is pre-trained with the same hyper-parameters, setup, and number of parameters as our modular model. We also compare to the published results of the **mT5** encoder-decoder model (Xue et al., 2021). In addition, we compare to several encoder-only models including **mBERT** (Devlin et al., 2019), **X-Mod** (Pfeiffer et al., 2022), and **XLM-R** (Conneau et al., 2020b). Encoder-only models are generally smaller as they lack a decoder but cannot easily be used for generation tasks. We provide an overview of the model sizes of the baselines and our method in Table 1.

---

[5]We do not evaluate on Oromo, Kirundi, Pidgin, and Tigrinya as they were not seen during pre-training.

[6]We do not evaluate on Hebrew and Tagalog as they were not seen during pre-training.

**Decoder Freezing Configurations** To overcome the modular representation drift described in §3.3, we experiment with different configurations of freezing parts of the model when fine-tuning the model on a downstream task. We experiment with freezing the LayerNorm (LN), self-attention (Att), cross-attention (CrossAtt) and feed-forward component (FFN) in the encoder (Enc) and decoder (Dec) parts of the transformer model. We ablate freezing configurations in §6.1 and report test results of the freezing configuration that performs best on the dev set for each dataset for mmT5. For dense models, we observe no impact with freezing and report results using full fine-tuning.

## 5 Results

### 5.1 Pre-training

We first compare the language modeling perplexities of different model sizes for mmT5 and mT5$^S$ during pre-training in Figure 2. We find that mmT5 significantly outperforms its fully shared counterpart during the early stages of pre-training and maintains the gap throughout the pre-training process. From an efficiency perspective, mmT5 only requires 282k and 220k update steps respectively for the small and base versions to achieve the same final perplexity as the mT5$^S$ models at 1M update steps. This corresponds to a $\approx 4\times$ efficiency boost when training a modular multilingual model compared to a fully dense one.

### 5.2 Fine-tuning

We present our main results on the test sets for zero-shot cross-lingual transfer and multilingual training scenarios in Tables 2 and 3, respectively.

| | Zero-Shot | | XQuAD F1 / EM | TyDiQA(GoldP) F1 / EM | XNLI Acc | XL-Sum$^{en}$ RG$_1$ / RG$_2$ / RG$_L$ | XL-Sum$^{ar,en,ja,zh}$ RG$_1$ / RG$_2$ / RG$_L$ | MASSIVE EM |
|---|---|---|---|---|---|---|---|---|
| Encoder | base | mBERT | 64.5 / 49.4 | 59.7 / 43.9 | 65.4 | — | — | — |
| | | X-Mod | 72.8* / — | — / — | 73.5* | — | — | — |
| | | XLM-R | 70.6 / 55.5 | — / — | 76.2 | — | — | — |
| | large | XLM-R | 76.6 / 60.8 | 65.1 / 45.0 | 79.2 | — | — | — |
| Encoder-decoder | small | mT5 | 58.1 / 42.5 | 35.2 / 23.2 | 67.5 | — | — | — |
| | | mT5$^S$ | 61.9 / 46.2 | 44.5 / 31.1 | 63.2 | 15.5 / 2.2 / 14.2 | 17.0 / 4.7 / 15.1 | 21.7 |
| | | mmT5 | **66.5 / 50.4** | **50.8 / 36.3** | **68.5** | **16.7** / **4.6** / **14.4** | **29.4 / 12.6 / 23.3** | **27.7** |
| | base | mT5 | 67.0 / 49.0 | 59.1 / 42.4 | 75.4 | — | — | 34.7 |
| | | mT5$^S$ | 68.7 / 51.5 | 64.0 / 47.8 | 75.1 | 16.2 / 2.8 / 4.5 | 18.6 / 6.0 / 16.7 | 39.9 |
| | | mmT5 | **76.3 / 60.3** | **69.0 / 53.2** | **77.8** | **19.6** / **6.1** / **16.4** | **34.5 / 16.1 / 26.8** | **46.0** |

Table 2: Zero-shot cross-lingual transfer test results averaged over all languages. mBERT and XLM-R scores are from (Hu et al., 2020); XLM-R Base XNLI results are from (Conneau et al., 2020b); mT5 results are from (Xue et al., 2021); X-Mod results are from (Pfeiffer et al., 2022) (*average is only on a subset of languages).

| | Multilingual | | XL-Sum RG$_1$ / RG$_2$ / RG$_L$ | MASSIVE EM |
|---|---|---|---|---|
| Enc-dec | small | mT5$^S$ | 36.4 / 17.9 / 28.5 | 60.7 |
| | | mmT5 | **36.7 / 18.1 / 28.7** | **65.6** |
| | base | mT5$^S$ | 39.1 / 20.3 / 30.5 | 64.6 |
| | | mmT5 | **41.6 / 22.8 / 33.0** | **66.7** |

Table 3: Multilingual training test results averaged over all languages.

mmT5 outperforms both the original mT5 as well as mT5$^S$ across all model sizes. It achieves performance similar to XLM-R at the same parameter size—despite its encoder-decoder configuration—and significantly outperforms X-Mod, the only other modular model.

**Zero-shot** For zero shot cross-lingual transfer scenarios, we see large gains for generative tasks in particular. For question answering (XQuAD and TyDiQA), we observe an average relative F1 improvement of 5.5 and 6.3 for the small and base models respectively. For summarization, we see larger zero-shot gains when jointly training on more than one language. We suspect that this is due to the increase in training data and due to positive transfer during multi-source training, which modular methods are better able to harness. This is in line with previous findings that multi-source training improves cross-lingual transfer in adapter-based setups (Ansell et al., 2021b). We also see a gain of 6.1 EM points on MASSIVE. The smallest gains are achieved for the classification task XNLI. Here, mmT5 improves over the baselines only by 1–2.4 accuracy points. We hypothesize that due to the constrained formulation of the task,

which only requires predicting a single token, the full multilingual generative capabilities of mmT5 are under-utilized. Overall, we see a clear trend that our modular models significantly outperform their respective dense counterparts especially for generation tasks.

**Multilingual training** For multilingual training in Table 3, we also find that the modular models outperform their dense counterparts across all tasks we experiment with. Here we find the largest gains for semantic parsing (MASSIVE). For summarization (XL-SUM), we see smaller, but still consistent gains. These results indicate that modular representations are not only useful in transfer settings but that mmT5 can also leverage labeled data in the target language to deliver superior performance compared to the standard non-modular models.

## 6 Analysis and Ablations

### 6.1 Impact of Freezing Configuration

We investigate the impact of the freezing configuration on the performance of the model. In Table 5, we compare the best-performing freezing configurations with a non-frozen baseline for mmT5 base (we show the results of all freezing configurations in Appendix A.1). We observe significant improvements when freezing the feed-forward layer of the decoder during fine-tuning, particularly in zero-shot scenarios. For multilingual training, freezing of the decoder has less effect on the performance. We also find that freezing parts of the decoder has no effect on the dense mT5$^S$ model across all tasks (see Appendix A.1).

| Emb | Enc$_{LN}$ | Dec$_{LN}$ | Dec$_{Att}$ | Dec$_{CrossAtt}$ | Dec$_{FFN}$ | Zero-Shot | | | | Multilingual |
| | | | | | | XQuAD dev (en) f1 / em | XQuAD test f1 / em | XNLI dev acc | MASSIVE dev EM | XL-Sum dev Rg$_1$ / Rg$_2$ / Rg$_L$ |
|---|---|---|---|---|---|---|---|---|---|---|
| | | | | | | 90.7 / 83.6 | 66.9 / 49.3 | 75.5 | 32.1 | 41.2 / 22.4 / 32.4 |
| ✗ | | ✗ | | ✗ | ✗ | 91.9 / 85.1 | 75.8 / 59.5 | 75.6 | 43.2 | 41.2 / 22.4 / 32.6 |
| ✗ | | ✗ | | | ✗ | 91.8 / 85.1 | 75.8 / 59.8 | 77.3 | 41.0 | **41.9 / 23.1 / 33.2** |
| ✗ | ✗ | ✗ | | ✗ | ✗ | **92.1 / 85.5** | **76.3 / 60.3** | 76.1 | **45.4** | 40.8 / 22.1 / 32.3 |
| ✗ | ✗ | ✗ | | | ✗ | 91.8 / 85.1 | 75.0 / 59.2 | **77.7** | 39.9 | 41.8 / 23.0 / 33.1 |

Table 4: Results of different freezing configurations for mmT5 base on different tasks. Dev results for most. We always fine-tune Enc$_{Att}$, and Enc$_{FFN}$ and always freeze Enc$_{Mod}$ and Dec$_{Mod}$. ✗ indicates that this component is frozen during task-level fine-tuning.

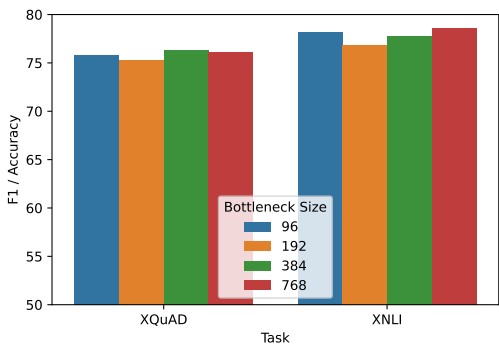

Figure 3: Comparison of bottleneck sizes of base mmT5 for XQuAD (F1) and XNLI (Accuracy).

Figure 4: Comparison of model sizes for XQuAD (F1) and XNLI (Accuracy).

## 6.2 Impact of Bottleneck Size

We experiment with different bottleneck sizes of the modular components to understand the impact of providing each language with more capacity. We report results for XQuAD, and XNLI in Figure 3 using mmT5 base and bottleneck sizes of 96, 192, 384, and 768. We find that for all three tasks the bottleneck size has little effect on the downstream task performance, achieving only 0.5–2 absolute points difference between the larger and the smaller bottleneck sizes. This suggests that it is sufficient to provide the model with only a small amount of language-specific parameters in order to learn idiosyncratic information and mitigate catastrophic interference, and highlights the parameter-efficiency of modular models.

## 6.3 Impact of Model Size

In Figure 4, we plot the performance difference of mmT5 and mT5$^S$ for the small and base variants. We find that the modular model outperforms the dense variant across model sizes with a similar gap, indicating that the positive effect of modularity may not diminish at scale.

## 6.4 Source Language Hallucination

We perform an analysis of the generated text on the XL-Sum dev sets for mT5$^S$ and mmT5 models trained in a zero-shot setting on XL-Sum$^{ar,en,ja,zh}$ using full fine-tuning and a decoder freezing configuration. We automatically detect the language of the generated text using the Language Detection from the Google Cloud Translation API[7] (Caswell et al., 2020). We show the results in Figure 6. We find that most models tend to generate text in one of the source languages (in this setting: Arabic, English, Japanese, and Chinese). This holds true also for mmT5 when we fine-tune the decoder. However, when freezing the decoder we observe a dramatic improvement in the target language generation rate from 1% to 99% of examples for mmT5, essentially solving the issue of source language hallucination in cross-lingual transfer scenarios. This improvement in language consistency also helps explain the significant improvement of the modular model over its dense counterparts on natural language generation tasks.

[7] https://cloud.google.com/translate/docs/basic/detecting-language

Südkalifornien besteht aus [. . . ] einer internationalen Metropolregion und Großstadtgebieten. Die Region ist die Heimat von `zwei` erweiterten Metropolregionen mit jeweils mehr als fünf Millionen Einwohnern. [. . . ]
**Question**: Wie viele erweiterte Metropolregionen gibt es ?

- - - - - - - - - - - - - - - - - - - - - - - - - - -

**mmT5**: `zwei`
**mT5**$^S$: `two`

[. . . ] Analysen [. . . ] waren irreführend, da es `mehrere Jahre` dauert, bis die Auswirkungen zu Veränderungen des Wirtschaftswachstums führen. [. . . ]
**Question**: Wie lange dauert es, bis sich die Auswirkungen als Veränderungen des wirtschaftlichen Wachstums manifestieren?

- - - - - - - - - - - - - - - - - - - - - - - - - - -

**mmT5**: `mehrere Jahre`
**mT5**$^S$: `more` `ere Jahre`

Figure 5: XQuAD examples where mT5$^S$ generates tokens with the correct meaning but in the wrong language. For the same examples, mmT5 is able to generate tokens in the correct language when freezing parts of the decoder.

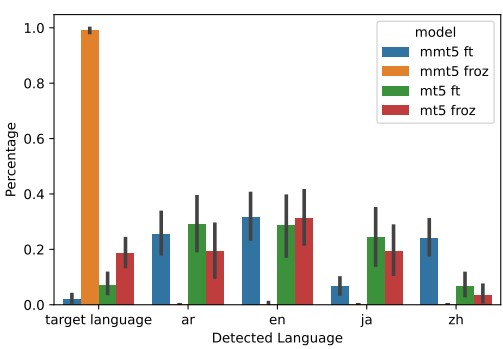

Figure 6: Detected languages of generated text of the development set of XL-Sum$^{ar,en,ja,zh}$. All models have base size. *ft indicates that the decoder was fine-tuned, *froz indicates that the decoder was partially frozen. High numbers are desireable for the first set of plots ("target language"), low numbers are desireable for the remaining four sets of plots ("ar", "en", "ja", "zh"). We only include zero-shot cross-lingual results, therefore exclude the four source languages; all models achieve 100% accuracy for those. For more granular results see Appendix Table 10.

In addition, we manually analyze outputs of mT5$^S$ and mmT5 on XQuAD and find similar issues of source language hallucinations. We show examples in Figure 5. Although the task is extractive QA, i.e., the answer is a substring of the input, mT5$^S$ tends to translate subwords into English (the source language). This does not happen to mmT5 when freezing parts of the decoder, partially explaining the large improvements of mmT5 over mT5$^S$ on TyDi QA in Table 2.

### 6.5 Module Re-Use for Unseen Languages

In the previous sections we have evaluated the cross-lingual performance of mmT5 on languages seen during pre-training. However, with more than 7000 languages spoken in the world (Joshi et al., 2020), mmT5 covers less than 1% of them. While extending the model to unseen languages is out of scope for this work[8], we evaluate the potential reusability of existing language modules for truly unseen languages with a case study on Tagalog. We utilize the base mmT5 model fine-tuned on the English MASSIVE training dataset (see Table 2). As a Tagalog language module does not exist within mmT5, we test all existing *other* language modules when evaluating on the Tagalog test set. In Figure 7, we report the Exact Match (EM) zero-shot accuracies for all languages. The module performing best corresponds to Javanese, which is the most closely related language to Tagalog as both belong to the Malayo-Polynesian subgroup of the Austronesian language family. This finding demonstrates the effectiveness of modular models; modular components specifically incorporate interpretable concepts, which can be re-used for unseen scenarios. Additionally, they can be further fine-tuned or adapted to the target domain if training data is available.

## 7 Conclusion

We have proposed mmT5, a modular multilingual encoder-decoder model. During multilingual pre-training the majority of parameters of mmT5 are shared between tasks, but each language is provided with a small amount of parameters only accessible to the respective language. We demonstrated that integrating modularity as an architec-

---

[8]Previous work has demonstrated that it is possible to extend multilingual models to unseen languages (Pfeiffer et al., 2020, 2021b, 2022).

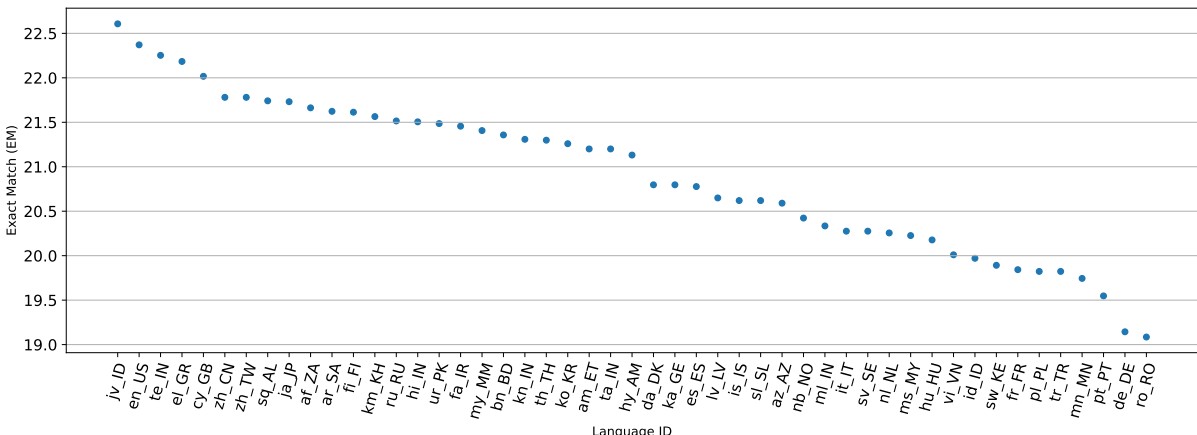

Figure 7: Average of the top-5 zero-shot EM accuracies on the Tagalog MASSIVE development set by varying the input language ID. Tagalog was not seen during mmT5 pre-training.

tural inductive bias significantly improves training efficiency, where the same perplexity as an equivalent fully dense model is achieved at a quarter of the update steps. mmT5 considerably outperforms comparable models on a large number of tasks including Question Answering, Semantic Parsing, Summarization and Classification in both zero-shot as well as multilingual scenarios. Finally, we show that by freezing parts of the decoder when fine-tuning mmT5 on a target task in a source language, the model consistently generates text in the target language. Consequently, modularity arguably solves source language hallucinations in cross-lingual transfer scenarios.

## 8 Limitations and Future Work

In this paper, we explored the use of modularity for multilingual language models. We showed that modularity significantly improves cross-lingual performance on a number of generative tasks by mitigating hallucinations in the source language. However, there are still many avenues for future work.

First, we did not consider placing the modules in different parts of the model. We only experimented with placing bottleneck layers after the feed-forward component of each transformer layer. Previous work has demonstrated that depending on the modality, different placements perform better (Pfeiffer et al., 2021a; Eichenberg et al., 2022).

Second, we only experimented with extending the vanilla transformer architecture with modular components. Future work might consider modularizing different parts of the transformer, such as the attention-components or entire feed-forward layers like in Kudugunta et al. (2021).

Third, we performed fixed routing under the assumption that the language ID is easy to obtain. We chose this path, as learning-to-route has many difficulties such as training instabilities (Pfeiffer et al., 2023). However, this architecture design limits the sharing of information (e.g. domains) across languages. Consequently, a combination of fixed routing and learned routing would allow the model to learn how to share information across subsets of languages.

Fourth, we did not try using mmT5 for machine translation. Using a modular design for this type of task setup is quite natural, as modules from the encoder and decoder can be easily replaced with the source and target language components, respectively. The effectiveness of modular sequence-to-sequence models for NMT has been investigated previously (Bapna and Firat, 2019; Philip et al., 2020; Chronopoulou et al., 2020; Le et al., 2021; Üstün et al., 2021; Stickland et al., 2021; Garcia et al., 2021; Dua et al., 2022).

Finally, we did not consider extending the model to languages beyond those we pre-trained on. While our preliminary results (see § 6.5) suggest that there are benefits of reusing related language modules to learn unseen languages, this requires further experimentation. However, previous works have demonstrated that modular (Pfeiffer et al., 2022) as well as dense models can be adapted to new languages and scripts (Pfeiffer et al., 2020, 2021b). Alternatively, future work might consider using post-hoc adaptation techniques, such as LoRA (Hu et al., 2022), to adapt modules to new languages.

## Acknowledgements

We thank Andrea Gesmundo, Marc'Aurelio Ranzato, Srini Narayanan, and Emanuele Bugliarello for helpful feedback on a draft of this paper.

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

# A  Appendix

## A.1  Freezing combinations

We show results with different freezing combinations in Table 5. We find that freezing the FFN component of the Decoder results in the biggest performance gains.

## A.2  Language-ID prediction on Cross-lingual Summarization

We report the languages predicted by the Language Detection model from the Google Cloud Translation API[9] (Caswell et al., 2020) for the XL-Sum$^{ar,en,ja,zh}$ task in Table 10. We find that mmT5 achieves near perfect performance for all target languages when freezing parts of the decoder (s7)–99% of the text is generated in the correct target language–significantly outperforming all other model variants. Interestingly, mmT5 hallucinates in the source language when the decoder is fine-tuned (s1), resulting in a drop down to only 2% in the correct target language. mT5$^S$ also benefits slightly from freezing parts of the decoder, with an improvement from 7% to 18% target language generation, however, this is no where close to the performance of mmT5.

## A.3  Language-level Results

**XNLI.**  We report XNLI validation results in Table 11 and test results in Table 6.

**XQuAD.**  We report XQuAD validation results in Table 9 and test results in Table 7.

**MASSIVE.**  We report MASSIVE validation results in Table 17 and test results in Table 8.

**TyDiQA.**  We report TyDiQA validation results in Table 23.

**Multilingual XL-Sum**  We report XL-Sum validation results in Tables 18, 19,20, 21, and 22 and test results in Table 15.

**Zeroshot XL-Sum$^{en}$**  We report XL-Sum validation results in Table 14 and test results in Table 13.

**Zeroshot XL-Sum$^{ar,en,ja,zh}$**  We report XL-Sum validation results in Table 16 and test results in Table 12.

## A.4  Language-level Pre-training Perplexities

We report the language-level perplexities of the different model variants and sizes in Figures 8, 9, 10, 11, 12, 13, 14.

---

[9] https://cloud.google.com/translate/docs/basic/detecting-language

| cfg | Emb | Enc$_{LN}$ | Dec$_{LN}$ | Dec$_{Att}$ | Dec$_{CrossAtt}$ | Dec$_{FFN}$ | XQuAD dev (en) f1 / em | XQuAD test f1 / em | XNLI dev acc | Zero-Shot XL-Sum$^{en}$ dev Rg$_1$ / Rg$_2$ / Rg$_L$ | XL-Sum$^{ar,en,zh,ja}$ dev Rg$_1$ / Rg$_2$ / Rg$_L$ | MASSIVE dev EM | Multi-Source XL-Sum dev Rg$_1$ / Rg$_2$ / Rg$_L$ | MASSIVE dev EM |
|---|---|---|---|---|---|---|---|---|---|---|---|---|---|---|
| s1 | | | | | | | 90.7 / 83.6 | 66.9 / 49.3 | 75.5 | 15.4 / 2.0 / 14.0 | 18.7 / 6.1 / 16.8 | 32.1 | 41.2 / 22.4 / 32.4 | |
| s2 | | ✗ | ✗ | | | | 90.7 / 83.4 | 65.6 / 48.0 | 75.0 | | | | | |
| s3 | ✗ | | | | | | 90.7 / 83.5 | 61.0 / 43.4 | 76.9 | | | | | |
| s4 | ✗ | ✗ | ✗ | | | | 90.9 / 83.6 | 64.6 / 47.1 | 77.5 | | | | | |
| s5 | ✗ | | ✗ | ✗ | ✗ | ✗ | 91.2 / 84.1 | 74.3 / 57.5 | 73.8 | | | | | |
| s6 | ✗ | | ✗ | | ✗ | ✗ | 91.9 / 85.1 | 75.8 / 59.5 | 75.6 | | | 43.2 | 41.2 / 22.4 / 32.6 | |
| s7 | ✗ | | ✗ | | | ✗ | 91.8 / 85.1 | 75.8 / 59.8 | 77.3 | **19.7 / 6.2 / 16.4** | **34.7 / 16.2 / 26.9** | 41.0 | **41.9 / 23.1 / 33.2** | |
| s8 | | ✗ | ✗ | | ✗ | ✗ | 91.2 / 84.5 | 75.0 / 59.3 | 73.3 | | | | | |
| s9 | | ✗ | ✗ | | | ✗ | 91.2 / 84.5 | 74.6 / 58.8 | 75.6 | | | | | |
| s10 | ✗ | ✗ | ✗ | | ✗ | ✗ | **92.1 / 85.5** | **76.3 / 60.3** | 76.1 | | | **45.4** | 40.8 / 22.1 / 32.3 | 66.78 |
| s11 | | | ✗ | ✗ | ✗ | ✗ | 90.9 / 84.0 | 74.8 / 58.8 | 73.1 | | | | | |
| s12 | | | ✗ | ✗ | | ✗ | 91.2 / 84.5 | 75.0 / 59.3 | 75.6 | | | | | |
| s13 | | | ✗ | | | ✗ | 91.3 / 84.5 | 74.9 / 58.9 | 76.3 | | | | | |
| s14 | ✗ | ✗ | ✗ | | | ✗ | 91.8 / 85.1 | 75.0 / 59.2 | **77.7** | | | 39.9 | 41.8 / 23.0 / 33.1 | |

Table 5: Results with different freezing combinations of mmT5 base on different tasks. ✗ indicates that the component is frozen in the respective configuration Dev results for most. We always finetune the attention in the encoder (Enc$_{Att}$), the feed forward layer in the encoder (Enc$_{FFN}$), and always freeze the modules in the encoder (Enc$_{Mod}$) and decoder (Dec$_{Mod}$). We find that configurations s1-s4 strongly underperform the respective other configurations (s5-s14), suggesting that freezing the feed forward layer of the decoder is essential for good cross-lingual transfer performance.

| | model | ar | bg | de | el | en | es | fr | hi | ru | sw | th | tr | ur | vi | zh | *avg* |
|---|---|---|---|---|---|---|---|---|---|---|---|---|---|---|---|---|---|
| small | mmT5 | **65.3** | **71.9** | **70.2** | **70.5** | **81.8** | **74.7** | **73.4** | **62.7** | **70.1** | **63.8** | **67.4** | **64.2** | **59.1** | **66.7** | **66.3** | **68.5** |
| small | mT5$^S$ | 63.8 | 69.1 | 67.6 | 68.1 | 80.0 | 71.6 | 69.3 | 60.4 | 68.7 | 53.2 | 64.1 | 59.1 | 58.4 | 63.9 | 64.4 | 65.5 |
| base | mmT5 | **75.0** | **81.2** | **80.3** | **79.5** | **86.9** | **82.6** | **80.9** | **73.4** | **78.3** | **74.0** | **74.9** | **76.3** | **69.9** | **76.7** | **77.2** | **77.8** |
| base | mT5$^S$ | 72.7 | 78.7 | 77.6 | 77.3 | 85.5 | 80.7 | 78.5 | 70.7 | 77.7 | 66.6 | 73.0 | 72.8 | 67.7 | 73.8 | 73.5 | 75.1 |

Table 6: XNLI test results for all language. We select the checkpoint performing best on the validation set.

| | model | ar F1 / EM | de F1 / EM | el F1 / EM | en F1 / EM | es F1 / EM | hi F1 / EM | ru F1 / EM | th F1 / EM | tr F1 / EM | vi F1 / EM | zh F1 / EM | *avg* F1 / EM |
|---|---|---|---|---|---|---|---|---|---|---|---|---|---|
| small | mmT5 | **60.6 / 44.6** | **71.0 / 53.6** | **64.9 / 47.1** | **82.5 / 70.3** | **74.1 / 56.1** | **59.2 / 43.9** | **69.5 / 50.8** | **58.9 / 47.0** | **62.4 / 43.4** | **64.3 / 45.4** | 64.2 / 52.4 | **66.5 / 50.4** |
| small | mT5$^S$ | 53.5 / 37.1 | 67.2 / 48.7 | 59.5 / 41.1 | 81.7 / 69.7 | 69.8 / 53.7 | 54.7 / 40.8 | 62.8 / 44.5 | 50.3 / 37.9 | 57.2 / 39.3 | 59.7 / 40.9 | **64.7 / 54.0** | 61.9 / 46.2 |
| base | mmT5 | **74.2 / 57.6** | **79.5 / 63.0** | **77.6 / 59.9** | **86.7 / 74.5** | **79.2 / 61.3** | **72.4 / 56.1** | **77.6 / 58.7** | **69.3 / 59.3** | **74.5 / 55.9** | **74.2 / 54.2** | **74.4 / 63.1** | **76.3 / 60.3** |
| base | mT5$^S$ | 63.3 / 43.0 | 75.9 / 57.2 | 63.3 / 40.3 | 84.3 / 71.9 | 76.1 / 58.7 | 62.8 / 47.1 | 64.0 / 42.6 | 59.6 / 48.5 | 70.1 / 51.7 | 70.4 / 50.4 | 66.0 / 55.5 | 68.7 / 51.5 |

Table 7: XQuAD test set results for all languages. We select the checkpoint performing best on the English development set.

| Language | Exact Match (EM) |
|---|---|
| af_ZA | 57.5 |
| am_ET | 29.6 |
| ar_SA | 38.3 |
| az_AZ | 41.3 |
| bn_BD | 37.2 |
| cy_GB | 35.5 |
| da_DK | 60.5 |
| de_DE | 55.3 |
| el_GR | 49.6 |
| en_US | 72.7 |
| es_ES | 53.8 |
| fa_IR | 48.2 |
| fi_FI | 54.4 |
| fr_FR | 51.4 |
| hi_IN | 44.1 |
| hu_HU | 47.4 |
| hy_AM | 38.6 |
| id_ID | 57.1 |
| is_IS | 42.8 |
| it_IT | 51.7 |
| ja_JP | 42.5 |
| jv_ID | 38.1 |
| ka_GE | 38.9 |
| km_KH | 40.7 |
| kn_IN | 34.4 |
| ko_KR | 39.1 |
| lv_LV | 50.3 |
| ml_IN | 36.0 |
| mn_MN | 34.2 |
| ms_MY | 52.5 |
| my_MM | 33.8 |
| nb_NO | 58.2 |
| nl_NL | 57.5 |
| pl_PL | 52.9 |
| pt_PT | 56.0 |
| ro_RO | 55.4 |
| ru_RU | 50.9 |
| sl_SL | 50.3 |
| sq_AL | 48.3 |
| sv_SE | 58.9 |
| sw_KE | 43.0 |
| ta_IN | 37.1 |
| te_IN | 35.4 |
| th_TH | 50.1 |
| tr_TR | 47.9 |
| ur_PK | 39.6 |
| vi_VN | 44.9 |
| zh_CN | 30.0 |
| zh_TW | 28.2 |
| Average | 46.0 |

Table 8: MASSIVE Exact Match (EM) test accuracies of the best model (s10 modular) for all languages

|  |  |  | en |
|---|---|---|---|
|  |  | cfg | F1 / EM |
| Small | mmT5 | s1 | 85.6 / 77.6 |
|  |  | s6 | 87.2 / 79.4 |
|  |  | s7 | **87.4** / 79.4 |
|  |  | s10 | 87.3 / **79.6** |
|  |  | s14 | **87.4** / 79.2 |
|  | mT5$^S$ | s1 | 84.9 / 76.5 |
|  |  | s6 | 85.8 / 77.6 |
|  |  | s7 | 85.9 / 77.7 |
|  |  | s10 | 86.1 / 77.9 |
|  |  | s14 | 85.9 / 77.6 |
| Base | mmT5 | s1 | 90.7 / 83.6 |
|  |  | s2 | 90.7 / 83.4 |
|  |  | s3 | 90.7 / 83.5 |
|  |  | s4 | 90.9 / 83.6 |
|  |  | s5 | 91.2 / 84.1 |
|  |  | s6 | 91.9 / 85.1 |
|  |  | s7 | 91.8 / 85.1 |
|  |  | s8 | 91.2 / 84.5 |
|  |  | s9 | 91.2 / 84.5 |
|  |  | s10 | **92.1** / **85.5** |
|  |  | s11 | 90.9 / 84.0 |
|  |  | s12 | 91.2 / 84.5 |
|  |  | s13 | 91.3 / 84.5 |
|  |  | s14 | 91.8 / 85.1 |
|  | mT5$^S$ | s1 | 89.9 / 82.5 |
|  |  | s6 | 90.2 / 83.0 |
|  |  | s7 | 90.5 / 83.6 |
|  |  | s10 | 90.2 / 82.8 |
|  |  | s14 | 90.4 / 83.5 |

Table 9: XQuAD validation results for English across the different freezing configurations.

**Group 1**

| tgt lang |  | am | | | | ar | | | | az | | | | bn | | | | cy | | | |
|---|---|---|---|---|---|---|---|---|---|---|---|---|---|---|---|---|---|---|---|---|---|
| pred lang | cfg | *am* | ar | en | ja | zh | *ar* | ar | en | ja | zh | *az* | ar | en | ja | zh | *bn* | ar | en | ja | zh |
| mmT5 | s1 | 0.00 | 0.50 | 0.07 | 0.02 | 0.36 | **1.0** | 1.0 | 0.0 | 0.0 | 0.0 | 0.00 | 0.18 | 0.29 | 0.06 | 0.29 | 0.00 | 0.4 | 0.10 | 0.05 | 0.41 |
| mmT5 | s7 | **0.99** | 0.00 | 0.00 | 0.00 | 0.00 | **1.0** | 1.0 | 0.0 | 0.0 | 0.0 | **1.00** | 0.00 | 0.00 | 0.00 | 0.00 | **1.00** | 0.0 | 0.00 | 0.00 | 0.00 |
| mT5$^S$ | s1 | 0.03 | 0.96 | 0.00 | 0.00 | 0.01 | **1.0** | 1.0 | 0.0 | 0.0 | 0.0 | 0.08 | 0.12 | 0.31 | 0.27 | 0.09 | 0.02 | 0.3 | 0.02 | 0.43 | 0.19 |
| mT5$^S$ | s7 | 0.02 | 0.95 | 0.00 | 0.00 | 0.00 | **1.0** | 1.0 | 0.0 | 0.0 | 0.0 | 0.36 | 0.03 | 0.35 | 0.05 | 0.02 | 0.13 | 0.1 | 0.13 | 0.47 | 0.11 |

| tgt lang |  | cy | | | |
|---|---|---|---|---|---|
| pred lang | cfg | *cy* | ar | en | ja | zh |
| mmT5 s1 | | 0.06 | 0.05 | 0.84 | 0.0 | 0.01 |
| mmT5 s7 | | **0.91** | 0.00 | 0.08 | 0.0 | 0.00 |
| mT5$^S$ s1 | | 0.13 | 0.10 | 0.76 | 0.0 | 0.00 |
| mT5$^S$ s7 | | 0.22 | 0.01 | 0.76 | 0.0 | 0.00 |

**Group 2**

| tgt lang |  | en | | | | es | | | | fa | | | | fr | | | | gd | | | |
|---|---|---|---|---|---|---|---|---|---|---|---|---|---|---|---|---|---|---|---|---|---|
| pred lang | cfg | *en* | ar | en | ja | zh | *es* | ar | en | ja | zh | *fa* | ar | en | ja | zh | *fr* | ar | en | ja | zh |
| mmT5 | s1 | **1.0** | 0.0 | 1.0 | 0.0 | 0.0 | 0.03 | 0.03 | 0.73 | 0.01 | 0.07 | 0.02 | 0.96 | 0.01 | 0.0 | 0.0 | 0.05 | 0.07 | 0.74 | 0.0 | 0.02 |
| mmT5 | s7 | **1.0** | 0.0 | 1.0 | 0.0 | 0.0 | **0.99** | 0.00 | 0.00 | 0.00 | 0.00 | **1.00** | 0.00 | 0.00 | 0.0 | 0.00 | **1.00** | 0.00 | 0.00 | 0.00 | 0.00 |
| mT5$^S$ | s1 | **1.0** | 0.0 | 1.0 | 0.0 | 0.0 | 0.02 | 0.00 | 0.95 | 0.00 | 0.00 | 0.01 | 0.99 | 0.00 | 0.0 | 0.0 | 0.01 | 0.00 | 0.98 | 0.0 | 0.00 |
| mT5$^S$ | s7 | **1.0** | 0.0 | 1.0 | 0.0 | 0.0 | 0.05 | 0.00 | 0.91 | 0.00 | 0.00 | 0.13 | 0.86 | 0.00 | 0.0 | 0.0 | 0.04 | 0.00 | 0.94 | 0.0 | 0.00 |

| tgt lang |  | gd | | | |
|---|---|---|---|---|---|
| pred lang | cfg | *gd* | ar | en | ja | zh |
| mmT5 s1 | | 0.25 | 0.01 | 0.67 | 0.0 | 0.01 |
| mmT5 s7 | | **1.00** | 0.00 | 0.00 | 0.0 | 0.00 |
| mT5$^S$ s1 | | 0.23 | 0.01 | 0.76 | 0.0 | 0.00 |
| mT5$^S$ s7 | | 0.56 | 0.00 | 0.43 | 0.0 | 0.00 |

**Group 3**

| tgt lang |  | gu | | | | ha | | | | hi | | | | id | | | | ig | | | |
|---|---|---|---|---|---|---|---|---|---|---|---|---|---|---|---|---|---|---|---|---|---|
| pred lang | cfg | *gu* | ar | en | ja | zh | *ha* | ar | en | ja | zh | *hi* | ar | en | ja | zh | *id* | ar | en | ja | zh |
| mmT5 | s1 | 0.00 | 0.24 | 0.18 | 0.10 | 0.41 | 0.03 | 0.17 | 0.62 | 0.01 | 0.01 | 0.00 | 0.33 | 0.26 | 0.04 | 0.30 | 0.01 | 0.13 | 0.62 | 0.03 | 0.13 |
| mmT5 | s7 | **1.00** | 0.00 | 0.00 | 0.00 | 0.00 | **1.00** | 0.00 | 0.00 | 0.00 | 0.00 | **1.00** | 0.00 | 0.00 | 0.00 | 0.00 | **0.98** | 0.00 | 0.00 | 0.00 | 0.00 |
| mT5$^S$ | s1 | 0.13 | 0.16 | 0.01 | 0.56 | 0.13 | 0.11 | 0.03 | 0.83 | 0.00 | 0.00 | 0.03 | 0.23 | 0.02 | 0.64 | 0.05 | 0.07 | 0.14 | 0.73 | 0.01 | 0.02 |
| mT5$^S$ | s7 | 0.23 | 0.05 | 0.12 | 0.49 | 0.06 | 0.18 | 0.01 | 0.76 | 0.00 | 0.00 | 0.12 | 0.08 | 0.23 | 0.48 | 0.01 | 0.25 | 0.03 | 0.67 | 0.00 | 0.00 |

| tgt lang |  | ig | | | |
|---|---|---|---|---|---|
| pred lang | cfg | *ig* | ar | en | ja | zh |
| mmT5 s1 | | 0.10 | 0.25 | 0.33 | 0.02 | 0.17 |
| mmT5 s7 | | **1.00** | 0.00 | 0.00 | 0.00 | 0.00 |
| mT5$^S$ s1 | | 0.63 | 0.01 | 0.34 | 0.00 | 0.00 |
| mT5$^S$ s7 | | 0.66 | 0.00 | 0.31 | 0.00 | 0.00 |

**Group 4**

| tgt lang |  | ja | | | | ko | | | | ky | | | | mr | | | | my | | | |
|---|---|---|---|---|---|---|---|---|---|---|---|---|---|---|---|---|---|---|---|---|---|
| pred lang | cfg | *ja* | ar | en | ja | zh | *ko* | ar | en | ja | zh | *ky* | ar | en | ja | zh | *mr* | ar | en | ja | zh |
| mmT5 | s1 | **1.0** | 0.0 | 0.0 | 1.0 | 0.0 | 0.00 | 0.01 | 0.04 | 0.50 | 0.43 | 0.00 | 0.05 | 0.12 | 0.13 | 0.65 | 0.00 | 0.12 | 0.24 | 0.04 | 0.55 |
| mmT5 | s7 | **1.0** | 0.0 | 0.0 | 1.0 | 0.0 | **1.00** | 0.00 | 0.00 | 0.00 | 0.00 | **0.99** | 0.00 | 0.00 | 0.00 | 0.00 | **1.00** | 0.00 | 0.00 | 0.00 | 0.00 |
| mT5$^S$ | s1 | **1.0** | 0.0 | 0.0 | 1.0 | 0.0 | 0.01 | 0.00 | 0.00 | 0.99 | 0.00 | 0.02 | 0.15 | 0.04 | 0.56 | 0.14 | 0.07 | 0.09 | 0.05 | 0.64 | 0.08 |
| mT5$^S$ | s7 | **1.0** | 0.0 | 0.0 | 1.0 | 0.0 | 0.02 | 0.00 | 0.00 | 0.98 | 0.00 | 0.08 | 0.02 | 0.24 | 0.40 | 0.07 | 0.11 | 0.02 | 0.19 | 0.37 | 0.08 |

| tgt lang |  | my | | | |
|---|---|---|---|---|---|
| pred lang | cfg | *my* | ar | en | ja | zh |
| mmT5 s1 | | 0.00 | 0.24 | 0.03 | 0.30 | 0.27 |
| mmT5 s7 | | **1.00** | 0.00 | 0.00 | 0.00 | 0.00 |
| mT5$^S$ s1 | | 0.05 | 0.03 | 0.00 | 0.91 | 0.01 |
| mT5$^S$ s7 | | 0.20 | 0.00 | 0.00 | 0.78 | 0.01 |

**Group 5**

| tgt lang |  | ne | | | | pa | | | | ps | | | | pt | | | | ru | | | |
|---|---|---|---|---|---|---|---|---|---|---|---|---|---|---|---|---|---|---|---|---|---|
| pred lang | cfg | *ne* | ar | en | ja | zh | *pa* | ar | en | ja | zh | *ps* | ar | en | ja | zh | *pt* | ar | en | ja | zh |
| mmT5 | s1 | 0.00 | 0.25 | 0.10 | 0.10 | 0.47 | 0.01 | 0.22 | 0.22 | 0.07 | 0.42 | 0.03 | 0.92 | 0.00 | 0.0 | 0.0 | 0.04 | 0.08 | 0.74 | 0.01 | 0.03 |
| mmT5 | s7 | **0.99** | 0.00 | 0.00 | 0.00 | 0.00 | **1.00** | 0.00 | 0.00 | 0.00 | 0.00 | **1.00** | 0.00 | 0.00 | 0.0 | 0.0 | **1.00** | 0.00 | 0.00 | 0.00 | 0.00 |
| mT5$^S$ | s1 | 0.00 | 0.15 | 0.00 | 0.67 | 0.14 | 0.05 | 0.24 | 0.04 | 0.48 | 0.16 | 0.02 | 0.96 | 0.00 | 0.0 | 0.0 | 0.02 | 0.00 | 0.97 | 0.00 | 0.00 |
| mT5$^S$ | s7 | 0.01 | 0.02 | 0.03 | 0.82 | 0.09 | 0.18 | 0.12 | 0.19 | 0.40 | 0.03 | 0.12 | 0.82 | 0.02 | 0.0 | 0.0 | 0.07 | 0.00 | 0.91 | 0.00 | 0.00 |

| tgt lang |  | ru | | | |
|---|---|---|---|---|---|
| pred lang | cfg | *ru* | ar | en | ja | zh |
| mmT5 s1 | | 0.00 | 0.50 | 0.21 | 0.02 | 0.20 |
| mmT5 s7 | | **1.00** | 0.00 | 0.00 | 0.00 | 0.00 |
| mT5$^S$ s1 | | 0.07 | 0.53 | 0.28 | 0.01 | 0.04 |
| mT5$^S$ s7 | | 0.27 | 0.05 | 0.54 | 0.00 | 0.00 |

**Group 6**

| tgt lang |  | si | | | | so | | | | sr | | | | sw | | | | ta | | | |
|---|---|---|---|---|---|---|---|---|---|---|---|---|---|---|---|---|---|---|---|---|---|
| pred lang | cfg | *si* | ar | en | ja | zh | *so* | ar | en | ja | zh | *sr* | ar | en | ja | zh | *sw* | ar | en | ja | zh |
| mmT5 | s1 | 0.00 | 0.30 | 0.22 | 0.11 | 0.26 | 0.01 | 0.22 | 0.45 | 0.03 | 0.04 | 0.00 | 0.110 | 0.590 | 0.035 | 0.185 | 0.01 | 0.18 | 0.51 | 0.02 | 0.06 |
| mmT5 | s7 | **1.00** | 0.00 | 0.00 | 0.00 | 0.00 | **1.00** | 0.00 | 0.00 | 0.00 | 0.00 | **0.95** | 0.000 | 0.010 | 0.000 | 0.000 | **1.00** | 0.00 | 0.00 | 0.00 | 0.00 |
| mT5$^S$ | s1 | 0.02 | 0.48 | 0.04 | 0.02 | 0.02 | 0.04 | 0.65 | 0.25 | 0.00 | 0.00 | 0.01 | 0.375 | 0.525 | 0.005 | 0.005 | 0.01 | 0.84 | 0.13 | 0.00 | 0.00 |
| mT5$^S$ | s7 | 0.07 | 0.63 | 0.10 | 0.11 | 0.00 | 0.06 | 0.71 | 0.15 | 0.00 | 0.00 | 0.02 | 0.260 | 0.545 | 0.000 | 0.000 | 0.02 | 0.87 | 0.09 | 0.00 | 0.00 |

| tgt lang |  | ta | | | |
|---|---|---|---|---|---|
| pred lang | cfg | *ta* | ar | en | ja | zh |
| mmT5 s1 | | 0.01 | 0.24 | 0.16 | 0.14 | 0.33 |
| mmT5 s7 | | **1.00** | 0.00 | 0.00 | 0.00 | 0.00 |
| mT5$^S$ s1 | | 0.07 | 0.09 | 0.02 | 0.68 | 0.07 |
| mT5$^S$ s7 | | 0.32 | 0.05 | 0.06 | 0.46 | 0.03 |

**Group 7**

| tgt lang |  | te | | | | th | | | | tr | | | | uk | | | | ur | | | |
|---|---|---|---|---|---|---|---|---|---|---|---|---|---|---|---|---|---|---|---|---|---|
| pred lang | cfg | *te* | ar | en | ja | zh | *th* | ar | en | ja | zh | *tr* | ar | en | ja | zh | *uk* | ar | en | ja | zh |
| mmT5 | s1 | 0.01 | 0.35 | 0.14 | 0.10 | 0.32 | 0.00 | 0.13 | 0.04 | 0.01 | 0.79 | 0.01 | 0.04 | 0.49 | 0.13 | 0.18 | 0.00 | 0.24 | 0.32 | 0.04 | 0.32 |
| mmT5 | s7 | **1.00** | 0.00 | 0.00 | 0.00 | 0.00 | **1.00** | 0.00 | 0.00 | 0.00 | 0.00 | **1.00** | 0.00 | 0.00 | 0.00 | 0.00 | **1.00** | 0.00 | 0.00 | 0.00 | 0.00 |
| mT5$^S$ | s1 | 0.09 | 0.30 | 0.02 | 0.49 | 0.09 | 0.08 | 0.18 | 0.01 | 0.00 | 0.68 | 0.08 | 0.01 | 0.44 | 0.39 | 0.03 | 0.02 | 0.79 | 0.12 | 0.00 | 0.03 |
| mT5$^S$ | s7 | 0.31 | 0.28 | 0.22 | 0.22 | 0.05 | 0.29 | 0.00 | 0.00 | 0.04 | 0.60 | 0.37 | 0.00 | 0.27 | 0.24 | 0.00 | 0.18 | 0.41 | 0.31 | 0.00 | 0.00 |

| tgt lang |  | ur | | | |
|---|---|---|---|---|---|
| pred lang | cfg | *ur* | ar | en | ja | zh |
| mmT5 s1 | | 0.01 | 0.88 | 0.04 | 0.00 | 0.00 |
| mmT5 s7 | | **1.00** | 0.00 | 0.00 | 0.00 | 0.00 |
| mT5$^S$ s1 | | 0.02 | 0.56 | 0.01 | 0.20 | 0.14 |
| mT5$^S$ s7 | | 0.20 | 0.20 | 0.06 | 0.24 | 0.03 |

**Group 8**

| tgt lang |  | uz | | | | vi | | | | yo | | | | zh | | | | avg (excluding source langs) | | | |
|---|---|---|---|---|---|---|---|---|---|---|---|---|---|---|---|---|---|---|---|---|---|
| pred lang | cfg | *uz* | ar | en | ja | zh | *vi* | ar | en | ja | zh | *yo* | ar | en | ja | zh | *zh* | ar | en | ja | zh | *tgt* | ar | en | ja | zh |
| mmT5 | s1 | 0.01 | 0.14 | 0.15 | 0.14 | 0.31 | 0.02 | 0.26 | 0.28 | 0.03 | 0.31 | 0.02 | 0.25 | 0.29 | 0.02 | 0.17 | **1.0** | 0.0 | 0.0 | 0.0 | 1.0 | 0.02 | 0.25 | 0.32 | 0.07 | 0.24 |
| mmT5 | s7 | **0.99** | 0.00 | 0.00 | 0.00 | 0.00 | **1.00** | 0.00 | 0.00 | 0.00 | 0.00 | **1.00** | 0.00 | 0.00 | 0.00 | 0.00 | **1.0** | 0.0 | 0.0 | 0.0 | 1.0 | **0.99** | 0.00 | 0.00 | 0.00 | 0.00 |
| mT5$^S$ | s1 | 0.00 | 0.36 | 0.01 | 0.45 | 0.11 | 0.08 | 0.24 | 0.49 | 0.00 | 0.16 | 0.22 | 0.07 | 0.66 | 0.00 | 0.01 | **1.0** | 0.0 | 0.0 | 0.0 | 1.0 | 0.07 | 0.29 | 0.29 | 0.24 | 0.07 |
| mT5$^S$ | s7 | 0.02 | 0.08 | 0.19 | 0.46 | 0.03 | 0.31 | 0.04 | 0.61 | 0.00 | 0.00 | 0.44 | 0.00 | 0.50 | 0.00 | 0.00 | **1.0** | 0.0 | 0.0 | 0.0 | 1.0 | 0.18 | 0.19 | 0.31 | 0.19 | 0.03 |

Table 10: Language prediction results on the XL-Sum$^{ar,en,ja,zh}$ task setup. The generated summarization text is passed into the language prediction model. We report the percentage of text which the model predicts to be in the correct *target* language, as well as each of the 4 source languages. It is possible that another language was predicted, the numbers therefore do not need to sum up to 1.0.

| | cfg | ar | bg | de | el | en | es | fr | hi | ru | sw | th | tr | ur | vi | zh | *avg* |
|---|---|---|---|---|---|---|---|---|---|---|---|---|---|---|---|---|---|
| Small / mmT5 | s1 | 63.7 | 68.4 | 68.4 | 66.7 | 77.6 | 71.6 | 69.3 | 60.8 | 66.3 | 60.2 | 62.2 | 62.5 | 56.2 | 64.5 | 63.4 | 65.4 |
| | s6 | 63.5 | 69.2 | 69.2 | 68.2 | **81.7** | 73.9 | 71.7 | 62.2 | 67.1 | 60.8 | 64.8 | 62.3 | 54.7 | 61.3 | 64.2 | 66.3 |
| | s7 | **65.4** | **70.8** | **70.5** | **70.1** | 81.5 | **74.5** | **73.0** | **63.1** | **69.1** | **62.7** | **66.1** | **63.1** | **58.4** | **65.7** | 65.7 | **68.0** |
| | s10 | 63.9 | 69.6 | 68.9 | 69.4 | 80.6 | 73.7 | 70.4 | 62.4 | 67.8 | 59.9 | 65.2 | 61.4 | 54.6 | 62.6 | 65.6 | 66.4 |
| | s14 | 64.7 | 70.7 | 70.4 | 69.5 | 81.6 | 74.5 | 72.6 | 63.2 | 68.8 | 60.9 | 65.3 | 62.4 | 58.0 | 65.1 | **65.9** | 67.6 |
| Small / mT5$^S$ | s1 | 63.2 | 68.6 | 67.8 | 67.3 | 80.2 | 70.8 | 70.1 | 59.8 | 67.6 | 52.9 | 62.5 | 58.2 | 56.9 | 63.5 | 63.2 | 64.8 |
| | s6 | 58.8 | 64.5 | 64.2 | 62.4 | 77.2 | 67.3 | 65.9 | 55.0 | 64.0 | 48.2 | 60.3 | 53.1 | 50.9 | 58.6 | 61.2 | 60.8 |
| | s7 | 62.2 | 66.9 | 66.6 | 65.6 | 77.9 | 70.8 | 69.3 | 59.4 | 67.1 | 53.2 | 60.6 | 57.6 | 56.1 | 61.7 | 63.5 | 63.9 |
| | s10 | 57.5 | 64.2 | 66.8 | 63.6 | 77.6 | 68.2 | 66.3 | 56.4 | 64.4 | 51.0 | 59.9 | 54.8 | 51.8 | 59.0 | 61.3 | 61.5 |
| | s14 | 61.8 | 66.9 | 67.7 | 66.3 | 77.6 | 71.0 | 69.3 | 59.4 | 67.6 | 53.3 | 61.6 | 58.0 | 56.3 | 61.1 | 62.9 | 64.1 |
| Base / mmT5 | s1 | 73.5 | 77.9 | 77.6 | 77.7 | 84.3 | 79.2 | 77.8 | 72.2 | 75.3 | 71.7 | 72.9 | 73.5 | 68.9 | 75.0 | 74.6 | 75.5 |
| | s2 | 73.3 | 77.1 | 77.2 | 77.1 | 84.1 | 79.8 | 76.9 | 71.2 | 75.2 | 70.7 | 72.8 | 72.6 | 68.7 | 74.2 | 74.9 | 75.0 |
| | s3 | 74.4 | 79.9 | 79.7 | 78.2 | 85.8 | 81.3 | 79.5 | 73.6 | 76.7 | 72.9 | 75.0 | 74.4 | 69.2 | 76.2 | 77.0 | 76.9 |
| | s4 | 75.9 | 79.8 | 79.5 | **79.6** | 86.2 | 81.9 | **80.1** | 73.5 | **78.0** | 73.5 | 74.3 | 75.2 | **70.1** | **77.0** | 77.4 | 77.5 |
| | s5 | 72.4 | 75.5 | 75.4 | 76.1 | 83.3 | 79.2 | 76.8 | 70.9 | 73.4 | 71.5 | 69.9 | 70.7 | 67.1 | 71.8 | 72.8 | 73.8 |
| | s6 | 73.5 | 77.6 | 78.2 | 77.2 | 84.6 | 81.0 | 79.0 | 72.3 | 76.2 | 71.8 | 72.4 | 72.2 | 68.7 | 73.8 | 74.7 | 75.6 |
| | s7 | 75.3 | 80.1 | 79.8 | 79.1 | 86.3 | 82.2 | 79.4 | **73.5** | 77.8 | 73.0 | 75.1 | 74.7 | 70.0 | 75.8 | 77.5 | 77.3 |
| | s8 | 72.2 | 76.4 | 73.7 | 74.8 | 83.2 | 77.8 | 73.2 | 70.2 | 73.5 | 69.7 | 71.7 | 70.7 | 67.1 | 71.7 | 73.5 | 73.3 |
| | s9 | 74.1 | 77.7 | 77.8 | 76.9 | 84.7 | 80.2 | 78.3 | 71.6 | 75.8 | 71.4 | 73.2 | 73.7 | 69.6 | 74.0 | 75.3 | 75.6 |
| | s10 | 74.1 | 77.7 | 78.0 | 77.5 | 84.3 | 81.4 | 78.3 | 72.9 | 75.9 | 73.2 | 73.5 | 73.7 | 69.4 | 75.1 | 76.2 | 76.1 |
| | s11 | 71.3 | 75.5 | 73.9 | 74.7 | 82.2 | 77.8 | 75.2 | 70.0 | 73.3 | 69.6 | 71.1 | 70.0 | 66.9 | 70.8 | 73.4 | 73.1 |
| | s12 | 74.1 | 78.0 | 77.5 | 77.4 | 84.6 | 79.4 | 78.0 | 71.3 | 75.7 | 71.8 | 73.4 | 73.5 | 69.3 | 74.7 | 75.7 | 75.6 |
| | s13 | 75.2 | 78.1 | 78.3 | 78.2 | 84.4 | 80.1 | 78.4 | 73.1 | 75.9 | 73.0 | 73.9 | 73.8 | 69.7 | 75.8 | 76.5 | 76.3 |
| | s14 | **76.0** | **80.6** | **81.0** | 79.0 | **86.9** | **82.4** | 79.9 | 73.2 | 77.6 | **74.0** | **74.9** | **75.7** | 70.0 | 76.3 | **77.6** | **77.7** |
| Base / mT5$^S$ | s1 | 72.0 | 76.0 | 76.4 | 76.3 | 84.4 | 78.0 | 78.2 | 69.8 | 74.7 | 66.1 | 71.3 | 71.1 | 67.8 | 73.0 | 72.4 | 73.8 |
| | s6 | 71.2 | 75.0 | 75.0 | 75.7 | 84.3 | 78.7 | 78.0 | 68.3 | 74.0 | 61.6 | 71.0 | 69.6 | 64.9 | 70.9 | 71.9 | 72.7 |
| | s7 | 72.5 | 77.6 | 77.5 | 76.6 | 85.0 | 79.2 | 79.5 | 70.6 | 75.2 | 65.5 | 72.6 | 72.1 | 68.2 | 73.5 | 73.6 | 74.6 |
| | s10 | 66.7 | 70.8 | 71.3 | 69.9 | 78.9 | 73.3 | 71.8 | 64.5 | 69.2 | 60.8 | 66.9 | 64.9 | 61.8 | 66.1 | 68.7 | 68.4 |
| | s14 | 69.2 | 73.9 | 73.7 | 74.0 | 81.9 | 75.6 | 75.3 | 67.2 | 72.5 | 63.2 | 68.9 | 68.2 | 64.7 | 70.1 | 71.2 | 71.3 |

Table 11: XNLI validation results for all languages. We report the results of different freezing configurations.

| | lang | amharic
Rg$_1$ / Rg$_2$ / Rg$_L$ | arabic
Rg$_1$ / Rg$_2$ / Rg$_L$ | azerbaijani
Rg$_1$ / Rg$_2$ / Rg$_L$ | bengali
Rg$_1$ / Rg$_2$ / Rg$_L$ | burmese
Rg$_1$ / Rg$_2$ / Rg$_L$ | chinese_simplified
Rg$_1$ / Rg$_2$ / Rg$_L$ | chinese_traditional
Rg$_1$ / Rg$_2$ / Rg$_L$ |
|---|---|---|---|---|---|---|---|---|
| small | mmT5 | 29.7 / 13.4 / 22.7 | 37.5 / 17.3 / 29.4 | 18.5 / 6.8 / 15.5 | 27.6 / 13.1 / 22.5 | 38.7 / 18.2 / 30.0 | 38.7 / 19.4 / 33.3 | 39.3 / 20.9 / 33.6 |
| | mT5$^S$ | 13.2 / 3.4 / 12.2 | 37.5 / 17.4 / 29.4 | 13.3 / 2.5 / 12.1 | 7.6 / 1.1 / 7.4 | 12.7 / 2.0 / 12.1 | 38.2 / 19.1 / 33.1 | 39.1 / 20.9 / 33.4 |
| base | mmT5 | 34.2 / 17.1 / 25.8 | 42.3 / 22.2 / 33.4 | 26.4 / 11.1 / 21.3 | 33.8 / 17.0 / 26.1 | 40.0 / 18.9 / 30.7 | 48.1 / 29.1 / 42.9 | 49.0 / 30.8 / 43.1 |
| | mT5$^S$ | 15.0 / 4.5 / 13.7 | 40.2 / 20.1 / 31.5 | 16.0 / 4.4 / 14.2 | 11.9 / 3.4 / 11.2 | 15.7 / 3.7 / 14.5 | 44.2 / 26.1 / 39.2 | 45.2 / 27.7 / 39.4 |

| | lang | english
Rg$_1$ / Rg$_2$ / Rg$_L$ | french
Rg$_1$ / Rg$_2$ / Rg$_L$ | gujarati
Rg$_1$ / Rg$_2$ / Rg$_L$ | hausa
Rg$_1$ / Rg$_2$ / Rg$_L$ | hindi
Rg$_1$ / Rg$_2$ / Rg$_L$ | igbo
Rg$_1$ / Rg$_2$ / Rg$_L$ | indonesian
Rg$_1$ / Rg$_2$ / Rg$_L$ |
|---|---|---|---|---|---|---|---|---|
| small | mmT5 | 38.9 / 14.4 / 30.4 | 34.7 / 13.7 / 26.7 | 26.8 / 12.2 / 21.5 | 28.7 / 12.1 / 23.1 | 30.6 / 13.8 / 24.5 | 37.1 / 16.6 / 28.1 | 31.4 / 11.9 / 25.2 |
| | mT5$^S$ | 39.0 / 14.7 / 30.4 | 22.1 / 4.4 / 19.4 | 8.8 / 1.5 / 8.6 | 16.4 / 5.0 / 14.9 | 12.8 / 3.2 / 12.1 | 22.0 / 6.1 / 19.1 | 17.1 / 3.5 / 15.8 |
| base | mmT5 | 43.6 / 19.2 / 34.5 | 37.0 / 15.2 / 27.8 | 32.0 / 15.8 / 24.6 | 36.7 / 16.5 / 27.7 | 34.9 / 16.9 / 27.2 | 40.0 / 19.1 / 29.6 | 36.1 / 15.2 / 27.9 |
| | mT5$^S$ | 42.6 / 18.3 / 33.5 | 23.4 / 5.3 / 20.1 | 11.7 / 3.7 / 10.8 | 20.6 / 7.1 / 18.2 | 15.0 / 4.9 / 13.8 | 30.3 / 10.5 / 24.3 | 20.8 / 5.5 / 18.3 |

| | lang | japanese
Rg$_1$ / Rg$_2$ / Rg$_L$ | korean
Rg$_1$ / Rg$_2$ / Rg$_L$ | kyrgyz
Rg$_1$ / Rg$_2$ / Rg$_L$ | marathi
Rg$_1$ / Rg$_2$ / Rg$_L$ | nepali
Rg$_1$ / Rg$_2$ / Rg$_L$ | pashto
Rg$_1$ / Rg$_2$ / Rg$_L$ | persian
Rg$_1$ / Rg$_2$ / Rg$_L$ |
|---|---|---|---|---|---|---|---|---|
| small | mmT5 | 44.9 / 25.6 / 37.0 | 31.3 / 13.3 / 25.8 | 20.0 / 7.1 / 17.0 | 19.4 / 8.2 / 15.9 | 29.6 / 13.8 / 24.2 | 37.8 / 16.9 / 28.6 | 37.7 / 17.0 / 29.0 |
| | mT5$^S$ | 41.0 / 21.7 / 33.1 | 8.3 / 2.3 / 7.7 | 10.3 / 1.6 / 9.7 | 11.7 / 3.6 / 10.7 | 11.7 / 3.6 / 10.7 | 18.1 / 3.4 / 16.2 | 21.4 / 5.5 / 18.7 |
| base | mmT5 | 48.2 / 28.3 / 39.5 | 34.4 / 16.1 / 27.9 | 26.0 / 10.7 / 21.2 | 25.5 / 11.4 / 20.3 | 34.6 / 17.0 / 26.9 | 42.1 / 19.8 / 30.7 | 41.0 / 19.6 / 30.6 |
| | mT5$^S$ | 44.3 / 24.8 / 36.0 | 10.9 / 4.3 / 9.9 | 12.3 / 2.4 / 11.5 | 13.4 / 5.1 / 12.0 | 12.0 / 3.1 / 11.1 | 21.1 / 4.8 / 18.2 | 23.6 / 6.4 / 19.9 |

| | lang | portuguese
Rg$_1$ / Rg$_2$ / Rg$_L$ | punjabi
Rg$_1$ / Rg$_2$ / Rg$_L$ | russian
Rg$_1$ / Rg$_2$ / Rg$_L$ | scottish_gaelic
Rg$_1$ / Rg$_2$ / Rg$_L$ | serbian_cyrillic
Rg$_1$ / Rg$_2$ / Rg$_L$ | serbian_latin
Rg$_1$ / Rg$_2$ / Rg$_L$ | sinhala
Rg$_1$ / Rg$_2$ / Rg$_L$ |
|---|---|---|---|---|---|---|---|---|
| small | mmT5 | 33.4 / 12.2 / 26.2 | 32.1 / 14.0 / 24.2 | 25.4 / 8.3 / 20.4 | 29.3 / 11.5 / 24.3 | 21.4 / 6.5 / 17.9 | 10.8 / 3.1 / 9.5 | 28.6 / 14.2 / 21.8 |
| | mT5$^S$ | 23.1 / 5.2 / 20.5 | 10.8 / 1.3 / 10.5 | 12.9 / 1.4 / 12.5 | 19.4 / 5.9 / 17.3 | 11.1 / 0.9 / 11.0 | 11.5 / 2.3 / 10.4 | 9.2 / 1.1 / 9.0 |
| base | mmT5 | 38.3 / 15.5 / 28.2 | 37.6 / 19.1 / 27.5 | 30.8 / 11.3 / 23.7 | 30.6 / 13.7 / 24.3 | 29.2 / 10.0 / 22.6 | 19.0 / 5.4 / 16.1 | 34.0 / 18.8 / 25.5 |
| | mT5$^S$ | 25.5 / 6.5 / 21.5 | 13.7 / 3.0 / 12.8 | 16.2 / 3.0 / 14.9 | 23.2 / 7.2 / 20.1 | 13.8 / 1.9 / 13.3 | 18.8 / 4.2 / 16.6 | 10.0 / 2.0 / 9.4 |

| | lang | somali
Rg$_1$ / Rg$_2$ / Rg$_L$ | spanish
Rg$_1$ / Rg$_2$ / Rg$_L$ | swahili
Rg$_1$ / Rg$_2$ / Rg$_L$ | tamil
Rg$_1$ / Rg$_2$ / Rg$_L$ | telugu
Rg$_1$ / Rg$_2$ / Rg$_L$ | thai
Rg$_1$ / Rg$_2$ / Rg$_L$ | turkish
Rg$_1$ / Rg$_2$ / Rg$_L$ |
|---|---|---|---|---|---|---|---|---|
| small | mmT5 | 28.8 / 11.3 / 22.5 | 28.5 / 8.3 / 22.3 | 30.8 / 11.5 / 23.4 | 25.0 / 11.8 / 20.4 | 23.2 / 10.6 / 19.0 | 26.6 / 12.9 / 20.4 | 24.9 / 10.0 / 20.6 |
| | mT5$^S$ | 16.7 / 2.6 / 15.1 | 18.7 / 3.6 / 16.8 | 14.5 / 2.7 / 13.4 | 7.9 / 1.8 / 7.5 | 8.4 / 1.3 / 8.2 | 7.1 / 2.1 / 6.6 | 16.7 / 4.1 / 14.8 |
| base | mmT5 | 33.6 / 14.4 / 25.1 | 32.7 / 10.7 / 24.1 | 36.1 / 15.3 / 26.7 | 29.0 / 14.4 / 23.0 | 26.5 / 12.9 / 21.1 | 30.3 / 15.3 / 22.9 | 31.3 / 14.0 / 25.3 |
| | mT5$^S$ | 18.0 / 3.6 / 16.2 | 21.0 / 4.6 / 18.2 | 16.1 / 3.6 / 14.7 | 10.5 / 3.6 / 9.6 | 10.8 / 2.9 / 10.0 | 10.5 / 3.5 / 9.5 | 18.7 / 5.7 / 16.3 |

| | lang | ukrainian
Rg$_1$ / Rg$_2$ / Rg$_L$ | urdu
Rg$_1$ / Rg$_2$ / Rg$_L$ | uzbek
Rg$_1$ / Rg$_2$ / Rg$_L$ | vietnamese
Rg$_1$ / Rg$_2$ / Rg$_L$ | welsh
Rg$_1$ / Rg$_2$ / Rg$_L$ | yoruba
Rg$_1$ / Rg$_2$ / Rg$_L$ | *avg*
Rg$_1$ / Rg$_2$ / Rg$_L$ |
|---|---|---|---|---|---|---|---|---|
| small | mmT5 | 22.1 / 7.1 / 18.1 | 34.4 / 15.5 / 26.1 | 4.4 / 0.9 / 4.2 | 36.4 / 16.9 / 25.9 | 27.6 / 10.8 / 21.6 | 29.9 / 12.9 / 22.9 | 29.4 / 12.6 / 23.3 |
| | mT5$^S$ | 11.4 / 1.2 / 11.1 | 13.5 / 3.0 / 12.5 | 9.5 / 1.5 / 9.1 | 16.7 / 3.2 / 15.3 | 18.2 / 3.4 / 16.7 | 18.7 / 3.9 / 16.8 | 17.0 / 4.7 / 15.1 |
| base | mmT5 | 28.6 / 10.5 / 22.4 | 38.1 / 18.4 / 28.4 | 14.7 / 3.0 / 13.7 | 42.2 / 20.9 / 28.8 | 26.9 / 11.3 / 20.7 | 37.7 / 16.4 / 28.0 | **34.5** / **16.1** / **26.8** |
| | mT5$^S$ | 14.0 / 2.4 / 13.1 | 15.5 / 3.6 / 14.2 | 11.9 / 2.3 / 11.3 | 23.1 / 6.4 / 19.7 | 22.6 / 4.8 / 19.8 | 24.9 / 6.3 / 21.2 | 18.6 / 6.0 / 16.7 |

Table 12: XL-Sum$^{ar,en,ja,zh}$ test set results for all languages. We evaluate using the best performing model on the four source languages on the validation set.

| | lang | amharic Rg$_1$ / Rg$_2$ / Rg$_L$ | arabic Rg$_1$ / Rg$_2$ / Rg$_L$ | azerbaijani Rg$_1$ / Rg$_2$ / Rg$_L$ | bengali Rg$_1$ / Rg$_2$ / Rg$_L$ | burmese Rg$_1$ / Rg$_2$ / Rg$_L$ | chinese_simplified Rg$_1$ / Rg$_2$ / Rg$_L$ | chinese_traditional Rg$_1$ / Rg$_2$ / Rg$_L$ |
|---|---|---|---|---|---|---|---|---|
| small | mmT5 | 12.3 / 1.7 / 11.7 | 16.9 / 3.3 / 15.4 | 13.7 / 3.6 / 11.8 | 8.5 / 2.1 / 7.7 | 17.6 / 3.7 / 15.8 | 4.0 / 1.2 / 3.7 | 5.3 / 2.0 / 4.9 |
| | mT5$^S$ | 13.6 / 1.2 / 12.9 | 15.2 / 0.6 / 14.9 | 13.9 / 1.7 / 12.7 | 9.8 / 0.4 / 9.6 | 18.3 / 0.6 / 18.0 | 5.5 / 1.2 / 5.3 | 5.9 / 1.7 / 5.7 |
| base | mmT5 | 16.6 / 5.2 / 14.4 | 19.3 / 4.5 / 17.1 | 17.5 / 5.2 / 14.6 | 15.1 / 5.2 / 12.7 | 22.0 / 4.9 / 19.6 | 9.2 / 2.8 / 8.3 | 8.3 / 2.6 / 7.5 |
| | mT5$^S$ | 13.7 / 1.7 / 13.1 | 15.8 / 1.3 / 15.1 | 15.3 / 2.5 / 13.5 | 10.5 / 0.8 / 10.0 | 18.1 / 1.6 / 17.2 | 6.7 / 1.8 / 6.4 | 6.9 / 2.3 / 6.6 |

| | lang | english Rg$_1$ / Rg$_2$ / Rg$_L$ | french Rg$_1$ / Rg$_2$ / Rg$_L$ | gujarati Rg$_1$ / Rg$_2$ / Rg$_L$ | hausa Rg$_1$ / Rg$_2$ / Rg$_L$ | hindi Rg$_1$ / Rg$_2$ / Rg$_L$ | igbo Rg$_1$ / Rg$_2$ / Rg$_L$ | indonesian Rg$_1$ / Rg$_2$ / Rg$_L$ |
|---|---|---|---|---|---|---|---|---|
| small | mmT5 | 41.5 / 17.0 / 32.5 | 27.3 / 8.3 / 22.2 | 11.8 / 2.6 / 10.9 | 19.8 / 6.9 / 17.5 | 13.8 / 3.2 / 12.7 | 28.2 / 9.7 / 22.9 | 22.4 / 6.1 / 19.1 |
| | mT5$^S$ | 42.0 / 17.7 / 33.1 | 21.5 / 3.4 / 18.4 | 10.3 / 0.3 / 10.1 | 18.8 / 4.5 / 16.5 | 13.1 / 0.9 / 12.8 | 25.6 / 5.1 / 22.1 | 18.1 / 2.8 / 16.1 |
| base | mmT5 | 45.8 / 21.7 / 36.5 | 27.7 / 8.3 / 22.0 | 15.9 / 5.6 / 13.7 | 27.8 / 11.0 / 22.3 | 16.9 / 4.6 / 15.0 | 25.6 / 9.0 / 20.8 | 26.2 / 8.2 / 21.5 |
| | mT5$^S$ | 44.6 / 20.5 / 35.3 | 21.5 / 3.8 / 18.0 | 10.9 / 0.6 / 10.6 | 20.2 / 5.2 / 17.5 | 14.1 / 1.7 / 13.4 | 26.0 / 5.4 / 21.9 | 19.2 / 3.5 / 16.8 |

| | lang | japanese Rg$_1$ / Rg$_2$ / Rg$_L$ | korean Rg$_1$ / Rg$_2$ / Rg$_L$ | kyrgyz Rg$_1$ / Rg$_2$ / Rg$_L$ | marathi Rg$_1$ / Rg$_2$ / Rg$_L$ | nepali Rg$_1$ / Rg$_2$ / Rg$_L$ | pashto Rg$_1$ / Rg$_2$ / Rg$_L$ | persian Rg$_1$ / Rg$_2$ / Rg$_L$ |
|---|---|---|---|---|---|---|---|---|
| small | mmT5 | 4.0 / 1.0 / 3.8 | 17.2 / 4.8 / 15.4 | 9.9 / 1.7 / 9.2 | 9.1 / 1.4 / 8.5 | 12.7 / 3.6 / 11.4 | 17.4 / 3.1 / 16.1 | 18.2 / 4.0 / 16.8 |
| | mT5$^S$ | 4.2 / 1.1 / 4.1 | 13.7 / 1.0 / 13.3 | 12.7 / 0.6 / 12.3 | 11.1 / 0.9 / 10.7 | 10.8 / 0.4 / 10.6 | 14.5 / 0.4 / 14.3 | 16.3 / 0.7 / 16.0 |
| base | mmT5 | 6.1 / 2.1 / 5.6 | 19.4 / 5.8 / 16.8 | 13.0 / 2.1 / 11.7 | 11.5 / 2.2 / 10.5 | 13.7 / 3.6 / 12.2 | 18.4 / 4.4 / 16.2 | 23.0 / 6.8 / 19.9 |
| | mT5$^S$ | 4.6 / 1.2 / 4.4 | 15.4 / 2.3 / 14.4 | 12.9 / 0.8 / 12.2 | 12.0 / 1.4 / 11.3 | 11.8 / 1.1 / 11.1 | 16.0 / 1.4 / 15.2 | 17.3 / 1.8 / 16.3 |

| | lang | portuguese Rg$_1$ / Rg$_2$ / Rg$_L$ | punjabi Rg$_1$ / Rg$_2$ / Rg$_L$ | russian Rg$_1$ / Rg$_2$ / Rg$_L$ | scottish_gaelic Rg$_1$ / Rg$_2$ / Rg$_L$ | serbian_cyrillic Rg$_1$ / Rg$_2$ / Rg$_L$ | serbian_latin Rg$_1$ / Rg$_2$ / Rg$_L$ | sinhala Rg$_1$ / Rg$_2$ / Rg$_L$ |
|---|---|---|---|---|---|---|---|---|
| small | mmT5 | 30.4 / 9.9 / 24.7 | 12.5 / 3.3 / 10.9 | 14.3 / 2.5 / 13.0 | 27.8 / 10.0 / 23.3 | 8.3 / 1.5 / 7.7 | 11.1 / 2.2 / 10.2 | 14.1 / 3.6 / 12.9 |
| | mT5$^S$ | 23.5 / 5.0 / 20.2 | 12.1 / 0.3 / 11.9 | 14.6 / 1.1 / 13.8 | 21.7 / 5.2 / 19.0 | 13.3 / 0.7 / 12.9 | 17.1 / 2.2 / 15.3 | 10.8 / 0.4 / 10.6 |
| base | mmT5 | 30.9 / 10.2 / 24.3 | 18.0 / 5.9 / 14.8 | 18.7 / 3.8 / 16.2 | 25.0 / 10.3 / 20.5 | 15.5 / 2.8 / 13.9 | 15.6 / 3.5 / 13.2 | 17.5 / 6.6 / 14.8 |
| | mT5$^S$ | 23.6 / 5.5 / 19.8 | 12.7 / 0.8 / 12.2 | 14.8 / 1.2 / 13.7 | 22.7 / 6.3 / 19.2 | 13.8 / 0.9 / 13.1 | 17.3 / 2.3 / 15.1 | 10.7 / 0.6 / 10.4 |

| | lang | somali Rg$_1$ / Rg$_2$ / Rg$_L$ | spanish Rg$_1$ / Rg$_2$ / Rg$_L$ | swahili Rg$_1$ / Rg$_2$ / Rg$_L$ | tamil Rg$_1$ / Rg$_2$ / Rg$_L$ | telugu Rg$_1$ / Rg$_2$ / Rg$_L$ | thai Rg$_1$ / Rg$_2$ / Rg$_L$ | turkish Rg$_1$ / Rg$_2$ / Rg$_L$ |
|---|---|---|---|---|---|---|---|---|
| small | mmT5 | 20.2 / 5.0 / 17.0 | 21.2 / 5.6 / 17.4 | 22.0 / 5.8 / 18.1 | 13.4 / 4.2 / 12.0 | 12.8 / 3.5 / 11.7 | 13.0 / 4.0 / 11.6 | 15.7 / 4.8 / 13.4 |
| | mT5$^S$ | 18.6 / 2.2 / 16.2 | 19.3 / 3.3 / 16.9 | 19.1 / 3.2 / 16.6 | 9.5 / 0.7 / 9.2 | 10.0 / 0.7 / 9.8 | 9.4 / 1.1 / 8.8 | 16.3 / 2.9 / 14.4 |
| base | mmT5 | 22.0 / 6.4 / 17.3 | 25.5 / 6.9 / 20.0 | 25.6 / 8.0 / 20.1 | 17.3 / 6.4 / 14.4 | 15.2 / 5.2 / 13.0 | 14.3 / 4.9 / 12.5 | 21.6 / 7.1 / 17.5 |
| | mT5$^S$ | 19.6 / 2.6 / 16.7 | 19.9 / 3.8 / 17.0 | 20.1 / 4.2 / 17.3 | 9.8 / 1.0 / 9.3 | 10.3 / 1.0 / 9.9 | 10.0 / 1.3 / 9.2 | 17.6 / 3.7 / 15.2 |

| | lang | ukrainian Rg$_1$ / Rg$_2$ / Rg$_L$ | urdu Rg$_1$ / Rg$_2$ / Rg$_L$ | uzbek Rg$_1$ / Rg$_2$ / Rg$_L$ | vietnamese Rg$_1$ / Rg$_2$ / Rg$_L$ | welsh Rg$_1$ / Rg$_2$ / Rg$_L$ | yoruba Rg$_1$ / Rg$_2$ / Rg$_L$ | _avg_ Rg$_1$ / Rg$_2$ / Rg$_L$ |
|---|---|---|---|---|---|---|---|---|
| small | mmT5 | 10.8 / 1.7 / 10.1 | 16.6 / 3.8 / 15.0 | 8.6 / 0.6 / 8.4 | 29.8 / 11.4 / 22.1 | 20.5 / 5.4 / 17.5 | 28.8 / 10.0 / 23.3 | 16.7 / 4.6 / 14.4 |
| | mT5$^S$ | 13.5 / 0.9 / 12.9 | 13.1 / 0.3 / 12.9 | 12.5 / 0.6 / 12.1 | 21.2 / 3.6 / 18.1 | 22.2 / 3.5 / 19.9 | 23.7 / 3.9 / 20.8 | 15.5 / 2.2 / 14.2 |
| base | mmT5 | 16.9 / 3.6 / 14.9 | 17.6 / 4.7 / 15.3 | 8.5 / 0.7 / 8.2 | 32.9 / 13.8 / 23.6 | 22.8 / 6.4 / 18.9 | 24.8 / 8.5 / 19.8 | **19.6 / 6.1 / 16.4** |
| | mT5$^S$ | 14.1 / 1.2 / 13.1 | 14.0 / 1.3 / 13.4 | 13.0 / 0.8 / 12.4 | 22.1 / 4.8 / 18.5 | 23.0 / 4.4 / 19.6 | 23.2 / 3.9 / 19.9 | 16.2 / 2.8 / 4.5 |

Table 13: XL-Sum$^{en}$ test set results for all languages. We evaluate using the best performing model on the English language of the validation set.

| | | | amharic Rg$_1$/Rg$_2$/Rg$_L$ | arabic Rg$_1$/Rg$_2$/Rg$_L$ | azerbaijani Rg$_1$/Rg$_2$/Rg$_L$ | bengali Rg$_1$/Rg$_2$/Rg$_L$ | burmese Rg$_1$/Rg$_2$/Rg$_L$ | chinese_simplified Rg$_1$/Rg$_2$/Rg$_L$ | chinese_traditional Rg$_1$/Rg$_2$/Rg$_L$ |
|---|---|---|---|---|---|---|---|---|---|
| Small | mmT5 | s7 | 12.5 / 1.8 / 11.9 | 17.2 / 3.3 / 15.7 | 14.0 / 4.0 / 12.1 | 8.7 / 2.2 / 7.9 | 17.8 / 3.6 / 16.2 | 3.9 / 1.2 / 3.6 | 5.4 / 2.0 / 4.9 |
| Small | mT5$^S$ | s1 | 13.6 / 1.2 / 13.0 | 15.3 / 0.6 / 15.0 | 14.0 / 1.9 / 12.7 | 9.5 / 0.3 / 9.4 | 18.1 / 0.5 / 17.8 | 5.6 / 1.2 / 5.4 | 5.8 / 1.6 / 5.6 |
| Base | mmT5 | s1 | 12.6 / 0.2 / 12.4 | 15.0 / 0.3 / 14.7 | 13.9 / 1.4 / 12.5 | 9.5 / 0.2 / 9.3 | 17.8 / 0.2 / 17.6 | 5.3 / 0.5 / 5.1 | 5.0 / 0.5 / 4.9 |
| Base | mmT5 | s7 | 16.3 / 5.0 / 14.2 | 19.8 / 4.7 / 17.4 | 17.7 / 5.4 / 14.7 | 15.0 / 5.0 / 12.5 | 22.0 / 4.8 / 19.5 | 9.1 / 2.7 / 8.1 | 8.1 / 2.5 / 7.3 |
| Base | mT5$^S$ | s1 | 13.8 / 1.6 / 13.0 | 16.0 / 1.3 / 15.3 | 15.4 / 2.9 / 13.6 | 9.9 / 0.6 / 9.5 | 17.8 / 1.7 / 16.9 | 6.6 / 1.8 / 6.3 | 6.8 / 2.2 / 6.5 |
| Base | mT5$^S$ | s7 | 13.7 / 1.4 / 13.1 | 16.7 / 1.3 / 16.0 | 16.7 / 3.6 / 14.7 | 9.9 / 0.5 / 9.6 | 18.2 / 1.4 / 17.4 | 6.4 / 1.6 / 6.2 | 6.0 / 1.6 / 5.8 |

| | | | english Rg$_1$/Rg$_2$/Rg$_L$ | french Rg$_1$/Rg$_2$/Rg$_L$ | gujarati Rg$_1$/Rg$_2$/Rg$_L$ | hausa Rg$_1$/Rg$_2$/Rg$_L$ | hindi Rg$_1$/Rg$_2$/Rg$_L$ | igbo Rg$_1$/Rg$_2$/Rg$_L$ | indonesian Rg$_1$/Rg$_2$/Rg$_L$ |
|---|---|---|---|---|---|---|---|---|---|
| Small | mmT5 | s7 | 41.6 / 17.0 / 32.6 | 27.8 / 8.7 / 22.5 | 12.0 / 2.9 / 11.1 | 19.3 / 6.6 / 17.2 | 13.6 / 3.0 / 12.6 | 28.1 / 9.7 / 22.9 | 22.5 / 6.0 / 19.1 |
| Small | mT5$^S$ | s1 | 42.3 / 17.9 / 33.2 | 22.1 / 3.6 / 18.9 | 10.6 / 0.4 / 10.4 | 18.3 / 4.3 / 16.3 | 13.1 / 0.9 / 12.8 | 25.4 / 5.0 / 21.8 | 18.1 / 2.8 / 16.2 |
| Base | mmT5 | s1 | 45.0 / 20.8 / 35.6 | 21.1 / 3.6 / 17.8 | 10.6 / 0.2 / 10.4 | 20.1 / 5.5 / 17.7 | 12.4 / 0.2 / 12.1 | 24.5 / 4.0 / 21.4 | 18.4 / 2.9 / 16.2 |
| Base | mmT5 | s7 | 46.1 / 21.8 / 36.6 | 27.9 / 8.4 / 22.2 | 15.9 / 5.6 / 13.6 | 27.5 / 11.1 / 22.0 | 16.6 / 4.4 / 14.7 | 25.4 / 9.3 / 20.6 | 26.1 / 8.0 / 21.4 |
| Base | mT5$^S$ | s1 | 44.7 / 20.6 / 35.4 | 21.4 / 3.9 / 18.2 | 11.1 / 0.8 / 10.7 | 19.5 / 4.9 / 16.9 | 14.0 / 1.7 / 13.3 | 26.2 / 5.4 / 22.0 | 19.2 / 3.5 / 16.8 |
| Base | mT5$^S$ | s7 | 43.3 / 18.7 / 34.1 | 22.7 / 4.7 / 19.3 | 11.3 / 0.7 / 11.0 | 21.7 / 6.3 / 18.7 | 14.2 / 1.5 / 13.6 | 27.1 / 6.5 / 22.8 | 20.9 / 4.3 / 18.2 |

| | | | japanese Rg$_1$/Rg$_2$/Rg$_L$ | korean Rg$_1$/Rg$_2$/Rg$_L$ | kyrgyz Rg$_1$/Rg$_2$/Rg$_L$ | marathi Rg$_1$/Rg$_2$/Rg$_L$ | nepali Rg$_1$/Rg$_2$/Rg$_L$ | pashto Rg$_1$/Rg$_2$/Rg$_L$ | persian Rg$_1$/Rg$_2$/Rg$_L$ |
|---|---|---|---|---|---|---|---|---|---|
| Small | mmT5 | s7 | 3.7 / 0.8 / 3.4 | 16.7 / 4.5 / 14.8 | 10.3 / 1.6 / 9.6 | 8.9 / 1.6 / 8.4 | 13.0 / 3.8 / 11.6 | 17.7 / 3.2 / 16.1 | 18.2 / 4.0 / 16.1 |
| Small | mT5$^S$ | s1 | 4.0 / 1.0 / 3.9 | 13.6 / 0.8 / 13.3 | 12.5 / 0.5 / 12.1 | 11.2 / 1.0 / 10.8 | 10.7 / 0.5 / 10.5 | 14.3 / 0.3 / 14.1 | 16.4 / 0.7 / 16.1 |
| Base | mmT5 | s1 | 3.3 / 0.4 / 3.2 | 13.2 / 0.4 / 12.9 | 12.2 / 0.3 / 11.8 | 10.9 / 0.3 / 10.6 | 10.5 / 0.1 / 10.4 | 14.5 / 0.1 / 14.4 | 15.9 / 0.2 / 15.7 |
| Base | mmT5 | s7 | 6.3 / 2.3 / 5.8 | 19.7 / 5.7 / 16.9 | 13.3 / 2.2 / 12.0 | 11.5 / 2.2 / 10.6 | 14.1 / 3.8 / 12.4 | 18.0 / 4.4 / 15.8 | 23.0 / 6.8 / 19.8 |
| Base | mT5$^S$ | s1 | 4.5 / 1.1 / 4.4 | 15.4 / 2.5 / 14.4 | 12.8 / 0.8 / 12.2 | 12.0 / 1.3 / 11.3 | 11.5 / 1.1 / 11.0 | 15.9 / 1.5 / 15.1 | 17.4 / 1.8 / 16.4 |
| Base | mT5$^S$ | s7 | 4.2 / 0.9 / 4.1 | 15.2 / 1.8 / 14.2 | 13.5 / 0.9 / 12.8 | 11.9 / 1.0 / 11.4 | 11.9 / 1.1 / 11.4 | 16.0 / 1.1 / 15.3 | 17.8 / 1.6 / 17.0 |

| | | | portuguese Rg$_1$/Rg$_2$/Rg$_L$ | punjabi Rg$_1$/Rg$_2$/Rg$_L$ | russian Rg$_1$/Rg$_2$/Rg$_L$ | scottish_gaelic Rg$_1$/Rg$_2$/Rg$_L$ | serbian_cyrillic Rg$_1$/Rg$_2$/Rg$_L$ | serbian_latin Rg$_1$/Rg$_2$/Rg$_L$ | sinhala Rg$_1$/Rg$_2$/Rg$_L$ |
|---|---|---|---|---|---|---|---|---|---|
| Small | mmT5 | s7 | 30.7 / 10.1 / 24.9 | 12.1 / 3.1 / 10.6 | 14.3 / 2.5 / 13.0 | 27.6 / 10.7 / 23.3 | 8.3 / 1.4 / 7.7 | 10.9 / 2.2 / 9.8 | 13.9 / 3.3 / 12.8 |
| Small | mT5$^S$ | s1 | 23.7 / 5.1 / 20.3 | 12.1 / 0.3 / 11.9 | 14.6 / 1.1 / 13.8 | 21.4 / 5.1 / 18.6 | 13.5 / 0.6 / 13.0 | 17.2 / 2.2 / 15.3 | 10.5 / 0.3 / 10.3 |
| Base | mmT5 | s1 | 24.6 / 5.5 / 20.6 | 12.2 / 0.2 / 12.0 | 13.9 / 0.6 / 13.1 | 21.6 / 5.6 / 18.5 | 13.0 / 0.4 / 12.7 | 16.2 / 1.6 / 14.5 | 10.3 / 0.1 / 10.2 |
| Base | mmT5 | s7 | 31.3 / 10.5 / 24.5 | 17.8 / 5.8 / 14.6 | 18.9 / 3.9 / 16.3 | 24.8 / 10.5 / 20.2 | 15.6 / 2.8 / 13.8 | 15.8 / 3.5 / 13.6 | 17.5 / 6.7 / 14.8 |
| Base | mT5$^S$ | s1 | 23.7 / 5.6 / 19.8 | 12.5 / 0.7 / 12.0 | 14.8 / 1.3 / 13.7 | 22.3 / 6.3 / 18.8 | 13.9 / 0.9 / 13.2 | 17.3 / 2.3 / 15.2 | 10.6 / 0.5 / 10.3 |
| Base | mT5$^S$ | s7 | 24.9 / 6.5 / 20.8 | 12.1 / 0.4 / 11.8 | 15.6 / 1.6 / 14.4 | 24.6 / 8.3 / 20.3 | 14.2 / 0.9 / 13.5 | 18.7 / 2.8 / 16.4 | 11.0 / 0.6 / 10.8 |

| | | | somali Rg$_1$/Rg$_2$/Rg$_L$ | spanish Rg$_1$/Rg$_2$/Rg$_L$ | swahili Rg$_1$/Rg$_2$/Rg$_L$ | tamil Rg$_1$/Rg$_2$/Rg$_L$ | telugu Rg$_1$/Rg$_2$/Rg$_L$ | thai Rg$_1$/Rg$_2$/Rg$_L$ | turkish Rg$_1$/Rg$_2$/Rg$_L$ |
|---|---|---|---|---|---|---|---|---|---|
| Small | mmT5 | s7 | 20.1 / 4.9 / 16.9 | 21.5 / 5.8 / 17.6 | 22.0 / 5.9 / 18.2 | 13.9 / 4.3 / 12.3 | 12.8 / 3.6 / 11.7 | 13.0 / 4.1 / 11.5 | 15.2 / 4.5 / 13.0 |
| Small | mT5$^S$ | s1 | 19.2 / 2.5 / 16.5 | 19.4 / 3.4 / 17.0 | 19.3 / 3.4 / 16.7 | 9.5 / 0.7 / 9.2 | 10.0 / 0.6 / 9.8 | 9.2 / 1.0 / 8.6 | 16.3 / 2.8 / 14.5 |
| Base | mmT5 | s1 | 19.5 / 2.7 / 16.7 | 19.9 / 3.5 / 17.0 | 18.5 / 3.2 / 16.2 | 9.0 / 0.2 / 8.7 | 9.7 / 0.2 / 9.5 | 8.9 / 0.4 / 8.3 | 16.6 / 2.6 / 14.5 |
| Base | mmT5 | s7 | 22.0 / 6.5 / 17.3 | 25.6 / 7.0 / 20.2 | 25.6 / 8.3 / 20.2 | 17.6 / 6.6 / 14.5 | 15.9 / 5.5 / 13.6 | 14.1 / 4.7 / 12.4 | 21.6 / 7.0 / 17.5 |
| Base | mT5$^S$ | s1 | 19.6 / 2.6 / 16.8 | 20.0 / 3.8 / 17.1 | 20.3 / 4.3 / 17.2 | 9.9 / 1.0 / 9.4 | 10.3 / 0.9 / 9.8 | 10.0 / 1.2 / 9.2 | 17.6 / 3.5 / 15.1 |
| Base | mT5$^S$ | s7 | 21.1 / 3.3 / 17.9 | 20.8 / 4.3 / 17.7 | 21.8 / 4.9 / 18.5 | 10.3 / 1.1 / 9.8 | 10.6 / 0.9 / 10.2 | 9.8 / 1.0 / 9.0 | 19.6 / 4.6 / 16.7 |

| | | | ukrainian Rg$_1$/Rg$_2$/Rg$_L$ | urdu Rg$_1$/Rg$_2$/Rg$_L$ | uzbek Rg$_1$/Rg$_2$/Rg$_L$ | vietnamese Rg$_1$/Rg$_2$/Rg$_L$ | welsh Rg$_1$/Rg$_2$/Rg$_L$ | yoruba Rg$_1$/Rg$_2$/Rg$_L$ | *avg* Rg$_1$/Rg$_2$/Rg$_L$ |
|---|---|---|---|---|---|---|---|---|---|
| Small | mmT5 | s7 | 10.7 / 1.7 / 9.9 | 16.8 / 3.8 / 15.2 | 8.9 / 0.6 / 8.7 | 29.7 / 11.6 / 22.1 | 20.6 / 5.4 / 17.6 | 30.0 / 10.5 / 24.2 | 16.7 / 4.7 / 14.4 |
| Small | mT5$^S$ | s1 | 13.4 / 0.9 / 12.8 | 13.1 / 0.3 / 13.0 | 12.8 / 0.6 / 12.3 | 21.2 / 3.7 / 18.2 | 22.3 / 3.3 / 19.9 | 23.8 / 4.0 / 20.9 | 15.5 / 2.2 / 14.2 |
| Base | mmT5 | s1 | 13.3 / 0.6 / 12.6 | 13.1 / 0.2 / 13.0 | 12.7 / 0.4 / 12.2 | 21.1 / 3.4 / 18.1 | 21.3 / 3.1 / 18.5 | 23.9 / 3.6 / 21.3 | 15.4 / 2.0 / 14.0 |
| Base | mmT5 | s7 | 17.0 / 3.7 / 15.0 | 17.6 / 4.6 / 15.4 | 8.6 / 0.7 / 8.1 | 32.7 / 13.7 / 23.6 | 22.8 / 6.6 / 19.1 | 24.8 / 8.8 / 20.0 | **19.7 / 6.2 / 16.4** |
| Base | mT5$^S$ | s1 | 14.1 / 1.2 / 13.1 | 14.0 / 1.2 / 13.4 | 13.5 / 0.8 / 12.8 | 22.3 / 5.0 / 18.6 | 23.0 / 4.3 / 19.5 | 23.5 / 4.0 / 20.1 | 16.2 / 2.8 / 14.5 |
| Base | mT5$^S$ | s7 | 15.0 / 1.6 / 14.0 | 14.2 / 1.0 / 13.8 | 14.0 / 1.0 / 13.2 | 23.0 / 5.4 / 19.3 | 24.7 / 5.7 / 20.8 | 25.0 / 5.4 / 21.4 | 16.8 / 3.0 / 15.0 |

Table 14: XL-Sum$^{en}$ validation results for each of the languages. We report results of different combinations of freezing the model.

| | | amharic Rg$_1$ / Rg$_2$ / Rg$_L$ | arabic Rg$_1$ / Rg$_2$ / Rg$_L$ | azerbaijani Rg$_1$ / Rg$_2$ / Rg$_L$ | bengali Rg$_1$ / Rg$_2$ / Rg$_L$ | burmese Rg$_1$ / Rg$_2$ / Rg$_L$ | chinese_simplified Rg$_1$ / Rg$_2$ / Rg$_L$ | chinese_traditional Rg$_1$ / Rg$_2$ / Rg$_L$ |
|---|---|---|---|---|---|---|---|---|
| small | mmT5 | **37.5 / 19.2** / 27.9 | 36.6 / 16.6 / 28.5 | **28.3 / 13.4 / 23.5** | **35.4 / 19.5 / 28.0** | 43.3 / 23.1 / **34.1** | 36.7 / 17.9 / 31.7 | 37.9 / 19.6 / 32.1 |
| | mT5$^S$ | 37.2 / **19.2 / 28.1** | **36.9 / 16.7 / 28.9** | 27.8 / 13.2 / 23.1 | 34.5 / 18.8 / 27.5 | **43.4 / 23.4** / 34.0 | **37.1 / 18.1 / 32.1** | **38.1 / 19.8 / 32.3** |
| base | mmT5 | **42.9 / 25.1 / 33.0** | **41.9 / 21.5 / 33.0** | **33.6 / 18.3 / 28.4** | **40.7 / 24.7 / 32.4** | **50.0 / 29.9 / 39.6** | **42.2 / 22.3 / 36.6** | **43.0 / 24.0 / 36.7** |
| | mT5$^S$ | 40.1 / 22.0 / 30.0 | 39.7 / 19.4 / 30.9 | 30.3 / 15.3 / 25.2 | 37.9 / 21.7 / 29.5 | 45.7 / 25.3 / 35.6 | 39.8 / 20.4 / 34.3 | 40.8 / 22.3 / 34.7 |

| | | english Rg$_1$ / Rg$_2$ / Rg$_L$ | french Rg$_1$ / Rg$_2$ / Rg$_L$ | gujarati Rg$_1$ / Rg$_2$ / Rg$_L$ | hausa Rg$_1$ / Rg$_2$ / Rg$_L$ | hindi Rg$_1$ / Rg$_2$ / Rg$_L$ | igbo Rg$_1$ / Rg$_2$ / Rg$_L$ | indonesian Rg$_1$ / Rg$_2$ / Rg$_L$ |
|---|---|---|---|---|---|---|---|---|
| small | mmT5 | **38.4 / 14.0 / 30.0** | 38.2 / 17.3 / 29.5 | **33.5 / 18.2 / 26.8** | **41.6 / 21.0 / 32.2** | **39.1 / 21.0 / 31.3** | 41.9 / 21.2 / 30.9 | **37.7** / 17.4 / 29.8 |
| | mT5$^S$ | 37.9 / 13.6 / 29.5 | **38.2 / 17.6 / 29.6** | 33.0 / 17.7 / 26.4 | 40.9 / 20.2 / 31.8 | 38.7 / 20.8 / 30.9 | **43.1 / 21.8 / 31.4** | **37.7** / 17.3 / 30.0 |
| base | mmT5 | **42.4 / 17.7 / 33.3** | **42.6 / 22.1 / 33.3** | **38.3 / 23.1 / 31.1** | **46.2 / 25.3 / 36.4** | **44.1 / 26.4 / 35.9** | **47.2 / 25.8 / 34.4** | **42.5 / 21.9 / 34.2** |
| | mT5$^S$ | 40.9 / 16.3 / 31.8 | 40.2 / 19.4 / 31.1 | 36.0 / 20.4 / 28.7 | 43.4 / 22.6 / 33.8 | 42.0 / 24.1 / 33.7 | 45.0 / 23.6 / 32.9 | 40.5 / 19.9 / 32.1 |

| | | japanese Rg$_1$ / Rg$_2$ / Rg$_L$ | korean Rg$_1$ / Rg$_2$ / Rg$_L$ | kyrgyz Rg$_1$ / Rg$_2$ / Rg$_L$ | marathi Rg$_1$ / Rg$_2$ / Rg$_L$ | nepali Rg$_1$ / Rg$_2$ / Rg$_L$ | pashto Rg$_1$ / Rg$_2$ / Rg$_L$ | persian Rg$_1$ / Rg$_2$ / Rg$_L$ |
|---|---|---|---|---|---|---|---|---|
| small | mmT5 | **41.7** / 22.5 / 33.6 | 37.1 / 19.9 / 31.6 | 26.5 / 11.9 / 21.2 | 32.6 / 18.1 / **26.7** | 38.7 / 22.1 / 31.4 | **43.8 / 22.6 / 34.0** | **43.0 / 22.3 / 33.7** |
| | mT5$^S$ | **41.7 / 22.5 / 33.8** | **37.4 / 20.2 / 32.0** | **26.8 / 12.4 / 21.5** | **32.7 / 18.5** / 26.7 | **39.0 / 22.3 / 31.6** | 43.4 / 22.1 / 33.4 | 42.7 / 21.9 / 33.3 |
| base | mmT5 | **48.5 / 28.5 / 40.0** | **44.2 / 26.9 / 38.0** | **31.3 / 16.5 / 25.2** | **37.3 / 23.0 / 30.6** | **44.4 / 27.7 / 36.3** | **49.1 / 27.9 / 38.4** | **47.0 / 26.4 / 37.1** |
| | mT5$^S$ | 45.4 / 25.6 / 36.7 | 39.0 / 21.8 / 32.9 | 27.6 / 12.9 / 21.7 | 35.4 / 20.8 / 28.8 | 41.6 / 24.9 / 33.5 | 47.0 / 25.3 / 36.0 | 45.3 / 24.5 / 35.3 |

| | | portuguese Rg$_1$ / Rg$_2$ / Rg$_L$ | punjabi Rg$_1$ / Rg$_2$ / Rg$_L$ | russian Rg$_1$ / Rg$_2$ / Rg$_L$ | scottish_gaelic Rg$_1$ / Rg$_2$ / Rg$_L$ | serbian_cyrillic Rg$_1$ / Rg$_2$ / Rg$_L$ | serbian_latin Rg$_1$ / Rg$_2$ / Rg$_L$ | sinhala Rg$_1$ / Rg$_2$ / Rg$_L$ |
|---|---|---|---|---|---|---|---|---|
| small | mmT5 | **40.5 / 18.2 / 30.2** | **40.6 / 21.8 / 30.0** | **32.9 / 13.5 / 25.3** | **36.9 / 17.5 / 28.0** | **30.8 / 10.8 / 23.3** | **30.2 / 10.7 / 22.6** | **36.1 / 21.8 / 28.4** |
| | mT5$^S$ | 40.3 / 17.9 / 30.1 | 40.1 / 21.4 / 29.8 | 32.2 / 13.0 / 24.8 | 36.5 / 16.9 / 27.4 | 30.6 / 10.5 / 23.2 | 29.4 / 10.5 / 22.4 | 35.8 / 21.5 / 28.1 |
| base | mmT5 | **44.2 / 22.1 / 33.3** | **45.4 / 27.0 / 34.6** | **36.8 / 17.1 / 28.6** | **42.0 / 22.7 / 32.0** | **36.5 / 15.6 / 28.1** | **35.0 / 14.7 / 26.4** | **41.7 / 27.4 / 33.9** |
| | mT5$^S$ | 42.5 / 20.3 / 31.7 | 42.7 / 24.2 / 31.7 | 35.0 / 15.6 / 27.0 | 38.2 / 18.6 / 28.4 | 33.7 / 13.0 / 25.2 | 32.8 / 12.4 / 24.1 | 38.0 / 23.3 / 29.7 |

| | | somali Rg$_1$ / Rg$_2$ / Rg$_L$ | spanish Rg$_1$ / Rg$_2$ / Rg$_L$ | swahili Rg$_1$ / Rg$_2$ / Rg$_L$ | tamil Rg$_1$ / Rg$_2$ / Rg$_L$ | telugu Rg$_1$ / Rg$_2$ / Rg$_L$ | thai Rg$_1$ / Rg$_2$ / Rg$_L$ | turkish Rg$_1$ / Rg$_2$ / Rg$_L$ |
|---|---|---|---|---|---|---|---|---|
| small | mmT5 | 37.3 / **17.3 / 27.3** | 31.8 / 11.2 / 23.7 | **39.3 / 19.2 / 30.5** | **31.7 / 18.0 / 26.6** | **28.3 / 15.2 / 23.2** | **31.7 / 17.2** / 24.7 | **33.3 / 17.1 / 28.0** |
| | mT5$^S$ | **37.5** / 17.1 / 27.0 | **32.1 / 11.2 / 24.0** | 39.2 / 18.9 / 30.3 | 31.6 / **18.0** / 26.5 | 27.6 / 14.7 / 22.6 | 31.5 / 17.1 / **24.9** | 33.2 / 17.0 / 27.8 |
| base | mmT5 | **41.2 / 21.0 / 30.3** | **36.3 / 15.0 / 27.4** | **43.8 / 23.6 / 34.5** | **38.0 / 23.6 / 32.2** | **33.4 / 19.8 / 27.4** | **36.8 / 21.6 / 29.3** | **38.4 / 21.7 / 32.6** |
| | mT5$^S$ | 39.3 / 19.1 / 28.6 | 34.4 / 13.4 / 25.8 | 41.3 / 20.8 / 32.0 | 34.9 / 20.5 / 29.0 | 30.6 / 17.2 / 24.7 | 33.2 / 18.4 / 25.9 | 36.1 / 19.7 / 30.4 |

| | | ukrainian Rg$_1$ / Rg$_2$ / Rg$_L$ | urdu Rg$_1$ / Rg$_2$ / Rg$_L$ | uzbek Rg$_1$ / Rg$_2$ / Rg$_L$ | vietnamese Rg$_1$ / Rg$_2$ / Rg$_L$ | welsh Rg$_1$ / Rg$_2$ / Rg$_L$ | yoruba Rg$_1$ / Rg$_2$ / Rg$_L$ | *avg* Rg$_1$ / Rg$_2$ / Rg$_L$ |
|---|---|---|---|---|---|---|---|---|
| small | mmT5 | **31.8 / 13.4 / 25.0** | **42.3 / 22.8 / 33.0** | **28.5** / 13.0 / **23.3** | **44.8 / 23.0 / 30.8** | **41.2 / 20.6 / 31.1** | **43.7 / 21.1 / 32.0** | **36.7 / 18.1 / 28.7** |
| | mT5$^S$ | 31.4 / 13.2 / 24.8 | 41.5 / 22.1 / 32.3 | 28.0 / **13.0** / 22.9 | 44.7 / 22.7 / 30.6 | 39.9 / 19.4 / 30.3 | 43.1 / 20.8 / 31.7 | 36.4 / 17.9 / 28.5 |
| base | mmT5 | **36.2 / 17.3 / 28.5** | **46.7 / 27.8 / 37.3** | **32.8 / 17.0 / 27.0** | **49.0 / 27.6 / 34.6** | **44.0 / 23.9 / 33.7** | **47.5 / 25.0 / 35.6** | **41.6 / 22.8 / 33.0** |
| | mT5$^S$ | 34.1 / 15.5 / 26.9 | 44.9 / 25.6 / 35.2 | 30.7 / 15.0 / 25.0 | 47.6 / 25.6 / 32.7 | 42.5 / 22.0 / 31.8 | 45.9 / 23.1 / 33.6 | 39.1 / 20.3 / 30.5 |

Table 15: Test set results for XL-Sum in the multisource setup. We evaluate using the model which performed best on all the languages in the validation set.

| | | lang | amharic Rg$_1$/Rg$_2$/Rg$_L$ | arabic Rg$_1$/Rg$_2$/Rg$_L$ | azerbaijani Rg$_1$/Rg$_2$/Rg$_L$ | bengali Rg$_1$/Rg$_2$/Rg$_L$ | burmese Rg$_1$/Rg$_2$/Rg$_L$ | chinese_simplified Rg$_1$/Rg$_2$/Rg$_L$ | chinese_traditional Rg$_1$/Rg$_2$/Rg$_L$ |
|---|---|---|---|---|---|---|---|---|---|
| Small | mmT5 | s7 | 29.1 / 13.0 / 22.2 | 37.7 / 17.3 / 29.6 | 19.0 / 7.2 / 15.9 | 27.6 / 12.8 / 22.3 | 38.3 / 17.7 / 29.7 | 38.3 / 19.1 / 33.2 | 39.8 / 21.4 / 34.0 |
| | mT5$^S$ | s1 | 13.0 / 3.1 / 12.0 | 37.6 / 17.3 / 29.4 | 13.4 / 2.6 / 12.1 | 7.4 / 1.1 / 7.2 | 11.9 / 1.9 / 11.4 | 38.5 / 19.5 / 33.4 | 39.9 / 21.6 / 34.1 |
| Base | mmT5 | s1 | 9.4 / 0.3 / 9.3 | 41.2 / 20.9 / 31.9 | 12.3 / 1.8 / 11.2 | 8.3 / 0.4 / 8.1 | 15.1 / 0.3 / 14.9 | 64.1 / 51.2 / 60.8 | 62.9 / 49.8 / 58.8 |
| | | s7 | 34.0 / 16.6 / 25.3 | 42.4 / 22.3 / 33.5 | 26.3 / 11.0 / 21.1 | 34.5 / 18.0 / 26.7 | 40.1 / 19.0 / 30.9 | 48.4 / 29.6 / 43.2 | 49.4 / 31.3 / 43.6 |
| | mT5$^S$ | s1 | 14.9 / 4.2 / 13.6 | 40.4 / 20.1 / 31.5 | 15.9 / 4.4 / 14.1 | 11.4 / 3.1 / 10.6 | 16.5 / 4.1 / 15.1 | 44.3 / 26.2 / 39.3 | 45.4 / 27.8 / 39.7 |
| | | s7 | 10.3 / 3.4 / 9.4 | 40.5 / 20.1 / 31.8 | 12.3 / 4.2 / 10.9 | 6.3 / 2.0 / 5.9 | 22.3 / 6.3 / 19.9 | 41.5 / 21.9 / 36.0 | 42.7 / 24.2 / 36.8 |

| | | lang | english Rg$_1$/Rg$_2$/Rg$_L$ | french Rg$_1$/Rg$_2$/Rg$_L$ | gujarati Rg$_1$/Rg$_2$/Rg$_L$ | hausa Rg$_1$/Rg$_2$/Rg$_L$ | hindi Rg$_1$/Rg$_2$/Rg$_L$ | igbo Rg$_1$/Rg$_2$/Rg$_L$ | indonesian Rg$_1$/Rg$_2$/Rg$_L$ |
|---|---|---|---|---|---|---|---|---|---|
| Small | mmT5 | s7 | 39.0 / 14.5 / 30.6 | 35.4 / 14.2 / 27.1 | 27.1 / 12.5 / 21.7 | 28.4 / 14.2 / 21.7 | 30.1 / 13.4 / 24.1 | 37.7 / 17.4 / 29.0 | 31.4 / 11.8 / 25.0 |
| | mT5$^S$ | s1 | 39.1 / 14.7 / 30.4 | 22.1 / 4.4 / 19.4 | 9.1 / 1.6 / 8.8 | 16.4 / 5.0 / 14.9 | 12.7 / 3.2 / 12.0 | 21.8 / 6.2 / 19.1 | 17.1 / 3.4 / 15.7 |
| Base | mmT5 | s1 | 44.4 / 20.2 / 35.1 | 23.2 / 5.3 / 19.5 | 8.6 / 0.4 / 8.4 | 15.3 / 4.7 / 14.0 | 10.6 / 0.4 / 10.5 | 22.9 / 5.1 / 20.1 | 17.9 / 3.8 / 16.1 |
| | | s7 | 43.9 / 19.4 / 34.8 | 37.3 / 15.2 / 27.8 | 32.1 / 16.1 / 25.0 | 36.6 / 16.4 / 27.6 | 34.8 / 16.7 / 27.1 | 40.3 / 19.7 / 30.3 | 35.8 / 15.0 / 27.6 |
| | mT5$^S$ | s1 | 42.9 / 18.4 / 33.7 | 23.7 / 5.4 / 20.1 | 12.3 / 4.1 / 11.4 | 20.3 / 6.8 / 18.0 | 14.9 / 4.8 / 13.8 | 31.5 / 11.6 / 25.4 | 20.9 / 5.5 / 18.4 |
| | | s7 | 41.9 / 17.2 / 32.8 | 23.9 / 6.4 / 20.5 | 9.3 / 2.9 / 8.7 | 11.9 / 4.2 / 10.8 | 13.4 / 4.1 / 12.6 | 21.3 / 8.2 / 17.2 | 19.3 / 5.9 / 17.0 |

| | | lang | japanese Rg$_1$/Rg$_2$/Rg$_L$ | korean Rg$_1$/Rg$_2$/Rg$_L$ | kyrgyz Rg$_1$/Rg$_2$/Rg$_L$ | marathi Rg$_1$/Rg$_2$/Rg$_L$ | nepali Rg$_1$/Rg$_2$/Rg$_L$ | pashto Rg$_1$/Rg$_2$/Rg$_L$ | persian Rg$_1$/Rg$_2$/Rg$_L$ |
|---|---|---|---|---|---|---|---|---|---|
| Small | mmT5 | s7 | 45.0 / 25.6 / 37.1 | 31.1 / 13.3 / 25.5 | 19.3 / 6.7 / 16.5 | 19.2 / 8.1 / 16.0 | 30.1 / 14.3 / 24.7 | 38.1 / 17.0 / 28.7 | 37.7 / 17.0 / 29.1 |
| | mT5$^S$ | s1 | 41.5 / 22.3 / 33.7 | 8.6 / 2.4 / 7.8 | 9.9 / 1.4 / 9.4 | 11.8 / 3.4 / 10.8 | 8.2 / 1.4 / 7.9 | 18.1 / 3.6 / 16.1 | 21.5 / 5.6 / 18.7 |
| Base | mmT5 | s1 | 46.7 / 27.0 / 38.1 | 10.3 / 1.0 / 10.0 | 10.9 / 0.6 / 10.5 | 9.0 / 0.5 / 8.8 | 9.1 / 0.3 / 9.0 | 18.7 / 3.6 / 16.2 | 18.8 / 4.3 / 16.6 |
| | | s7 | 49.0 / 29.1 / 40.3 | 34.5 / 15.7 / 27.4 | 26.1 / 10.7 / 20.9 | 26.1 / 11.7 / 20.7 | 35.6 / 18.1 / 27.8 | 42.5 / 20.3 / 30.9 | 41.2 / 19.7 / 30.7 |
| | mT5$^S$ | s1 | 45.2 / 25.7 / 36.6 | 11.0 / 4.3 / 10.0 | 12.0 / 2.1 / 11.3 | 13.4 / 4.9 / 12.0 | 12.6 / 3.6 / 11.5 | 21.0 / 4.9 / 18.1 | 23.6 / 6.5 / 19.9 |
| | | s7 | 48.1 / 28.6 / 39.7 | 16.8 / 5.9 / 15.1 | 9.0 / 1.9 / 8.5 | 11.1 / 3.7 / 10.1 | 9.8 / 2.9 / 9.2 | 19.1 / 5.8 / 16.8 | 27.3 / 10.2 / 22.7 |

| | | lang | portuguese Rg$_1$/Rg$_2$/Rg$_L$ | punjabi Rg$_1$/Rg$_2$/Rg$_L$ | russian Rg$_1$/Rg$_2$/Rg$_L$ | scottish_gaelic Rg$_1$/Rg$_2$/Rg$_L$ | serbian_cyrillic Rg$_1$/Rg$_2$/Rg$_L$ | serbian_latin Rg$_1$/Rg$_2$/Rg$_L$ | sinhala Rg$_1$/Rg$_2$/Rg$_L$ |
|---|---|---|---|---|---|---|---|---|---|
| Small | mmT5 | s7 | 33.8 / 12.5 / 26.4 | 31.5 / 13.6 / 23.5 | 25.4 / 8.3 / 20.3 | 29.6 / 12.1 / 24.5 | 21.7 / 6.3 / 18.0 | 11.0 / 3.2 / 9.5 | 29.0 / 14.4 / 22.2 |
| | mT5$^S$ | s1 | 23.3 / 5.3 / 20.7 | 10.6 / 1.1 / 10.3 | 13.0 / 1.5 / 12.5 | 18.6 / 5.6 / 16.9 | 11.3 / 0.9 / 11.2 | 11.3 / 2.3 / 10.2 | 8.7 / 1.1 / 8.5 |
| Base | mmT5 | s1 | 24.9 / 6.7 / 21.0 | 10.4 / 0.7 / 10.3 | 13.7 / 1.0 / 13.0 | 17.4 / 5.4 / 15.4 | 11.9 / 0.8 / 11.6 | 14.9 / 2.3 / 13.4 | 8.1 / 0.3 / 8.0 |
| | | s7 | 38.8 / 15.8 / 28.4 | 37.0 / 17.9 / 26.7 | 30.7 / 11.1 / 23.5 | 31.2 / 14.5 / 24.5 | 28.6 / 9.4 / 22.1 | 19.0 / 5.2 / 16.0 | 34.4 / 19.0 / 25.5 |
| | mT5$^S$ | s1 | 25.6 / 6.6 / 21.6 | 13.2 / 2.7 / 12.4 | 16.1 / 2.9 / 14.8 | 22.8 / 7.6 / 19.6 | 13.8 / 1.7 / 13.1 | 18.4 / 3.8 / 16.2 | 10.2 / 2.3 / 9.7 |
| | | s7 | 26.2 / 7.7 / 22.3 | 8.5 / 1.9 / 8.0 | 12.9 / 2.8 / 11.7 | 20.0 / 7.7 / 17.5 | 10.6 / 1.4 / 10.2 | 7.3 / 1.8 / 6.5 | 5.5 / 1.3 / 5.2 |

| | | lang | somali Rg$_1$/Rg$_2$/Rg$_L$ | spanish Rg$_1$/Rg$_2$/Rg$_L$ | swahili Rg$_1$/Rg$_2$/Rg$_L$ | tamil Rg$_1$/Rg$_2$/Rg$_L$ | telugu Rg$_1$/Rg$_2$/Rg$_L$ | thai Rg$_1$/Rg$_2$/Rg$_L$ | turkish Rg$_1$/Rg$_2$/Rg$_L$ |
|---|---|---|---|---|---|---|---|---|---|
| Small | mmT5 | s7 | 29.3 / 11.6 / 22.7 | 28.5 / 8.2 / 22.3 | 30.0 / 10.9 / 22.9 | 25.3 / 12.0 / 20.6 | 23.4 / 10.6 / 19.2 | 26.3 / 12.8 / 20.3 | 24.8 / 9.9 / 20.5 |
| | mT5$^S$ | s1 | 16.8 / 2.7 / 15.1 | 18.8 / 3.7 / 16.8 | 14.6 / 2.9 / 13.5 | 8.0 / 1.8 / 7.6 | 8.5 / 1.3 / 8.3 | 7.1 / 2.2 / 6.6 | 16.4 / 3.8 / 14.7 |
| Base | mmT5 | s1 | 15.9 / 3.0 / 14.0 | 19.7 / 4.3 / 17.0 | 16.4 / 3.5 / 14.5 | 7.5 / 0.6 / 7.3 | 8.3 / 0.4 / 8.1 | 7.4 / 0.6 / 7.0 | 16.0 / 3.4 / 14.0 |
| | | s7 | 34.0 / 15.0 / 25.1 | 32.6 / 10.4 / 24.0 | 36.0 / 15.0 / 26.4 | 29.6 / 14.8 / 23.2 | 26.8 / 12.8 / 21.1 | 30.7 / 15.7 / 23.3 | 31.2 / 13.8 / 25.1 |
| | mT5$^S$ | s1 | 18.6 / 3.8 / 16.4 | 21.2 / 4.7 / 18.3 | 16.8 / 4.1 / 15.2 | 10.6 / 3.5 / 9.7 | 10.8 / 2.9 / 10.2 | 10.9 / 3.9 / 9.9 | 18.4 / 5.4 / 16.0 |
| | | s7 | 12.6 / 2.5 / 11.5 | 19.8 / 4.9 / 17.3 | 13.8 / 3.5 / 12.7 | 10.0 / 3.9 / 9.1 | 8.7 / 2.7 / 8.0 | 11.1 / 4.7 / 9.8 | 17.8 / 6.7 / 15.4 |

| | | lang | ukrainian Rg$_1$/Rg$_2$/Rg$_L$ | urdu Rg$_1$/Rg$_2$/Rg$_L$ | uzbek Rg$_1$/Rg$_2$/Rg$_L$ | vietnamese Rg$_1$/Rg$_2$/Rg$_L$ | welsh Rg$_1$/Rg$_2$/Rg$_L$ | yoruba Rg$_1$/Rg$_2$/Rg$_L$ | *avg* Rg$_1$/Rg$_2$/Rg$_L$ |
|---|---|---|---|---|---|---|---|---|---|
| Small | mmT5 | s7 | 22.4 / 7.4 / 18.3 | 34.5 / 15.6 / 26.0 | 4.2 / 1.0 / 4.0 | 36.6 / 16.9 / 26.0 | 27.8 / 11.2 / 21.7 | 30.8 / 13.6 / 23.7 | 29.4 / 12.6 / 23.4 |
| | mT5$^S$ | s1 | 11.4 / 1.2 / 11.1 | 15.6 / 3.0 / 12.6 | 1.0 / 1.6 / 10.0 | 16.9 / 3.2 / 15.2 | 11.2 / 3.0 / 16.2 | 13.6 / 4.1 / 17.3 | 17.0 / 4.8 / 15.1 |
| Base | mmT5 | s1 | 12.6 / 1.0 / 12.1 | 13.5 / 1.6 / 12.5 | 10.7 / 0.5 / 10.4 | 18.0 / 3.7 / 15.8 | 20.1 / 3.6 / 17.8 | 20.2 / 3.2 / 18.3 | 20.4 / 6.8 / 17.8 |
| | | s7 | 29.1 / 10.9 / 22.6 | 38.5 / 18.6 / 28.6 | 15.2 / 3.1 / 14.2 | 42.2 / 21.1 / 28.9 | 26.6 / 11.1 / 20.6 | 38.6 / 17.3 / 28.5 | **34.7** / **16.2** / **26.9** |
| | mT5$^S$ | s1 | 14.2 / 2.5 / 13.2 | 15.6 / 3.5 / 14.3 | 12.3 / 2.1 / 11.5 | 23.2 / 6.3 / 19.7 | 22.8 / 4.9 / 19.9 | 26.1 / 6.6 / 22.3 | 18.7 / 6.1 / 16.8 |
| | | s7 | 12.0 / 2.8 / 11.0 | 13.2 / 4.6 / 11.9 | 8.8 / 2.2 / 8.2 | 21.1 / 7.3 / 17.6 | 12.7 / 3.3 / 11.2 | 18.0 / 5.7 / 15.4 | 17.8 / 6.6 / 15.4 |

Table 16: XL-Sum$^{ar,en,ja,zh}$ validation set results for all languages. We report results for the different freezing configurations.

Table of MASSIVE Exact Match (EM) dev accuracies.

| Size | Model | Setting | af_ZA | am_ET | ar_SA | az_AZ | bn_BD | cy_GB | da_DK | de_DE | el_GR | en_US |
|---|---|---|---|---|---|---|---|---|---|---|---|---|
| Small | mmT5 | s1 | 27.4 | 5.3 | 14.0 | 12.9 | 9.7 | 8.8 | 31.1 | 30.8 | 21.1 | 69.4 |
| Small | mmT5 | s6 | 35.6 | 6.2 | 19.2 | 16.9 | 11.4 | 7.8 | 45.4 | 41.4 | 28.6 | 72.8 |
| Small | mmT5 | s7 | 39.5 | 5.6 | 20.9 | 18.3 | 11.9 | 10.0 | 46.6 | 42.6 | 31.4 | 73.0 |
| Small | mmT5 | s10 | 38.6 | 6.9 | 19.9 | 19.0 | 12.4 | 9.2 | 47.0 | 41.8 | 30.2 | 72.2 |
| Small | mmT5 | s14 | 38.6 | 6.6 | 20.1 | 18.4 | 13.6 | 10.3 | 46.7 | 43.5 | 30.1 | 73.2 |
| Small | mT5$^S$ | s1 | 25.6 | 5.5 | 16.6 | 12.8 | 9.8 | 5.6 | 35.1 | 31.0 | 22.8 | 71.0 |
| Small | mT5$^S$ | s6 | 26.4 | 4.8 | 16.8 | 13.0 | 8.6 | 4.1 | 34.6 | 30.6 | 24.1 | 71.3 |
| Small | mT5$^S$ | s7 | 27.3 | 4.4 | 16.9 | 13.2 | 10.0 | 3.9 | 36.6 | 32.3 | 25.9 | 72.4 |
| Small | mT5$^S$ | s10 | 26.7 | 5.0 | 17.3 | 13.7 | 9.3 | 4.8 | 35.7 | 31.3 | 24.7 | 71.4 |
| Small | mT5$^S$ | s14 | 28.4 | 4.8 | 17.0 | 12.4 | 10.4 | 5.1 | 36.5 | 34.5 | 27.1 | 72.3 |
| Base | mmT5 | s1 | 38.4 | 21.7 | 27.4 | 27.0 | 23.9 | 22.0 | 44.7 | 43.2 | 31.6 | 70.8 |
| Base | mmT5 | s6 | 53.5 | 24.1 | 35.6 | 36.8 | 31.1 | 26.9 | 58.1 | 54.4 | 46.9 | 73.8 |
| Base | mmT5 | s7 | 52.2 | 20.6 | 34.5 | 34.4 | 30.1 | 24.2 | 56.6 | 54.7 | 44.9 | 73.4 |
| Base | mmT5 | s10 | 55.4 | 30.6 | 39.2 | 40.7 | 36.5 | 33.7 | 58.9 | 55.0 | 48.4 | 73.4 |
| Base | mmT5 | s14 | 51.6 | 21.4 | 33.8 | 33.7 | 28.5 | 25.5 | 55.3 | 53.1 | 40.5 | 72.9 |
| Base | mT5$^S$ | s1 | 41.5 | 15.7 | 27.0 | 25.1 | 24.1 | 12.3 | 49.9 | 48.1 | 38.6 | 72.8 |
| Base | mT5$^S$ | s6 | 42.1 | 16.3 | 28.6 | 29.2 | 26.5 | 11.6 | 49.9 | 51.1 | 40.4 | 73.0 |
| Base | mT5$^S$ | s7 | 42.8 | 14.7 | 27.2 | 27.6 | 24.1 | 9.5 | 51.6 | 51.5 | 40.7 | 74.6 |
| Base | mT5$^S$ | s10 | 43.6 | 19.1 | 29.8 | 30.6 | 28.8 | 11.7 | 51.2 | 50.9 | 40.5 | 73.3 |
| Base | mT5$^S$ | s14 | 43.6 | 16.4 | 28.0 | 28.3 | 26.9 | 10.3 | 51.5 | 51.1 | 40.5 | 74.1 |

| Size | Model | Setting | es_ES | fa_IR | fi_FI | fr_FR | hi_IN | hu_HU | hy_AM | id_ID | is_IS | it_IT |
|---|---|---|---|---|---|---|---|---|---|---|---|---|
| Small | mmT5 | s1 | 29.6 | 15.9 | 23.4 | 31.0 | 12.7 | 21.0 | 10.8 | 28.5 | 18.1 | 29.3 |
| Small | mmT5 | s6 | 42.9 | 23.3 | 33.0 | 42.1 | 19.3 | 29.5 | 16.8 | 36.4 | 23.9 | 40.2 |
| Small | mmT5 | s7 | 44.0 | 25.5 | 36.5 | 43.8 | 22.1 | 30.2 | 19.0 | 39.9 | 25.1 | 42.1 |
| Small | mmT5 | s10 | 44.1 | 24.6 | 33.5 | 43.1 | 20.2 | 30.1 | 19.6 | 39.1 | 24.9 | 41.6 |
| Small | mmT5 | s14 | 44.9 | 23.9 | 34.7 | 42.6 | 21.7 | 31.9 | 18.6 | 40.5 | 25.8 | 41.2 |
| Small | mT5$^S$ | s1 | 34.2 | 18.9 | 21.2 | 36.0 | 17.2 | 20.6 | 14.6 | 29.5 | 15.9 | 32.2 |
| Small | mT5$^S$ | s6 | 36.4 | 19.4 | 21.1 | 36.5 | 14.2 | 21.9 | 14.1 | 30.0 | 15.7 | 32.1 |
| Small | mT5$^S$ | s7 | 37.5 | 19.2 | 22.0 | 36.5 | 15.2 | 22.3 | 13.8 | 32.3 | 16.6 | 33.9 |
| Small | mT5$^S$ | s10 | 36.2 | 19.4 | 23.0 | 36.6 | 14.2 | 22.2 | 14.9 | 29.7 | 16.6 | 31.7 |
| Small | mT5$^S$ | s14 | 38.3 | 20.3 | 22.2 | 38.2 | 15.6 | 22.6 | 16.0 | 33.1 | 16.6 | 33.0 |
| Base | mmT5 | s1 | 35.9 | 32.4 | 31.3 | 38.5 | 28.5 | 30.8 | 26.0 | 42.9 | 29.0 | 36.9 |
| Base | mmT5 | s6 | 51.7 | 45.1 | 51.9 | 51.2 | 42.6 | 41.1 | 37.6 | 54.1 | 41.2 | 51.6 |
| Base | mmT5 | s7 | 50.8 | 43.1 | 49.7 | 52.4 | 39.6 | 41.2 | 35.1 | 51.8 | 38.2 | 50.8 |
| Base | mmT5 | s10 | 54.4 | 46.6 | 53.8 | 51.1 | 44.5 | 45.5 | 40.3 | 55.0 | 43.3 | 52.5 |
| Base | mmT5 | s14 | 49.4 | 40.2 | 48.8 | 50.5 | 36.2 | 40.3 | 33.1 | 50.3 | 37.7 | 50.3 |
| Base | mT5$^S$ | s1 | 45.8 | 33.5 | 36.9 | 47.9 | 35.1 | 33.2 | 29.5 | 44.6 | 27.6 | 45.5 |
| Base | mT5$^S$ | s6 | 49.4 | 36.5 | 39.5 | 49.7 | 33.2 | 34.5 | 33.6 | 45.9 | 29.2 | 48.0 |
| Base | mT5$^S$ | s7 | 49.9 | 35.5 | 39.6 | 49.1 | 31.8 | 33.0 | 29.6 | 43.3 | 26.7 | 48.6 |
| Base | mT5$^S$ | s10 | 49.6 | 39.9 | 39.1 | 50.1 | 35.7 | 34.3 | 35.4 | 47.6 | 28.7 | 48.5 |
| Base | mT5$^S$ | s14 | 51.0 | 36.8 | 40.0 | 50.7 | 34.9 | 33.3 | 29.8 | 44.7 | 29.1 | 47.9 |

| Size | Model | Setting | ja_JP | jv_ID | ka_GE | km_KH | kn_IN | ko_KR | lv_LV | ml_IN | mn_MN | ms_MY |
|---|---|---|---|---|---|---|---|---|---|---|---|---|
| Small | mmT5 | s1 | 23.5 | 14.4 | 11.9 | 16.1 | 10.3 | 14.9 | 17.1 | 12.7 | 8.4 | 25.6 |
| Small | mmT5 | s6 | 26.7 | 16.0 | 16.2 | 21.0 | 11.7 | 16.7 | 25.9 | 15.2 | 9.3 | 32.6 |
| Small | mmT5 | s7 | 28.8 | 20.3 | 16.2 | 22.2 | 13.8 | 18.0 | 29.4 | 14.6 | 10.6 | 36.8 |
| Small | mmT5 | s10 | 26.9 | 18.3 | 18.1 | 21.5 | 12.9 | 18.9 | 28.2 | 16.2 | 10.9 | 32.7 |
| Small | mmT5 | s14 | 27.4 | 18.7 | 15.2 | 23.1 | 13.4 | 18.9 | 29.0 | 16.7 | 10.6 | 35.7 |
| Small | mT5$^S$ | s1 | 24.5 | 9.6 | 13.0 | 15.7 | 10.8 | 15.5 | 15.2 | 10.1 | 7.7 | 24.8 |
| Small | mT5$^S$ | s6 | 24.4 | 8.0 | 11.5 | 15.8 | 10.0 | 12.4 | 14.5 | 10.3 | 7.1 | 24.9 |
| Small | mT5$^S$ | s7 | 24.6 | 7.4 | 11.3 | 16.6 | 11.1 | 14.9 | 15.8 | 10.9 | 6.6 | 27.4 |
| Small | mT5$^S$ | s10 | 23.5 | 7.7 | 12.3 | 16.5 | 10.8 | 14.2 | 16.2 | 10.2 | 7.0 | 26.2 |
| Small | mT5$^S$ | s14 | 24.3 | 7.7 | 12.5 | 16.6 | 10.3 | 15.4 | 16.3 | 11.3 | 7.2 | 28.9 |
| Base | mmT5 | s1 | 33.1 | 23.2 | 22.8 | 27.4 | 22.8 | 30.1 | 33.1 | 23.3 | 18.8 | 39.8 |
| Base | mmT5 | s6 | 41.1 | 36.6 | 34.1 | 36.4 | 29.8 | 37.8 | 48.5 | 33.8 | 27.8 | 50.7 |
| Base | mmT5 | s7 | 37.0 | 31.8 | 30.8 | 34.1 | 26.8 | 36.8 | 45.3 | 31.1 | 25.8 | 48.1 |
| Base | mmT5 | s10 | 40.6 | 39.1 | 37.0 | 39.2 | 33.3 | 42.0 | 51.8 | 35.4 | 33.0 | 53.0 |
| Base | mmT5 | s14 | 33.8 | 29.6 | 33.1 | 33.5 | 25.8 | 33.9 | 44.9 | 30.2 | 26.5 | 47.4 |
| Base | mT5$^S$ | s1 | 32.8 | 16.5 | 23.7 | 28.1 | 24.8 | 28.8 | 30.1 | 25.6 | 16.3 | 40.3 |
| Base | mT5$^S$ | s6 | 37.0 | 16.1 | 26.7 | 29.9 | 26.0 | 31.1 | 30.7 | 27.0 | 17.3 | 41.0 |
| Base | mT5$^S$ | s7 | 38.0 | 15.3 | 25.6 | 30.0 | 24.0 | 30.7 | 29.0 | 24.6 | 17.5 | 41.2 |
| Base | mT5$^S$ | s10 | 37.2 | 17.4 | 29.8 | 31.6 | 28.2 | 32.8 | 32.0 | 29.4 | 20.0 | 43.3 |
| Base | mT5$^S$ | s14 | 35.8 | 14.5 | 26.9 | 28.9 | 24.5 | 31.9 | 30.6 | 25.6 | 18.6 | 41.1 |

| Size | Model | Setting | my_MM | nb_NO | nl_NL | pl_PL | pt_PT | ro_RO | ru_RU | sl_SL | sq_AL | sv_SE |
|---|---|---|---|---|---|---|---|---|---|---|---|---|
| Small | mmT5 | s1 | 13.3 | 29.5 | 31.8 | 28.1 | 30.2 | 26.2 | 19.8 | 22.0 | 21.6 | 32.0 |
| Small | mmT5 | s6 | 13.0 | 42.5 | 42.5 | 35.4 | 42.4 | 36.4 | 36.0 | 28.5 | 27.0 | 44.9 |
| Small | mmT5 | s7 | 14.4 | 46.0 | 45.9 | 36.7 | 44.3 | 39.9 | 36.9 | 30.6 | 30.8 | 46.5 |
| Small | mmT5 | s10 | 14.4 | 42.9 | 43.5 | 37.7 | 43.2 | 39.5 | 37.2 | 30.3 | 30.1 | 46.1 |
| Small | mmT5 | s14 | 16.0 | 45.3 | 46.3 | 37.5 | 44.0 | 40.8 | 36.4 | 30.0 | 30.1 | 47.2 |
| Small | mT5$^S$ | s1 | 14.9 | 27.7 | 38.3 | 28.3 | 34.0 | 28.6 | 29.1 | 16.8 | 18.7 | 34.2 |
| Small | mT5$^S$ | s6 | 13.0 | 29.7 | 37.2 | 28.7 | 34.2 | 29.8 | 29.0 | 16.3 | 18.6 | 36.1 |
| Small | mT5$^S$ | s7 | 13.2 | 32.0 | 38.5 | 28.6 | 35.4 | 30.6 | 29.2 | 17.4 | 19.1 | 37.5 |
| Small | mT5$^S$ | s10 | 15.3 | 31.1 | 37.1 | 27.7 | 34.7 | 29.2 | 30.6 | 17.3 | 17.9 | 35.3 |
| Small | mT5$^S$ | s14 | 14.4 | 31.7 | 39.6 | 29.3 | 36.4 | 31.8 | 31.0 | 18.2 | 18.4 | 37.1 |
| Base | mmT5 | s1 | 29.5 | 43.3 | 40.1 | 34.9 | 38.8 | 39.4 | 28.2 | 33.4 | 31.3 | 43.9 |
| Base | mmT5 | s6 | 28.3 | 54.1 | 56.6 | 49.7 | 52.9 | 55.7 | 46.5 | 49.2 | 46.5 | 57.0 |
| Base | mmT5 | s7 | 24.9 | 54.1 | 54.1 | 47.5 | 53.0 | 54.5 | 48.2 | 46.9 | 45.2 | 56.6 |
| Base | mmT5 | s10 | 33.6 | 56.3 | 56.1 | 50.6 | 55.3 | 56.0 | 48.4 | 51.2 | 48.5 | 58.6 |
| Base | mmT5 | s14 | 28.3 | 53.1 | 53.5 | 47.3 | 51.2 | 51.8 | 46.8 | 45.4 | 42.4 | 54.9 |
| Base | mT5$^S$ | s1 | 29.3 | 46.2 | 49.4 | 41.9 | 46.7 | 44.1 | 39.6 | 34.6 | 31.6 | 49.5 |
| Base | mT5$^S$ | s6 | 30.8 | 47.7 | 50.5 | 44.6 | 48.5 | 47.4 | 40.5 | 36.1 | 34.1 | 52.6 |
| Base | mT5$^S$ | s7 | 29.3 | 48.4 | 50.7 | 45.0 | 50.0 | 47.4 | 39.6 | 34.0 | 32.9 | 51.9 |
| Base | mT5$^S$ | s10 | 32.8 | 48.5 | 50.8 | 44.4 | 49.8 | 48.6 | 40.6 | 36.6 | 35.1 | 53.2 |
| Base | mT5$^S$ | s14 | 28.5 | 47.9 | 50.0 | 46.3 | 50.0 | 47.8 | 41.6 | 36.3 | 34.4 | 52.2 |

| Size | Model | Setting | sw_KE | ta_IN | te_IN | th_TH | tr_TR | ur_PK | vi_VN | zh_CN | zh_TW | avg |
|---|---|---|---|---|---|---|---|---|---|---|---|---|
| Small | mmT5 | s1 | 13.9 | 13.8 | 10.7 | 24.4 | 20.4 | 9.5 | 16.5 | 17.7 | 16.3 | 20.5 |
| Small | mmT5 | s6 | 15.9 | 15.7 | 11.8 | 32.1 | 22.5 | 12.3 | 17.0 | 19.9 | 16.2 | 26.4 |
| Small | mmT5 | s7 | 18.4 | 17.1 | 11.8 | 34.3 | 25.7 | 13.4 | 18.5 | 20.8 | 18.8 | 28.4 |
| Small | mmT5 | s10 | 18.1 | 18.4 | 12.9 | 33.6 | 24.9 | 12.7 | 18.8 | 19.4 | 17.3 | 27.8 |
| Small | mmT5 | s14 | 18.1 | 17.9 | 11.4 | 34.2 | 26.7 | 14.1 | 19.0 | 19.9 | 18.3 | 28.3 |
| Small | mT5$^S$ | s1 | 8.5 | 14.5 | 11.4 | 28.2 | 17.8 | 9.9 | 14.7 | 17.6 | 14.8 | 21.0 |
| Small | mT5$^S$ | s6 | 8.3 | 14.7 | 10.4 | 29.6 | 17.9 | 10.0 | 13.6 | 18.4 | 14.5 | 20.9 |
| Small | mT5$^S$ | s7 | 8.8 | 14.2 | 9.6 | 32.1 | 18.8 | 8.5 | 12.4 | 17.1 | 15.1 | 21.6 |
| Small | mT5$^S$ | s10 | 8.6 | 15.2 | 10.8 | 30.8 | 17.7 | 9.9 | 13.9 | 18.3 | 14.2 | 21.3 |
| Small | mT5$^S$ | s14 | 9.1 | 15.7 | 11.4 | 32.2 | 18.5 | 9.4 | 12.8 | 17.7 | 15.1 | 22.1 |
| Base | mmT5 | s1 | 28.3 | 24.4 | 21.7 | 37.0 | 33.7 | 24.4 | 31.6 | 25.3 | 24.4 | 32.1 |
| Base | mmT5 | s6 | 40.4 | 34.5 | 30.4 | 50.4 | 42.6 | 33.9 | 38.5 | 31.6 | 31.2 | 43.2 |
| Base | mmT5 | s7 | 35.0 | 32.7 | 28.0 | 47.9 | 40.9 | 29.4 | 36.2 | 26.0 | 23.9 | 41.0 |
| Base | mmT5 | s10 | 40.8 | 36.1 | 34.1 | 49.4 | 46.1 | 38.2 | 43.1 | 29.6 | 28.5 | 45.4 |
| Base | mmT5 | s14 | 35.2 | 32.0 | 27.0 | 44.4 | 34.8 | 29.1 | 34.8 | 24.6 | 21.7 | 39.9 |
| Base | mT5$^S$ | s1 | 18.2 | 27.1 | 24.1 | 40.9 | 32.4 | 23.4 | 32.2 | 22.8 | 22.1 | 33.8 |
| Base | mT5$^S$ | s6 | 19.3 | 28.3 | 26.9 | 44.3 | 37.1 | 25.6 | 33.4 | 27.9 | 26.4 | 35.8 |
| Base | mT5$^S$ | s7 | 17.2 | 27.0 | 24.9 | 42.6 | 35.9 | 23.7 | 31.5 | 25.9 | 26.1 | 34.9 |
| Base | mT5$^S$ | s10 | 19.4 | 29.6 | 28.8 | 45.8 | 38.5 | 28.2 | 34.1 | 29.5 | 27.9 | 37.0 |
| Base | mT5$^S$ | s14 | 17.6 | 26.6 | 25.1 | 42.5 | 36.4 | 24.6 | 30.9 | 25.2 | 24.2 | 35.4 |

Table 17: MASSIVE Exact Match (EM) dev accuracies of all models and settings for all languages.

| | | lang | english Rg$_1$/Rg$_2$/Rg$_L$ | french Rg$_1$/Rg$_2$/Rg$_L$ | gujarati Rg$_1$/Rg$_2$/Rg$_L$ | hausa Rg$_1$/Rg$_2$/Rg$_L$ | hindi Rg$_1$/Rg$_2$/Rg$_L$ | igbo Rg$_1$/Rg$_2$/Rg$_L$ | indonesian Rg$_1$/Rg$_2$/Rg$_L$ |
|---|---|---|---|---|---|---|---|---|---|
| Small | mmT5 | s1 | 38.5 / 30.0 / 33.5 | 38.1 / 29.3 / 31.5 | 33.5 / 27.0 / 31.2 | 41.9 / 32.6 / 36.0 | 39.0 / 31.2 / 35.7 | 43.0 / 31.8 / 34.5 | 38.0 / 30.0 / 33.8 |
| | | s6 | 36.1 / 28.4 / 32.7 | 38.0 / 29.4 / 32.4 | 32.0 / 26.0 / 31.9 | 39.4 / 30.9 / 36.1 | 36.8 / 29.5 / 35.3 | 38.2 / 28.8 / 34.6 | 35.6 / 28.1 / 33.2 |
| | | s7 | 37.3 / 29.1 / 33.5 | 37.9 / 29.1 / 33.0 | 32.8 / 26.5 / 31.8 | 40.4 / 31.4 / 37.1 | 38.1 / 30.4 / 35.8 | 41.0 / 30.8 / 35.0 | 36.5 / 28.9 / 34.1 |
| | | s10 | 36.4 / 28.5 / 32.4 | 37.9 / 29.5 / 32.0 | 32.5 / 26.5 / 31.2 | 39.4 / 30.7 / 35.6 | 36.9 / 29.6 / 34.9 | 38.5 / 29.3 / 33.8 | 35.9 / 28.3 / 32.9 |
| | | s14 | 36.7 / 28.7 / 33.3 | 38.3 / 29.5 / 33.1 | 32.8 / 26.7 / 31.5 | 40.4 / 31.4 / 36.6 | 37.8 / 30.2 / 35.7 | 41.3 / 31.2 / 34.9 | 36.3 / 28.6 / 34.1 |
| | mT5$^S$ | s1 | 38.3 / 29.8 / 31.8 | 38.3 / 29.6 / 30.9 | 33.0 / 26.5 / 29.1 | 41.3 / 32.0 / 34.2 | 38.6 / 30.8 / 33.7 | 42.8 / 31.6 / 31.9 | 37.7 / 30.0 / 32.1 |
| | | s6 | 36.2 / 28.3 / 30.0 | 37.2 / 28.8 / 30.3 | 31.9 / 26.0 / 29.2 | 38.0 / 29.6 / 32.3 | 36.9 / 29.4 / 32.5 | 37.4 / 28.5 / 31.7 | 35.0 / 27.6 / 30.7 |
| | | s7 | 36.8 / 28.5 / 30.7 | 37.6 / 28.9 / 30.7 | 32.3 / 26.1 / 28.8 | 38.3 / 29.9 / 33.1 | 37.0 / 29.6 / 32.9 | 38.4 / 28.7 / 31.9 | 35.5 / 27.9 / 31.0 |
| | | s10 | 35.4 / 27.7 / 30.6 | 36.8 / 28.6 / 30.5 | 31.8 / 25.9 / 29.3 | 37.2 / 28.9 / 32.6 | 36.1 / 28.8 / 32.9 | 35.6 / 27.4 / 32.4 | 34.3 / 27.0 / 30.9 |
| | | s14 | 36.7 / 28.5 / 31.0 | 37.6 / 28.9 / 30.8 | 32.3 / 26.0 / 29.5 | 38.7 / 30.1 / 33.6 | 37.4 / 29.8 / 33.6 | 38.4 / 28.9 / 32.5 | 35.6 / 28.1 / 31.5 |
| Base | mmT5 | s1 | 42.6 / 17.9 / 33.5 | 41.2 / 20.2 / 31.5 | 38.8 / 23.2 / 31.2 | 45.6 / 25.3 / 36.0 | 44.0 / 26.2 / 35.7 | 46.3 / 25.0 / 34.5 | 42.3 / 21.5 / 33.8 |
| | | s6 | 41.9 / 17.0 / 32.7 | 41.6 / 20.9 / 32.4 | 39.1 / 23.9 / 31.9 | 46.1 / 25.1 / 36.1 | 43.3 / 25.7 / 35.3 | 46.7 / 25.5 / 34.6 | 41.6 / 21.0 / 33.2 |
| | | s7 | 42.7 / 17.9 / 33.5 | 42.6 / 21.8 / 33.0 | 39.2 / 24.0 / 31.8 | 46.9 / 26.2 / 37.1 | 44.0 / 26.4 / 35.8 | 47.3 / 26.0 / 35.0 | 42.6 / 22.0 / 34.1 |
| | | s10 | 41.6 / 16.7 / 32.4 | 41.5 / 20.7 / 32.0 | 38.3 / 23.0 / 31.2 | 45.6 / 24.7 / 35.6 | 42.9 / 25.3 / 34.9 | 45.6 / 24.9 / 33.8 | 41.2 / 20.6 / 32.9 |
| | | s14 | 42.5 / 17.6 / 33.3 | 42.7 / 21.9 / 33.1 | 38.9 / 23.4 / 31.5 | 46.9 / 26.1 / 36.6 | 44.0 / 26.2 / 35.7 | 47.2 / 25.8 / 34.9 | 42.6 / 21.8 / 34.1 |
| | mT5$^S$ | s1 | 41.0 / 16.4 / 31.8 | 40.0 / 19.1 / 30.9 | 36.6 / 21.0 / 29.1 | 44.1 / 23.0 / 34.2 | 42.0 / 24.0 / 33.7 | 43.5 / 22.9 / 31.9 | 40.5 / 20.0 / 32.1 |
| | | s6 | 38.7 / 14.0 / 30.0 | 39.4 / 18.8 / 30.3 | 35.9 / 20.8 / 29.2 | 41.7 / 20.7 / 32.3 | 40.6 / 22.6 / 32.5 | 42.8 / 22.4 / 31.7 | 38.8 / 18.1 / 30.7 |
| | | s7 | 39.7 / 14.9 / 30.7 | 40.1 / 19.1 / 30.7 | 35.9 / 20.7 / 28.8 | 42.8 / 21.7 / 33.1 | 41.0 / 23.1 / 32.9 | 43.3 / 22.9 / 31.9 | 39.2 / 18.7 / 31.0 |
| | | s10 | 39.4 / 14.6 / 30.6 | 39.6 / 18.9 / 30.5 | 35.8 / 21.0 / 29.3 | 42.0 / 21.0 / 32.6 | 40.9 / 23.0 / 32.9 | 43.7 / 23.0 / 32.4 | 39.0 / 18.4 / 30.9 |
| | | s14 | 40.0 / 15.3 / 31.0 | 39.9 / 19.2 / 30.8 | 36.5 / 21.6 / 29.5 | 43.2 / 22.2 / 33.6 | 41.7 / 23.9 / 33.6 | 44.3 / 23.4 / 32.5 | 39.5 / 19.1 / 31.5 |

Table 18: XL-Sum results . Different configurations of freezing.

| | | lang | japanese Rg$_1$/Rg$_2$/Rg$_L$ | korean Rg$_1$/Rg$_2$/Rg$_L$ | kyrgyz Rg$_1$/Rg$_2$/Rg$_L$ | marathi Rg$_1$/Rg$_2$/Rg$_L$ | nepali Rg$_1$/Rg$_2$/Rg$_L$ | pashto Rg$_1$/Rg$_2$/Rg$_L$ | persian Rg$_1$/Rg$_2$/Rg$_L$ |
|---|---|---|---|---|---|---|---|---|---|
| Small | mmT5 | s1 | 42.2 / 34.3 / 39.5 | 38.6 / 33.2 / 36.7 | 26.7 / 21.1 / 23.8 | 32.6 / 26.5 / 30.2 | 39.8 / 32.6 / 36.5 | 44.4 / 34.1 / 38.6 | 43.1 / 33.8 / 37.3 |
| | | s6 | 39.1 / 32.5 / 39.9 | 37.9 / 32.3 / 38.8 | 25.6 / 21.2 / 25.3 | 30.6 / 25.4 / 29.8 | 36.8 / 30.1 / 36.9 | 42.7 / 33.1 / 38.4 | 41.4 / 32.6 / 36.8 |
| | | s7 | 41.5 / 34.3 / 41.1 | 39.3 / 34.0 / 39.2 | 26.4 / 21.6 / 25.4 | 31.9 / 26.3 / 30.6 | 38.3 / 31.3 / 36.7 | 43.7 / 33.9 / 38.9 | 42.0 / 32.7 / 37.6 |
| | | s10 | 39.8 / 32.9 / 39.5 | 38.3 / 32.9 / 38.7 | 26.3 / 21.8 / 25.2 | 30.9 / 25.8 / 30.2 | 37.3 / 30.4 / 36.3 | 42.5 / 33.1 / 38.0 | 41.6 / 32.7 / 36.6 |
| | | s14 | 41.4 / 34.0 / 41.1 | 39.4 / 34.0 / 39.2 | 26.1 / 21.7 / 25.4 | 32.4 / 26.6 / 30.5 | 38.3 / 31.1 / 37.3 | 43.4 / 33.6 / 38.5 | 41.8 / 32.7 / 37.4 |
| | mT5$^S$ | s1 | 42.9 / 34.8 / 37.9 | 38.6 / 32.9 / 35.1 | 26.4 / 21.4 / 22.1 | 32.9 / 26.9 / 28.7 | 39.0 / 31.6 / 34.5 | 44.2 / 34.0 / 36.2 | 42.8 / 33.5 / 35.8 |
| | | s6 | 39.0 / 32.3 / 37.2 | 38.3 / 33.1 / 34.7 | 24.6 / 20.7 / 23.6 | 31.1 / 25.7 / 25.7 | 36.9 / 30.3 / 33.6 | 41.3 / 32.1 / 35.5 | 40.8 / 31.9 / 34.2 |
| | | s7 | 40.1 / 33.1 / 37.5 | 38.2 / 33.0 / 36.1 | 25.4 / 21.1 / 23.7 | 31.9 / 26.2 / 28.3 | 37.3 / 30.4 / 34.1 | 40.4 / 31.3 / 36.0 | 41.7 / 32.5 / 34.6 |
| | | s10 | 37.8 / 31.2 / 37.5 | 36.5 / 31.5 / 34.9 | 23.9 / 20.0 / 23.9 | 27.9 / 22.5 / 28.4 | 36.0 / 29.4 / 34.0 | 40.4 / 31.3 / 36.0 | 40.4 / 31.4 / 34.6 |
| | | s14 | 40.4 / 33.1 / 38.6 | 38.3 / 33.0 / 37.1 | 25.2 / 21.0 / 23.7 | 32.0 / 26.2 / 28.8 | 37.3 / 30.2 / 34.8 | 42.5 / 33.1 / 36.3 | 41.5 / 32.4 / 35.0 |
| Base | mmT5 | s1 | 47.9 / 28.3 / 39.5 | 43.0 / 25.6 / 36.7 | 30.1 / 14.9 / 23.8 | 37.1 / 22.4 / 30.2 | 44.8 / 28.1 / 36.5 | 49.7 / 28.5 / 38.6 | 47.3 / 26.5 / 37.3 |
| | | s6 | 48.5 / 28.3 / 39.9 | 45.0 / 27.8 / 38.8 | 31.3 / 16.3 / 25.3 | 36.2 / 21.5 / 29.8 | 44.7 / 28.4 / 36.9 | 49.0 / 28.1 / 38.4 | 46.5 / 25.9 / 36.8 |
| | | s7 | 49.8 / 29.8 / 41.1 | 45.4 / 28.3 / 39.2 | 31.5 / 16.4 / 25.4 | 37.0 / 22.5 / 30.6 | 44.9 / 28.5 / 36.7 | 49.7 / 28.7 / 38.9 | 47.4 / 26.8 / 37.6 |
| | | s10 | 48.0 / 27.9 / 39.5 | 44.9 / 27.6 / 38.7 | 30.8 / 16.1 / 25.2 | 36.0 / 21.6 / 30.2 | 44.1 / 27.7 / 36.3 | 48.4 / 27.5 / 38.0 | 46.3 / 25.6 / 36.6 |
| | | s14 | 49.7 / 29.8 / 41.1 | 45.4 / 28.2 / 39.2 | 31.8 / 16.7 / 25.4 | 37.2 / 22.5 / 30.5 | 45.4 / 29.1 / 37.3 | 49.4 / 28.4 / 38.5 | 47.4 / 26.7 / 37.4 |
| | mT5$^S$ | s1 | 46.3 / 26.4 / 37.9 | 40.8 / 24.0 / 35.1 | 28.4 / 13.3 / 22.1 | 35.3 / 20.4 / 28.7 | 43.1 / 26.3 / 34.5 | 47.1 / 25.7 / 36.2 | 45.7 / 24.9 / 35.8 |
| | | s6 | 45.0 / 25.3 / 37.2 | 40.5 / 23.3 / 34.7 | 28.8 / 14.2 / 23.6 | 32.0 / 17.3 / 25.7 | 41.1 / 24.8 / 33.6 | 45.8 / 24.6 / 35.5 | 44.0 / 23.1 / 34.2 |
| | | s7 | 46.0 / 26.0 / 37.5 | 42.0 / 24.7 / 36.1 | 29.4 / 14.6 / 23.7 | 34.8 / 20.1 / 28.3 | 41.9 / 25.4 / 34.1 | 46.2 / 25.0 / 35.8 | 44.4 / 23.4 / 34.6 |
| | | s10 | 45.3 / 25.6 / 37.5 | 40.4 / 23.3 / 34.9 | 28.9 / 14.6 / 23.9 | 34.6 / 20.2 / 28.4 | 41.7 / 25.3 / 34.0 | 46.2 / 25.0 / 36.0 | 44.4 / 23.6 / 34.6 |
| | | s14 | 46.9 / 26.9 / 38.6 | 43.2 / 26.1 / 37.1 | 29.3 / 14.5 / 23.7 | 35.2 / 20.8 / 28.8 | 42.6 / 26.1 / 34.8 | 46.9 / 25.7 / 36.3 | 44.9 / 23.9 / 35.0 |

Table 19: Validation set results for XL-Sum in the multisource setup for languages English, French, Gujarati, Hausa, Hindi, Igbo, and Indonesian. We report results for the different freezing configurations.

| | | lang | portuguese Rg$_1$/Rg$_2$/Rg$_L$ | punjabi Rg$_1$/Rg$_2$/Rg$_L$ | russian Rg$_1$/Rg$_2$/Rg$_L$ | scottish_gaelic Rg$_1$/Rg$_2$/Rg$_L$ | serbian_cyrillic Rg$_1$/Rg$_2$/Rg$_L$ | serbian_latin Rg$_1$/Rg$_2$/Rg$_L$ | sinhala Rg$_1$/Rg$_2$/Rg$_L$ |
|---|---|---|---|---|---|---|---|---|---|
| Small | mmT5 | s1 | 40.6 / 30.4 / 33.5 | 39.7 / 29.0 / 33.3 | 33.1 / 25.5 / 28.8 | 37.1 / 28.1 / 30.9 | 30.5 / 23.1 / 25.6 | 30.4 / 23.0 / 26.1 | 36.4 / 29.2 / 31.9 |
| | | s6 | 38.6 / 29.0 / 32.9 | 37.7 / 28.2 / 33.7 | 31.0 / 23.9 / 27.9 | 34.8 / 27.4 / 32.2 | 29.0 / 22.5 / 27.3 | 23.6 / 19.1 / 26.4 | 35.2 / 28.3 / 34.7 |
| | | s7 | 39.5 / 29.3 / 33.7 | 38.8 / 28.8 / 33.9 | 31.7 / 24.4 / 28.8 | 36.3 / 28.2 / 32.1 | 29.4 / 22.7 / 27.9 | 26.1 / 20.5 / 27.0 | 35.5 / 28.7 / 35.0 |
| | | s10 | 38.8 / 29.0 / 32.7 | 37.7 / 28.3 / 33.2 | 31.1 / 23.9 / 27.0 | 35.3 / 27.9 / 32.4 | 29.2 / 22.5 / 26.7 | 23.5 / 18.8 / 25.5 | 35.5 / 28.4 / 34.2 |
| | | s14 | 39.4 / 29.3 / 33.5 | 38.3 / 28.3 / 33.5 | 31.0 / 23.9 / 28.7 | 35.3 / 27.6 / 32.1 | 29.2 / 22.6 / 27.9 | 24.4 / 19.3 / 26.9 | 35.0 / 28.1 / 34.3 |
| | mT5$^S$ | s1 | 40.4 / 30.2 / 32.1 | 39.7 / 29.1 / 31.3 | 32.2 / 24.8 / 26.9 | 36.9 / 27.9 / 28.3 | 30.4 / 22.9 / 24.4 | 30.2 / 23.0 / 24.6 | 35.8 / 27.9 / 30.4 |
| | | s6 | 38.4 / 28.6 / 30.6 | 37.9 / 28.1 / 30.9 | 31.1 / 23.9 / 25.2 | 35.0 / 27.6 / 29.2 | 28.7 / 22.3 / 24.6 | 24.3 / 19.3 / 23.1 | 34.5 / 27.7 / 31.2 |
| | | s7 | 38.6 / 28.8 / 31.2 | 38.3 / 28.1 / 30.9 | 30.7 / 23.5 / 25.6 | 34.7 / 27.1 / 29.5 | 28.6 / 21.8 / 24.9 | 25.1 / 19.6 / 23.8 | 35.6 / 28.4 / 30.9 |
| | | s10 | 37.3 / 28.0 / 30.9 | 36.9 / 27.4 / 31.2 | 30.4 / 23.3 / 25.4 | 33.1 / 26.4 / 29.5 | 28.0 / 21.6 / 24.9 | 23.4 / 18.8 / 23.5 | 33.9 / 27.1 / 31.9 |
| | | s14 | 38.8 / 28.9 / 31.5 | 38.3 / 28.4 / 31.5 | 31.2 / 23.8 / 25.7 | 34.9 / 27.4 / 30.1 | 29.2 / 22.3 / 25.2 | 25.9 / 20.4 / 24.6 | 35.3 / 28.3 / 31.8 |
| Base | mmT5 | s1 | 44.5 / 22.4 / 33.5 | 44.2 / 25.9 / 33.3 | 37.3 / 17.4 / 28.8 | 41.0 / 22.1 / 30.9 | 34.3 / 13.5 / 25.6 | 34.8 / 14.4 / 26.1 | 40.1 / 25.5 / 31.9 |
| | | s6 | 43.9 / 21.6 / 32.9 | 44.4 / 26.1 / 33.7 | 36.2 / 16.4 / 27.9 | 42.1 / 23.0 / 32.2 | 35.6 / 14.6 / 27.3 | 34.5 / 14.6 / 26.4 | 42.4 / 28.1 / 34.7 |
| | | s7 | 44.8 / 22.5 / 33.7 | 44.8 / 26.4 / 33.9 | 37.1 / 17.4 / 28.8 | 42.1 / 23.2 / 32.1 | 36.4 / 15.5 / 27.9 | 35.9 / 15.2 / 27.0 | 42.5 / 28.2 / 35.0 |
| | | s10 | 43.6 / 21.2 / 32.7 | 43.8 / 25.6 / 33.2 | 35.2 / 15.8 / 27.0 | 41.8 / 23.1 / 32.4 | 34.9 / 14.1 / 26.7 | 33.6 / 13.8 / 25.5 | 41.8 / 27.4 / 34.2 |
| | | s14 | 44.7 / 22.4 / 33.5 | 44.7 / 26.2 / 33.5 | 37.2 / 17.3 / 28.7 | 42.3 / 23.4 / 32.1 | 36.2 / 15.3 / 27.9 | 35.5 / 15.4 / 26.9 | 42.4 / 27.6 / 34.3 |
| | mT5$^S$ | s1 | 42.9 / 20.7 / 32.1 | 42.5 / 23.2 / 31.3 | 35.0 / 15.6 / 26.9 | 38.2 / 19.2 / 28.3 | 32.5 / 12.0 / 24.4 | 33.0 / 12.8 / 24.6 | 38.6 / 24.0 / 30.4 |
| | | s6 | 41.3 / 18.7 / 30.6 | 41.1 / 22.5 / 30.9 | 32.9 / 13.8 / 25.2 | 37.5 / 18.8 / 29.2 | 32.3 / 12.0 / 24.6 | 30.1 / 11.3 / 23.1 | 38.5 / 24.4 / 31.2 |
| | | s7 | 41.9 / 19.5 / 31.2 | 41.7 / 22.6 / 30.9 | 33.5 / 14.1 / 25.6 | 38.9 / 19.7 / 29.5 | 32.7 / 12.3 / 24.9 | 31.4 / 12.1 / 23.8 | 38.7 / 24.2 / 30.9 |
| | | s10 | 41.7 / 19.1 / 30.9 | 41.5 / 23.0 / 31.2 | 33.1 / 14.0 / 25.4 | 38.6 / 19.7 / 29.5 | 32.7 / 12.2 / 24.9 | 30.6 / 11.7 / 23.5 | 39.3 / 25.1 / 31.9 |
| | | s14 | 42.3 / 19.8 / 31.5 | 42.5 / 23.6 / 31.5 | 33.6 / 14.4 / 25.7 | 39.5 / 20.5 / 30.1 | 33.2 / 12.5 / 25.2 | 32.7 / 12.8 / 24.6 | 39.4 / 25.2 / 31.8 |

Table 20: Validation set results for XL-Sum in the multisource setup for languages Portuguese, Punjabi, Russian, Scottish Gaelic, Serbian, and Sinhala. We report results for the different freezing configurations.

| | lang | somali Rg₁ / Rg₂ / Rg_L | spanish Rg₁ / Rg₂ / Rg_L | swahili Rg₁ / Rg₂ / Rg_L | tamil Rg₁ / Rg₂ / Rg_L | telugu Rg₁ / Rg₂ / Rg_L | thai Rg₁ / Rg₂ / Rg_L | turkish Rg₁ / Rg₂ / Rg_L |
|---|---|---|---|---|---|---|---|---|
| Small · mmT5 | s1 | 38.4 / 28.3 / 30.6 | 31.3 / 23.5 / 27.3 | 38.9 / 30.0 / 33.3 | 31.8 / 26.6 / 31.2 | 29.0 / 23.6 / 27.5 | 31.5 / 24.8 / 28.2 | 33.2 / 27.8 / 32.0 |
| | s6 | 37.0 / 27.4 / 30.8 | 32.2 / 24.2 / 27.1 | 37.7 / 28.8 / 33.7 | 28.9 / 24.2 / 31.3 | 26.8 / 22.1 / 27.9 | 31.0 / 24.8 / 29.3 | 30.3 / 25.3 / 31.5 |
| | s7 | 37.5 / 27.7 / 31.2 | 30.8 / 23.1 / 27.6 | 38.3 / 29.5 / 34.7 | 30.8 / 26.0 / 32.2 | 27.6 / 22.7 / 28.4 | 31.8 / 25.3 / 29.8 | 31.5 / 26.4 / 32.4 |
| | s10 | 36.8 / 27.2 / 30.2 | 31.9 / 23.9 / 26.7 | 37.9 / 28.9 / 33.6 | 28.8 / 24.0 / 31.0 | 26.9 / 22.2 / 27.3 | 31.5 / 25.0 / 29.3 | 30.5 / 25.6 / 31.0 |
| | s14 | 37.6 / 27.6 / 31.0 | 31.1 / 23.3 / 27.5 | 38.3 / 29.4 / 34.3 | 30.4 / 25.6 / 32.1 | 27.7 / 22.6 / 28.2 | 31.8 / 25.1 / 29.6 | 31.2 / 26.1 / 32.1 |
| Small · mT5^S | s1 | 38.0 / 28.0 / 29.0 | 31.7 / 23.9 / 25.6 | 38.5 / 29.7 / 31.7 | 31.9 / 26.7 / 28.7 | 28.5 / 23.2 / 25.3 | 31.1 / 24.4 / 26.2 | 32.9 / 27.5 / 30.2 |
| | s6 | 35.4 / 26.3 / 28.4 | 32.1 / 23.9 / 25.1 | 36.1 / 27.9 / 31.3 | 29.7 / 25.0 / 26.2 | 26.9 / 22.3 / 25.0 | 30.6 / 24.7 / 26.7 | 29.8 / 25.1 / 28.3 |
| | s7 | 35.8 / 26.4 / 28.7 | 30.9 / 23.1 / 25.2 | 36.6 / 28.1 / 31.3 | 29.9 / 25.1 / 28.5 | 26.9 / 21.9 / 25.1 | 30.9 / 24.5 / 26.9 | 30.6 / 25.8 / 29.2 |
| | s10 | 34.6 / 25.8 / 28.7 | 31.6 / 23.6 / 25.6 | 35.3 / 27.0 / 31.5 | 27.5 / 22.7 / 28.3 | 26.0 / 21.6 / 25.3 | 30.4 / 24.2 / 27.0 | 29.2 / 24.5 / 28.8 |
| | s14 | 36.1 / 26.2 / 29.0 | 32.2 / 23.9 / 25.6 | 37.1 / 28.3 / 32.1 | 30.2 / 25.3 / 29.1 | 27.1 / 22.2 / 25.8 | 30.5 / 24.3 / 27.7 | 30.7 / 25.8 / 29.7 |
| Base · mmT5 | s1 | 41.7 / 21.4 / 30.6 | 36.2 / 14.8 / 27.3 | 42.7 / 22.6 / 33.3 | 37.0 / 22.4 / 31.2 | 33.8 / 20.0 / 27.5 | 35.5 / 20.3 / 28.2 | 38.0 / 21.3 / 32.0 |
| | s6 | 41.5 / 21.5 / 30.8 | 36.0 / 14.5 / 27.1 | 43.1 / 23.0 / 33.7 | 37.0 / 22.4 / 31.3 | 33.9 / 20.1 / 27.9 | 36.5 / 21.4 / 29.3 | 37.3 / 20.5 / 31.5 |
| | s7 | 41.7 / 21.7 / 31.2 | 36.5 / 15.1 / 27.6 | 44.2 / 24.0 / 34.7 | 37.9 / 23.4 / 32.2 | 34.4 / 20.7 / 28.4 | 37.4 / 22.0 / 29.8 | 38.3 / 21.6 / 32.4 |
| | s10 | 41.0 / 20.9 / 30.2 | 35.6 / 14.1 / 26.7 | 43.2 / 22.9 / 33.6 | 36.7 / 22.1 / 31.0 | 33.3 / 19.6 / 27.3 | 36.4 / 21.2 / 29.3 | 36.9 / 20.1 / 31.0 |
| | s14 | 41.7 / 21.7 / 31.0 | 36.3 / 14.9 / 27.5 | 43.8 / 23.6 / 34.3 | 37.9 / 23.1 / 32.1 | 34.4 / 20.6 / 28.2 | 37.2 / 21.7 / 29.6 | 38.1 / 21.3 / 32.1 |
| Base · mT5^S | s1 | 39.6 / 19.4 / 29.0 | 34.3 / 13.3 / 25.6 | 41.3 / 21.1 / 31.7 | 34.7 / 20.3 / 28.7 | 31.3 / 17.6 / 25.3 | 33.6 / 18.7 / 26.2 | 36.2 / 19.5 / 30.2 |
| | s6 | 38.5 / 18.2 / 28.4 | 33.6 / 12.2 / 25.1 | 40.4 / 20.2 / 31.3 | 31.5 / 17.8 / 26.2 | 30.6 / 17.0 / 25.0 | 33.5 / 18.6 / 26.7 | 33.9 / 17.3 / 28.3 |
| | s7 | 39.1 / 19.0 / 28.7 | 33.7 / 12.6 / 25.2 | 40.7 / 20.5 / 31.3 | 34.1 / 19.9 / 28.5 | 30.8 / 17.3 / 25.1 | 33.9 / 19.1 / 26.9 | 34.9 / 18.3 / 29.2 |
| | s10 | 38.8 / 18.7 / 28.7 | 34.1 / 12.6 / 25.6 | 40.8 / 20.5 / 31.5 | 33.7 / 19.8 / 28.3 | 30.8 / 17.4 / 25.3 | 33.8 / 18.9 / 27.0 | 34.5 / 18.0 / 28.8 |
| | s14 | 39.3 / 19.1 / 29.0 | 34.2 / 12.8 / 25.6 | 41.2 / 20.9 / 32.1 | 34.7 / 20.6 / 29.1 | 31.5 / 17.9 / 25.8 | 34.7 / 19.6 / 27.7 | 35.4 / 18.8 / 29.7 |

Table 21: Validation set results for XL-Sum in the multisource setup for languages Somali, Spanish, Swahili, Tamil, Telugu, Thai, and Turkish . We report results for the different freezing configurations.

| | lang | ukrainian Rg₁ / Rg₂ / Rg_L | urdu Rg₁ / Rg₂ / Rg_L | uzbek Rg₁ / Rg₂ / Rg_L | vietnamese Rg₁ / Rg₂ / Rg_L | welsh Rg₁ / Rg₂ / Rg_L | yoruba Rg₁ / Rg₂ / Rg_L | *avg* Rg₁ / Rg₂ / Rg_L |
|---|---|---|---|---|---|---|---|---|
| Small · mmT5 | s1 | 32.2 / 25.3 / 29.2 | 42.3 / 33.0 / 37.3 | 28.4 / 23.5 / 26.6 | 44.8 / 31.0 / 34.4 | 41.1 / 31.0 / 34.0 | 45.2 / 33.2 / 35.6 | **36.8 / 18.3 / 28.8** |
| | s6 | 30.1 / 23.6 / 28.6 | 39.5 / 30.6 / 36.8 | 26.8 / 22.1 / 27.1 | 42.9 / 29.9 / 34.1 | 37.4 / 28.3 / 33.5 | 42.9 / 31.8 / 35.8 | 34.8 / 16.5 / 27.5 |
| | s7 | 30.5 / 23.7 / 29.2 | 41.1 / 32.0 / 37.4 | 27.3 / 22.4 / 27.5 | 43.8 / 30.3 / 34.8 | 39.5 / 29.8 / 34.0 | 44.0 / 32.6 / 36.6 | 35.8 / 17.4 / 28.2 |
| | s10 | 30.1 / 23.7 / 28.1 | 39.8 / 30.9 / 36.4 | 26.9 / 22.2 / 26.9 | 43.2 / 30.1 / 34.1 | 37.7 / 28.4 / 33.5 | 42.8 / 31.7 / 35.5 | 35.0 / 16.7 / 27.6 |
| | s14 | 30.5 / 23.8 / 29.0 | 40.5 / 31.5 / 37.3 | 26.8 / 22.0 / 27.4 | 43.7 / 30.3 / 34.4 | 39.3 / 29.9 / 34.1 | 43.5 / 32.3 / 36.2 | 35.6 / 17.2 / 28.0 |
| Small · mT5^S | s1 | 31.7 / 25.0 / 27.3 | 41.6 / 32.4 / 35.3 | 28.2 / 23.1 / 24.4 | 44.8 / 30.8 / 33.0 | 40.4 / 30.4 / 31.8 | 44.3 / 32.7 / 34.2 | 36.6 / 18.0 / 28.6 |
| | s6 | 29.7 / 23.2 / 25.8 | 39.4 / 30.4 / 33.7 | 26.9 / 22.3 / 24.6 | 42.8 / 29.7 / 31.9 | 37.7 / 28.4 / 30.0 | 41.8 / 31.0 / 33.5 | 34.5 / 16.2 / 27.2 |
| | s7 | 29.9 / 23.3 / 26.0 | 39.9 / 30.9 / 34.4 | 27.7 / 23.0 / 24.5 | 43.1 / 29.6 / 32.1 | 37.8 / 28.4 / 30.5 | 43.2 / 31.6 / 33.3 | 34.9 / 16.5 / 27.4 |
| | s10 | 29.1 / 22.9 / 26.1 | 38.5 / 29.6 / 34.1 | 26.8 / 22.1 / 25.1 | 42.1 / 29.1 / 32.2 | 36.7 / 27.7 / 30.9 | 40.8 / 30.1 / 33.5 | 33.6 / 15.5 / 26.4 |
| | s14 | 29.9 / 23.3 / 26.5 | 40.2 / 31.1 / 34.9 | 27.4 / 22.4 / 25.2 | 43.2 / 29.9 / 32.7 | 37.7 / 28.6 / 30.7 | 42.9 / 31.5 / 33.5 | 35.0 / 16.6 / 27.5 |
| Base · mmT5 | s1 | 36.8 / 18.0 / 29.2 | 47.0 / 27.8 / 37.3 | 33.0 / 16.5 / 26.6 | 49.1 / 27.6 / 34.4 | 44.9 / 24.8 / 34.0 | 48.0 / 25.4 / 35.6 | 41.2 / 22.4 / 32.4 |
| | s6 | 36.1 / 17.2 / 28.6 | 46.4 / 27.2 / 36.8 | 33.2 / 16.9 / 27.1 | 48.5 / 27.0 / 34.1 | 43.9 / 23.7 / 33.5 | 47.8 / 25.8 / 35.8 | 41.2 / 22.4 / 32.6 |
| | s7 | 36.8 / 17.9 / 29.2 | 46.9 / 27.8 / 37.4 | 33.9 / 17.4 / 27.5 | 49.1 / 27.8 / 34.8 | 44.8 / 24.5 / 34.0 | 48.3 / 26.2 / 36.6 | **41.9 / 23.1 / 33.2** |
| | s10 | 35.6 / 16.7 / 28.1 | 46.0 / 26.8 / 36.4 | 32.8 / 16.8 / 26.9 | 48.4 / 27.0 / 34.1 | 44.2 / 23.8 / 33.5 | 47.4 / 25.2 / 35.5 | 40.8 / 22.1 / 32.3 |
| | s14 | 36.7 / 17.7 / 29.0 | 46.9 / 27.8 / 37.3 | 34.0 / 17.4 / 27.4 | 49.0 / 27.6 / 34.4 | 45.0 / 24.6 / 34.1 | 48.5 / 26.4 / 36.2 | 41.8 / 23.0 / 33.1 |
| Base · mT5^S | s1 | 34.7 / 16.1 / 27.3 | 45.1 / 25.6 / 35.3 | 30.5 / 14.3 / 24.4 | 47.8 / 25.9 / 33.0 | 42.4 / 22.1 / 31.8 | 46.7 / 24.1 / 34.2 | 39.2 / 20.4 / 30.6 |
| | s6 | 32.9 / 14.3 / 25.8 | 43.1 / 23.7 / 33.7 | 29.9 / 14.4 / 23.9 | 46.1 / 24.2 / 31.9 | 40.2 / 19.4 / 30.0 | 45.2 / 22.9 / 33.5 | 37.9 / 19.2 / 29.8 |
| | s7 | 33.3 / 14.5 / 26.0 | 43.8 / 24.4 / 34.4 | 30.1 / 14.5 / 24.5 | 46.6 / 24.7 / 32.1 | 40.5 / 19.9 / 30.5 | 45.1 / 22.8 / 33.3 | 38.5 / 19.8 / 30.2 |
| | s10 | 33.2 / 14.6 / 26.1 | 43.6 / 24.2 / 34.1 | 30.5 / 15.0 / 25.1 | 46.3 / 24.6 / 32.2 | 41.2 / 20.4 / 30.9 | 45.1 / 23.2 / 33.5 | 38.4 / 19.8 / 30.2 |
| | s14 | 33.9 / 15.1 / 26.5 | 44.6 / 25.2 / 34.9 | 30.9 / 15.2 / 25.2 | 47.1 / 25.2 / 32.7 | 41.2 / 20.4 / 30.7 | 45.6 / 23.0 / 33.5 | 39.1 / 20.4 / 30.7 |

Table 22: Validation set results for XL-Sum in the multisource setup for languages Ukrainian, Urdu, Uzbek, Vietnamese, Welsh, and Yoruba. We report results for the different freezing configurations.

| | cfg | ar F1 / EM | bn F1 / EM | en F1 / EM | fi F1 / EM | id F1 / EM | ko F1 / EM | ru F1 / EM | sw F1 / EM | te F1 / EM | avg F1 / EM |
|---|---|---|---|---|---|---|---|---|---|---|---|
| Small · mmT5 | s1 | 51.6 / 34.9 | 30.4 / 16.8 | 62.7 / 51.4 | 46.9 / 32.1 | 49.9 / 33.6 | 26.6 / 19.6 | 46.3 / 28.6 | 34.4 / 23.8 | 31.5 / 22.1 | 42.3 / 29.2 |
| | s6 | 57.5 / 39.4 | 39.5 / 24.8 | 68.3 / 58.2 | 51.7 / 35.2 | 54.3 / 40.0 | 26.5 / 15.9 | 58.2 / 36.5 | 44.8 / 32.5 | 43.5 / 29.6 | 49.4 / 34.7 |
| | s7 | 63.2 / 46.0 | 40.6 / 26.5 | 70.7 / 60.5 | 59.3 / 41.9 | 57.9 / 43.4 | 30.7 / 18.5 | 60.3 / 39.9 | 36.4 / 25.7 | 38.1 / 23.9 | 50.8 / 36.3 |
| | s10 | 39.7 / 56.8 | 25.7 / 40.6 | 57.7 / 69.4 | 34.7 / 49.1 | 40.2 / 53.6 | 17.0 / 26.9 | 34.7 / 57.8 | 31.5 / 45.9 | 28.7 / 39.5 | 34.4 / 48.7 |
| | s14 | 63.1 / 47.6 | 40.5 / 25.7 | 71.0 / 60.2 | 58.1 / 40.3 | 56.1 / 41.1 | 30.4 / 18.1 | 59.3 / 37.9 | 40.9 / 29.9 | 33.9 / 22.9 | 50.4 / 36.0 |
| Small · mT5^S | s1 | 45.6 / 30.2 | 25.5 / 14.2 | 61.7 / 50.0 | 45.1 / 29.9 | 49.7 / 33.8 | 27.1 / 20.3 | 50.8 / 35.0 | 33.1 / 21.8 | 18.5 / 13.6 | 39.7 / 27.6 |
| | s6 | 47.3 / 31.6 | 30.0 / 18.6 | 65.9 / 54.3 | 51.4 / 35.7 | 52.1 / 35.2 | 26.9 / 18.1 | 57.1 / 40.3 | 40.1 / 28.1 | 19.0 / 13.8 | 43.3 / 30.6 |
| | s7 | 51.4 / 31.4 | 31.3 / 18.6 | 68.4 / 57.0 | 52.4 / 33.5 | 54.4 / 36.1 | 27.7 / 19.2 | 54.7 / 35.5 | 33.4 / 23.6 | 20.1 / 14.6 | 43.8 / 30.0 |
| | s10 | 46.5 / 30.7 | 30.8 / 19.5 | 66.8 / 55.7 | 51.2 / 35.5 | 50.8 / 35.9 | 28.3 / 20.3 | 55.6 / 38.1 | 43.1 / 29.5 | 19.9 / 14.9 | 43.7 / 31.1 |
| | s14 | 50.4 / 33.7 | 26.6 / 15.9 | 69.3 / 58.6 | 55.5 / 37.2 | 56.6 / 39.5 | 28.3 / 18.8 | 59.9 / 40.8 | 36.2 / 23.0 | 17.3 / 12.4 | 44.5 / 31.1 |
| Base · mmT5 | s1 | 65.8 / 45.5 | 51.3 / 32.7 | 74.1 / 63.4 | 65.2 / 50.9 | 69.2 / 50.1 | 54.2 / 44.2 | 55.3 / 32.5 | 61.8 / 44.3 | 53.0 / 37.2 | 61.1 / 44.5 |
| | s6 | 75.2 / 59.2 | 57.9 / 38.1 | 76.5 / 65.7 | 71.6 / 57.4 | 76.8 / 62.3 | 54.6 / 42.0 | 67.4 / 45.7 | 69.8 / 52.1 | 54.7 / 39.8 | 67.2 / 51.4 |
| | s7 | 75.2 / 59.8 | 59.7 / 38.1 | 77.5 / 67.7 | 73.3 / 59.0 | 77.4 / 61.6 | 59.2 / 48.9 | 67.5 / 45.3 | 69.6 / 53.7 | 61.6 / 45.1 | 69.0 / 53.2 |
| | s10 | 74.3 / 57.0 | 56.9 / 37.2 | 75.7 / 64.1 | 72.1 / 58.2 | 77.0 / 62.1 | 56.0 / 43.8 | 66.1 / 43.7 | 69.3 / 51.9 | 52.1 / 38.0 | 66.7 / 50.7 |
| | s14 | 73.7 / 58.0 | 59.2 / 38.1 | 77.0 / 67.3 | 72.6 / 58.4 | 76.5 / 61.4 | 58.9 / 48.9 | 68.1 / 45.1 | 68.5 / 50.3 | 61.6 / 45.0 | 68.5 / 52.5 |
| Base · mT5^S | s1 | 64.1 / 40.7 | 41.4 / 25.7 | 72.5 / 61.1 | 64.1 / 48.0 | 70.3 / 56.5 | 42.1 / 30.1 | 58.4 / 36.5 | 58.2 / 41.3 | 48.7 / 39.2 | 57.8 / 42.1 |
| | s6 | 66.5 / 44.8 | 42.8 / 27.4 | 74.8 / 62.7 | 67.4 / 50.6 | 75.3 / 60.7 | 53.9 / 40.9 | 65.0 / 43.6 | 59.7 / 42.9 | 55.0 / 42.0 | 62.3 / 46.2 |
| | s7 | 70.2 / 47.7 | 52.7 / 32.7 | 74.8 / 64.3 | 68.5 / 52.9 | 74.3 / 58.9 | 49.8 / 39.9 | 63.6 / 39.7 | 58.4 / 40.1 | 56.8 / 46.2 | 63.2 / 46.9 |
| | s10 | 67.5 / 46.9 | 48.5 / 30.1 | 74.2 / 63.9 | 67.0 / 51.0 | 73.5 / 57.7 | 47.7 / 35.1 | 64.7 / 42.5 | 57.9 / 42.3 | 53.3 / 42.6 | 61.6 / 45.8 |
| | s14 | 68.3 / 45.8 | 55.4 / 38.1 | 75.7 / 64.1 | 67.8 / 52.2 | 75.3 / 60.4 | 52.8 / 40.9 | 63.7 / 40.6 | 61.4 / 43.1 | 55.8 / 44.7 | 64.0 / 47.8 |

Table 23: Results for the validation set of TyDiQA. We report results for the different configurations of freezing.

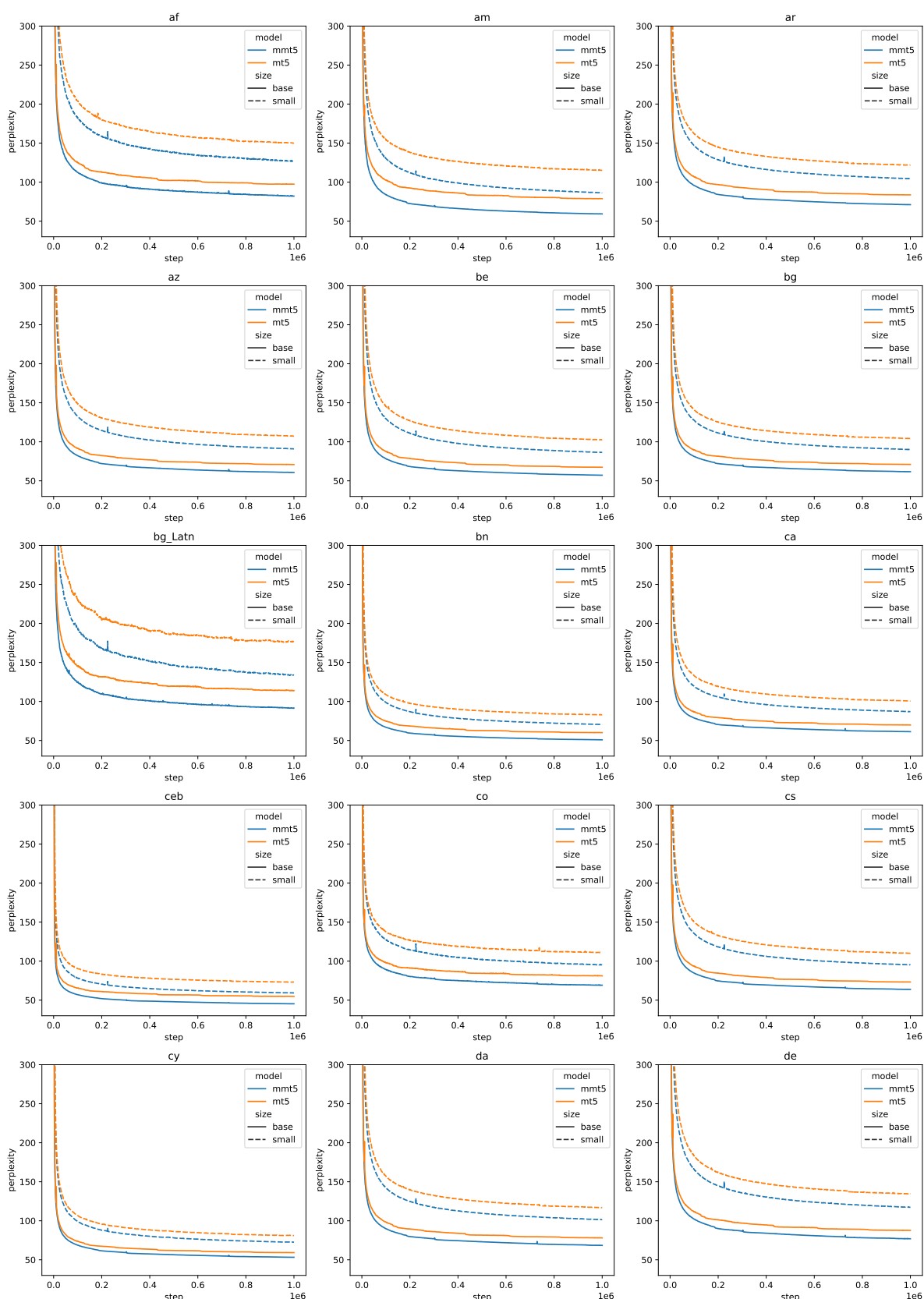

Figure 8: Per language perplexity of different model sizes for languages af-de.

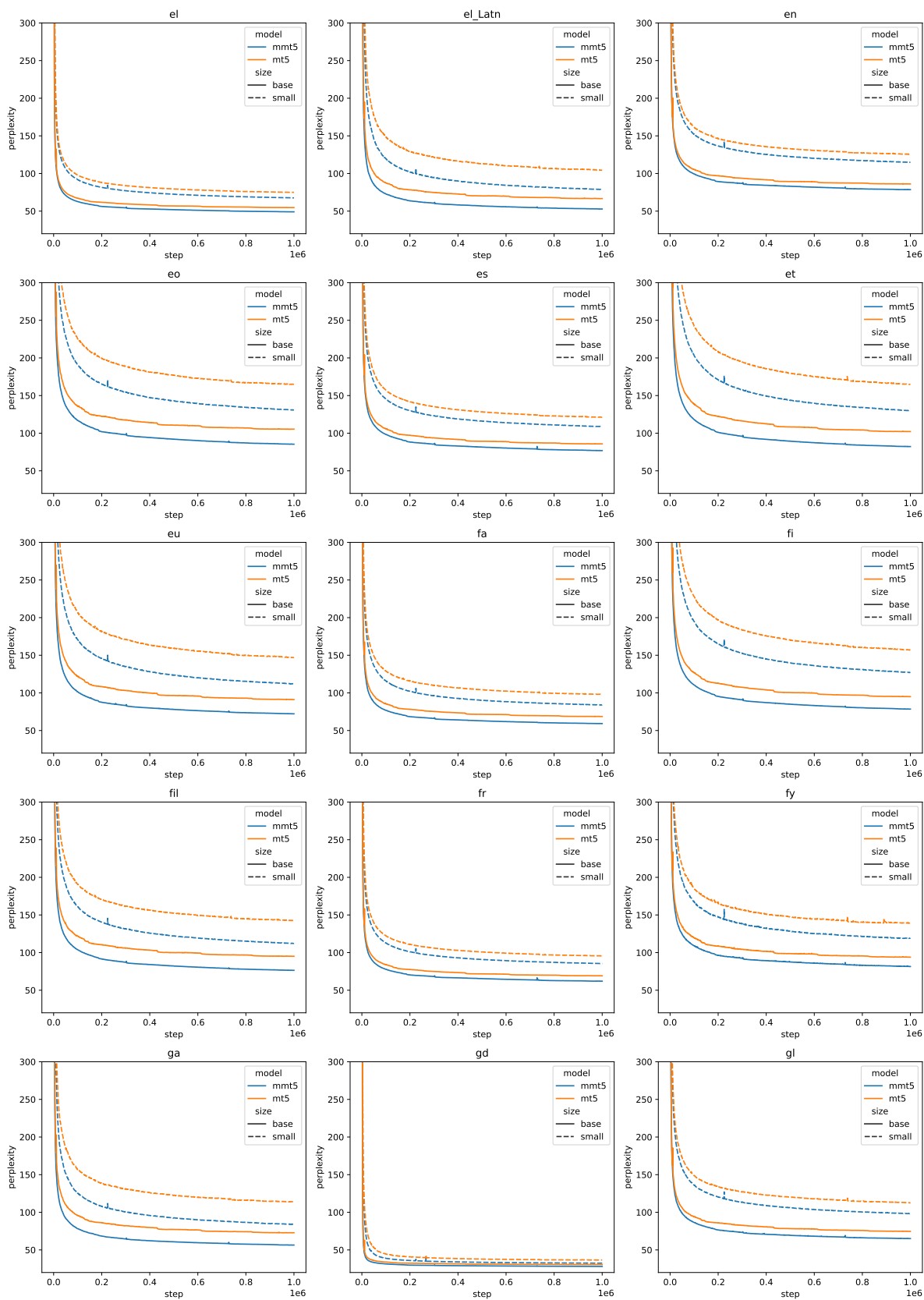

Figure 9: Per language perplexity of different model sizes for languages el-gl.

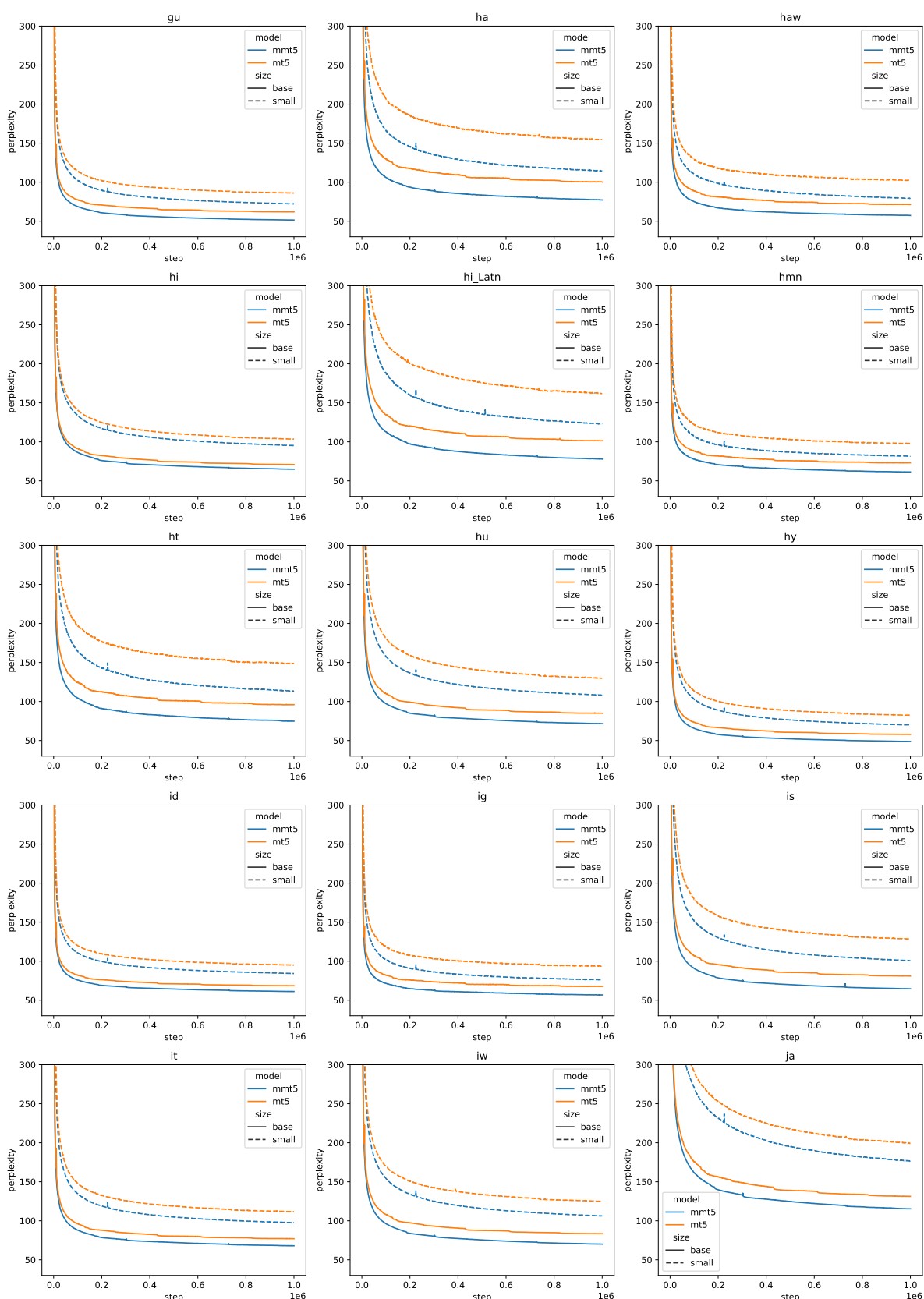

Figure 10: Per language perplexity of different model sizes for languages gu-ja.

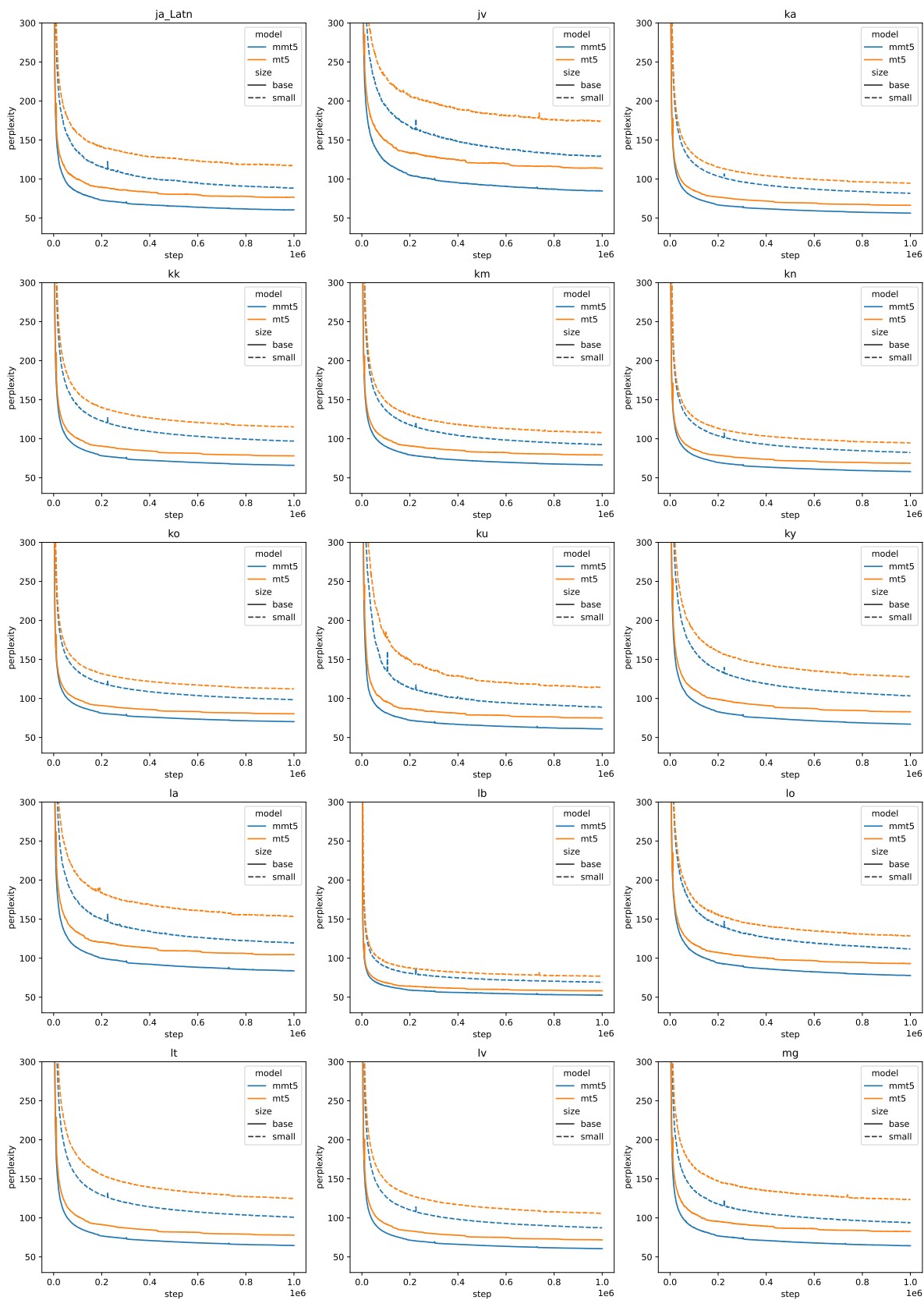

Figure 11: Per language perplexity of different model sizes for languages ja-mg.

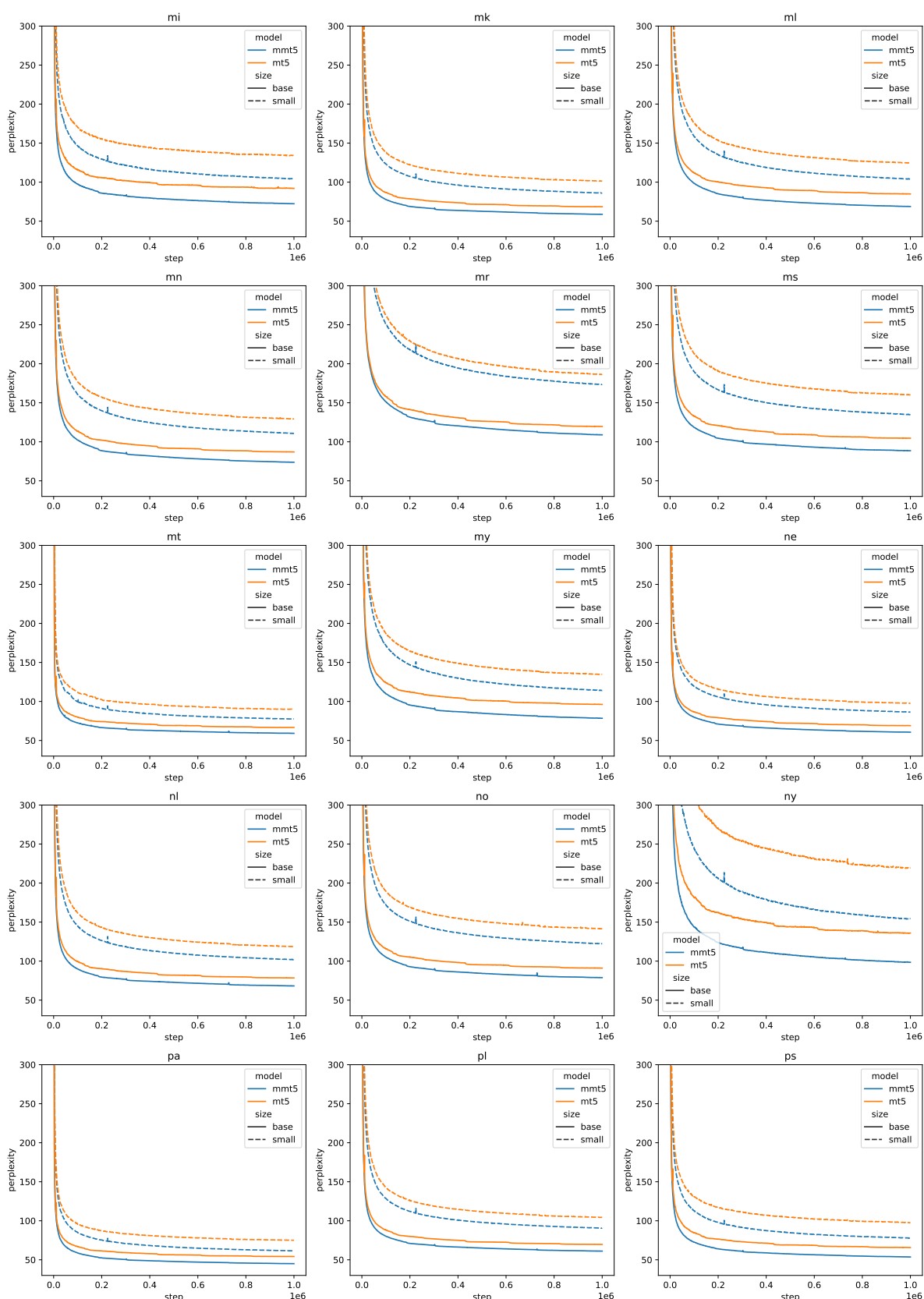

Figure 12: Per language perplexity of different model sizes for languages mi-gs.

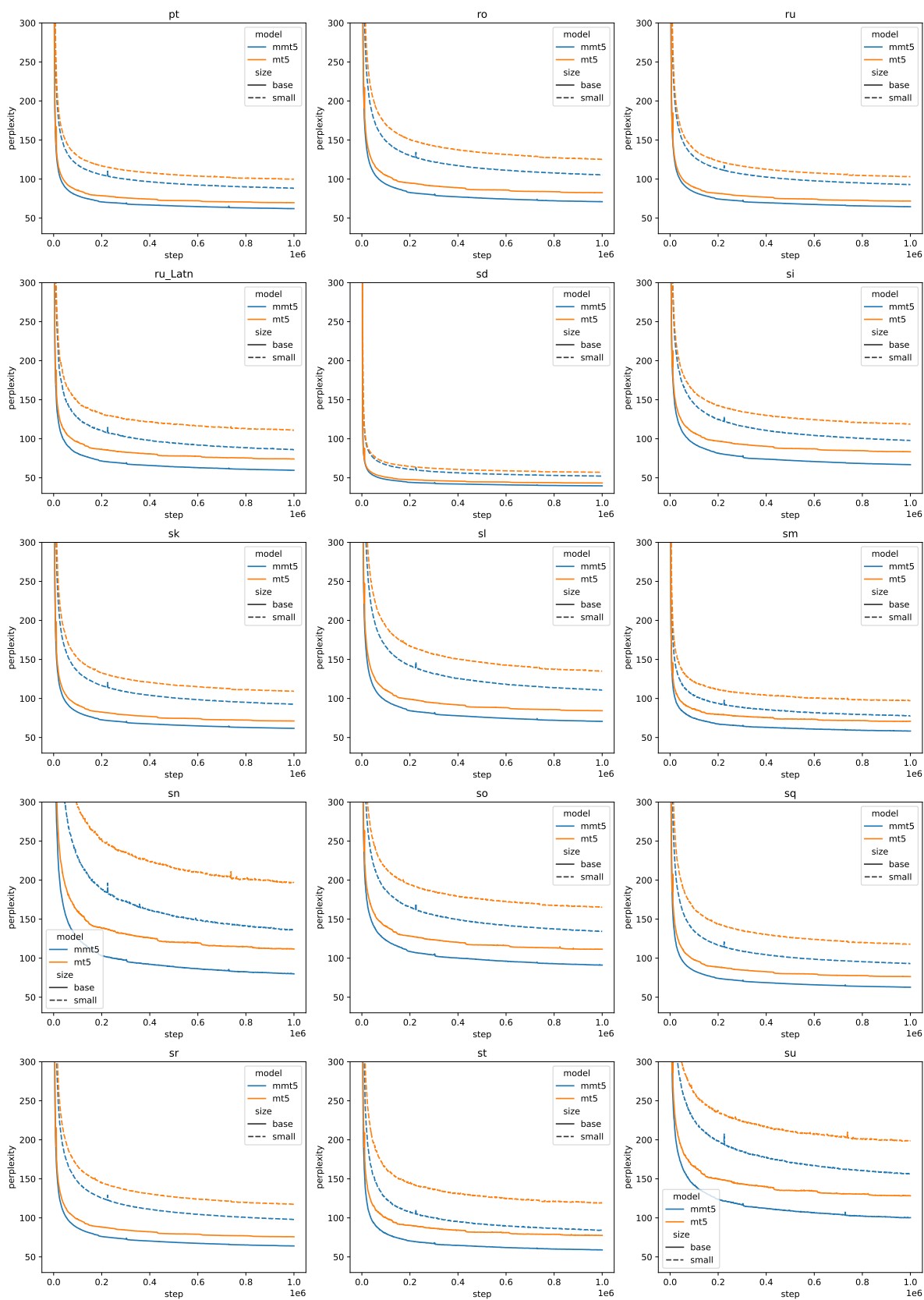

Figure 13: Per language perplexity of different model sizes for languages pt-sg.

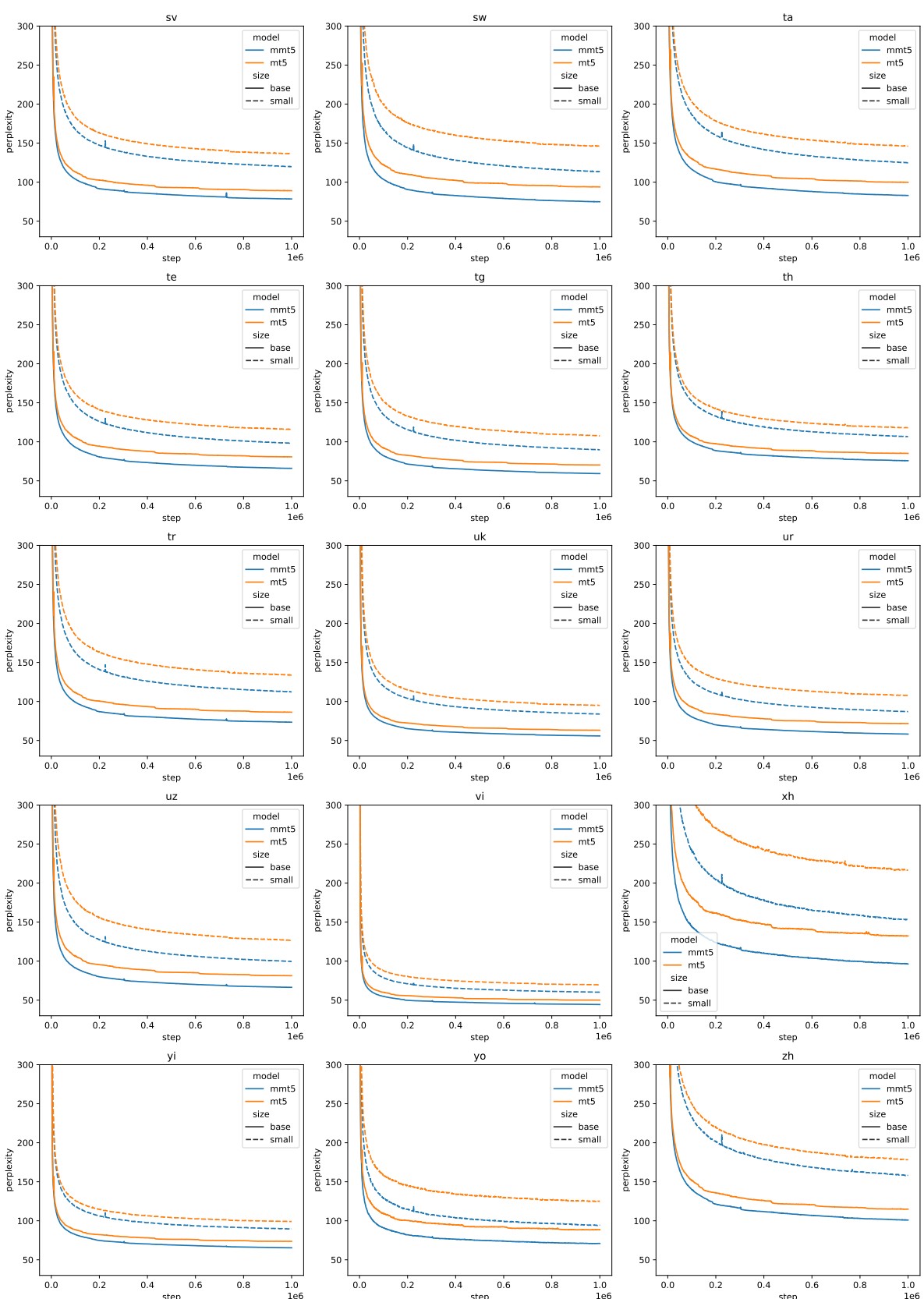

Figure 14: Per language perplexity of different model sizes for languages sv-zh.