# OpenReview forum: "mmT5: Modular Multilingual Pre-Training Solves Source Language Hallucinations"
_EMNLP/2023/Conference — EMNLP 2023 Findings_

### Official Review · Reviewer_Uj9i · 2023-08-03

**Soundness:** 4

**Excitement:**

4: Strong: This paper deepens the understanding of some phenomenon or lowers the barriers to an existing research direction.

**Missing References:**

You might want to cite the following work in your section on similar methods in machine translation:

Share or Not? Learning to Schedule Language-Specific Capacity for Multilingual Translation (Zhang et al., ICLR 2021)

Learning Language-Specific Layers for Multilingual Machine Translation (Pires et al., ACL 2023)

**Paper Topic And Main Contributions:**

This paper proposes mmT5: adding language specific components to multilingual language pre-training and freezing the decoder in cross-lingual transfer fine-tuning. Comprehensive experiments show that their method outperform standard training by a large margin in classification tasks. Moreover, their method solves the problem of the model generating incorrect language in generation tasks for small models.

**Questions For The Authors:**

A. Why do you think that freezing part of the parameters in the decoder brings gains? If it is because the model does not forget what it has learned during the pre-train process, would methods that mitigate catastrophic forgetting also helps (e.g. EWC)?

B. The added components sounds like a mixture-of-experts model to me, what is the difference between your method and a mixture-of-experts model where each language have its own expert.

C. What are the hyper-parameters for your generation setting? I found that the decoding strategy can have significant influence on the generation quality. I cannot seem to find it in the paper thus I have assigned a lower score on the reproducibility part.

**Reasons To Accept:**

1. The method is simple and elegant. The experiment results are much better than the baseline at the lost of a few million parameters which is negligible compare to a few hundred million parameters of the pre-trained model.

2. The authors conducted experiments on a wide range of tasks spanning Question Answering and generation, and conducted detailed ablation studies on which component to freeze during fine-tuning. Their findings provide insight to further research and practical settings on adapting language models to languages with limited task-specific training data.

3. The paper is well written with clear notations and self-contained figures and tables.

**Reasons To Reject:**

Limited Novelty: the paper extends the method in [1], which is on an encoder-only model to an encoder-decoder model. The main benefits come from the previous method. Although the authors also conducted decoder freezing, the idea of freezing part of the parameter mitigates the incorrect language problem is also studied in [2]. Therefore the paper offers limited contribution comparing to adding the findings of these two paper together. I wish the author could expand a bit more on section 6.1, where they studied freezing different parts of the model and provide insights on **why** does freezing the FFN component of the decoder brings the largest gains.

[1] Lifting the Curse of Multilinguality by Pre-training Modular Transformers (Pfeiffer et al., NAACL 2022)

[2] Overcoming Catastrophic Forgetting in Zero-Shot Cross-Lingual Generation (Vu et al., EMNLP 2022)

**Reproducibility:**

4: Could mostly reproduce the results, but there may be some variation because of sample variance or minor variations in their interpretation of the protocol or method.

**Reviewer Confidence:**

4: Quite sure. I tried to check the important points carefully. It's unlikely, though conceivable, that I missed something that should affect my ratings.

**Typos Grammar Style And Presentation Improvements:**

Figure 4 would be better if it is a table or a bar chart since there are only two sizes.

---

> ### Author Rebuttal · Authors · 2023-08-28
>
> We extend our heartfelt gratitude to the reviewers for their thoughtful engagement with our work and their valuable feedback. We have carefully considered each comment and question.
>
> Before addressing the individual questions, it is essential to emphasize that our primary contribution lies in the freezing strategy applied to modular networks to effectively combat the challenge of generating text in the incorrect language. The pronounced positive impact of this strategy on the modular model, while having a neutral effect on dense models, is a testament to the delicate interplay between these two factors: modularity and training regime of frozen components.
>
> To your questions and concerns:
>
> > [1], which is on an encoder-only model to an encoder-decoder model. The main benefits come from the previous method. Although the authors also conducted decoder freezing, the idea of freezing part of the parameter mitigates the incorrect language problem is also studied in [2]. Therefore the paper offers limited contribution comparing to adding the findings of these two paper together.
>
> We respectfully disagree that our contribution is a mere combination of the findings from these two papers. Our contributions extend beyond the individual concepts discussed in those works and present a unique approach for mitigating the source language hallucination problem within modular multilingual pretrained encoder-decoder models.
>
> Regarding the concept of freezing parts of the model, [2] primarily focuses on prompt tuning and its parameter-efficient approach for freezing parts of the model. However, our work goes beyond by demonstrating a specific freezing strategy that positively affects modular models while having a neutral effect on dense models. Our comprehensive experiments emphasize the importance of freezing the right components of the decoder to achieve optimal zero-shot cross-lingual generation, which aligns with the distinct challenges of our proposed architecture.
> In particular, our approach pays special attention to achieving near-perfect target language generation, which is a key focus in our XLSUM experiments. Our method substantially enhances the ability of our model to generate text in the correct language under zero-shot settings, as demonstrated in our results. This outcome stands in contrast to the issues highlighted in [2] where they mention in their paper that "we discover that both MODELTUNING and PROMPTTUNING often partially summarize non English articles into English instead of the target language." [2].
>
> Finally, in the study conducted by [2], it was observed that their parameter-efficient prompt-tuning approach led to a decline in performance in the source language (English) when compared to configurations where all parameters were fine-tuned. In contrast, our proposed freezing strategy yields performance enhancements across all languages, including the source language.
>
> >  I wish the author could expand a bit more on section 6.1, where they studied freezing different parts of the model and provide insights on why does freezing the FFN component of the decoder brings the largest gains
>
> The choice to freeze the FFN component of the decoder stems from its functional significance within the model architecture. This component houses the majority of parameters and is responsible for transforming the intermediate attention outputs into the final model outputs, often associated with generating language. Given its substantial parameter count, it is plausible that the FFN component encapsulates the model's language generative capabilities.
>
> Our observation of freezing the FFN component bringing the largest gains aligns with our hypothesis that during fine-tuning, models tend to overfit to generating text in the source language they were fine-tuned on. By freezing the FFN component, we effectively prevent the model from over-fitting on generating in the fine-tuning source language, allowing it to retain its capacity for cross-lingual generation. This hypothesis is supported by the fact that only fine-tuning the attention component forces the model to focus more on the input text, aiding in mitigating source language hallucination.
>
> Interestingly, our experiments indicate that this freezing strategy positively affects the modular model while having a neutral effect on the dense model. One potential interpretation of this outcome is that, during fine-tuning, the modular model might learn to disregard the language-specific modules when the FFN component is fine-tuned. This dynamic suggests that the modularity of our approach plays a role in the observed behavior.
>
> The differential impacts of freezing strategies on modular and dense models point to an intriguing area for future research. Investigating the interplay between the components, the modularity of the architecture, and the specific fine-tuning dynamics could offer deeper insights into the mechanisms that drive our observed improvements.
>
> > A. Why do you think that freezing part of the parameters in the decoder brings gains? If it is because the model does not forget what it has learned during the pre-train process, would methods that mitigate catastrophic forgetting also helps (e.g. EWC)?
>
> Your point is well-taken. The benefits we observe from freezing parts of the decoder could indeed relate to mitigating catastrophic forgetting, as you've rightly indicated. However, we would like to highlight that the intention behind our freezing strategy is more nuanced. Our approach aims to prevent the model from overfitting during zero-shot cross-lingual fine-tuning, particularly to generate text in the target language. While there might be some conceptual overlap between the idea of catastrophic forgetting and overfitting, our strategy focuses on addressing the specific challenges posed by the source language hallucination problem in natural language generation tasks.
> The suggestion to explore methods like Elastic Weight Consolidation (EWC) is indeed intriguing. EWC, by design, aims to prevent the model from forgetting previously learned tasks. However, it's important to consider the complexity of our setup, where the model would need to manage a multitude of "tasks" (i.e., 100s of languages) learned during pre-training. Retaining all aspects of the pre-training task (such as language modeling) might be complicated for EWC and not actually necessary for effective zero-shot cross-lingual generation. The challenge lies in determining which aspects to retain and how to best adapt them to the zero-shot fine-tuning context.
> We hypothesize that the positive effects of freezing the decoder stem from preventing the model from being biased toward generating in the fine-tuning language. By doing so, we ensure that the model's generative capacity remains versatile and adaptable to multiple languages, reducing the likelihood of hallucinations.
>
> > B. The added components sounds like a mixture-of-experts model to me, what is the difference between your method and a mixture-of-experts model where each language have its own expert.
>
> We appreciate your insightful comparison between our method and a mixture-of-experts (MoE) model with language-specific experts. While both approaches incorporate the concept of modularization, it is essential to clarify the distinct characteristics that set our method apart:
> The core distinction between our method and a MoE model lies in the routing strategy (for more details see [5; Section 4]). In our approach, we adopt a fixed routing regime, where each module is associated with a specific language and selected a priori during inference. This fixed selection prevents overfitting issues and provides clear interpretability in selecting the appropriate module for each language.
> In contrast, MoE models typically involve a learned routing process, where the model dynamically selects modules based on input data. This learned routing approach, while conceptually appealing, presents several challenges highlighted in [5; Section 4.2.1]. These challenges include "training instabilities," "module collapse," and "overfitting," which can lead to suboptimal performance and architectural inefficiencies.
>
> Our fixed routing strategy not only mitigates the challenges associated with learned routing but also offers a significant advantage in terms of interpretability and efficiency. During inference, we already know which modules to activate based on the language being processed. This allows for a streamlined and efficient inference process, as only the relevant modules are engaged.
> In contrast, a learned routing MoE-style setup could potentially involve the activation of multiple modules for each input, leading to higher computational demands. Additionally, the interpretability of a MoE-style model could be diminished, as the selection of modules would depend on learned factors that might be less transparent than our fixed routing approach.
>
> > C. What are the hyper-parameters for your generation setting? I found that the decoding strategy can have significant influence on the generation quality. I cannot seem to find it in the paper thus I have assigned a lower score on the reproducibility part.
>
> We perform greedy decoding.
>
> > You might want to cite the following work in your section on similar methods in machine translation:
> Share or Not? Learning to Schedule Language-Specific Capacity for Multilingual Translation (Zhang et al., ICLR 2021)
> Learning Language-Specific Layers for Multilingual Machine Translation (Pires et al., ACL 2023)
>
> Thanks for the pointers, we will make sure to include them in the updated version of our paper.
>
> > Figure 4 would be better if it is a table or a bar chart since there are only two sizes.
>
> Thank you for the suggestion, we will change the plot to a bar chart.
>
>
>
>
>
> [1] Lifting the Curse of Multilinguality by Pre-training Modular Transformers (Pfeiffer et al., NAACL 2022)
>
> [2] Overcoming Catastrophic Forgetting in Zero-Shot Cross-Lingual Generation (Vu et al., EMNLP 2022)
>
> [5] Modular Deep Learning (Pfeiffer et al., arXiv 2023)

---

### Official Review · Reviewer_14xE · 2023-08-03

**Typos Grammar Style And Presentation Improvements:** N/A
**Soundness:** 3

**Excitement:**

3: Ambivalent: It has merits (e.g., it reports state-of-the-art results, the idea is nice), but there are key weaknesses (e.g., it describes incremental work), and it can significantly benefit from another round of revision. However, I won't object to accepting it if my co-reviewers champion it.

**Missing References:**

N/A

**Paper Topic And Main Contributions:**

This paper enhances mT5 with language adapters. It then conducts experiments on zero-shot cross-lingual transfer and multilingual tasks for both NLU and NLG tasks. Experiments show that the model outperforms mT5 and performs very good on resolving the problem of source language hallucination.

**Questions For The Authors:**

Line 251-253 "For each dataset, we select the best model checkpoint based on performance on the validation set."  The validation set uses examples in the same language as the training dataset or test dataset under the zero-shot cross-lingual transfer setting?

**Reasons To Accept:**

1 The performance is good compared with mT5.

2 The paper is well written.

**Reasons To Reject:**

1 The technical contribution is incremental. The model architecture is very similar to that of "Multilingual unsupervised neural machine translation with denoising adapters. " However, the authors fail to talk about the relations and differences. However, it is interesting to see the performance of such model on NLU and other NLG tasks.

2 It is also recommended to compare the performance with "Multilingual unsupervised neural machine translation with denoising adapters. " and "MAD-X: An Adapter-Based Framework for Multi-Task Cross-Lingual Transfer", which trains adapters on top of a well trained multilingual pretrained model.



**Reproducibility:**

4: Could mostly reproduce the results, but there may be some variation because of sample variance or minor variations in their interpretation of the protocol or method.

**Reviewer Confidence:**

4: Quite sure. I tried to check the important points carefully. It's unlikely, though conceivable, that I missed something that should affect my ratings.

---

> ### Author Rebuttal · Authors · 2023-08-28
>
> We extend our heartfelt gratitude to the reviewers for their thoughtful engagement with our work and their valuable feedback. We have carefully considered each comment and question.
>
> Before addressing the individual questions, it is essential to emphasize that our primary contribution lies in the freezing strategy applied to modular networks to effectively combat the challenge of generating text in the incorrect language. The pronounced positive impact of this strategy on the modular model, while having a neutral effect on dense models, is a testament to the delicate interplay between these two factors: modularity and training regime of frozen components.
>
> To your questions and concerns:
>
> > “The technical contribution is incremental. The model architecture is very similar to that of "Multilingual unsupervised neural machine translation with denoising adapters. " [6]  However, the authors fail to talk about the relations and differences. “
>
> While it's true that there are architectural similarities between our work and the referenced study [6], our primary contribution is not centered around introducing a new architecture. Instead, our central focus is on addressing the source language hallucination problem within multilingual pre-trained models, particularly in the context of zero-shot cross-lingual sequence-to-sequence tasks. The core problem we address is fundamentally different from the work referenced. We introduce modular language-specific modules to mitigate the hallucination problem and propose novel strategies for freezing parts of the decoder to enhance zero-shot transfer, while the referenced work focuses on adapting dense multilingual models for zero-shot machine translation.The differences in problem formulation, methodology, and application contexts result in unique challenges and solutions in our work. This combination of modularity and freezing methods sets our contribution apart and provides substantial value to the field.
>
> However, we appreciate the reviewer's point regarding discussing the relationships and differences more explicitly in the related work section. We intend to revise and expand our discussion of related work to provide a more comprehensive comparison between our approach and the referenced work. This will enable readers to better understand the distinct contributions and contexts of the two studies.
>
> > “ It is also recommended to compare the performance with "Multilingual unsupervised neural machine translation with denoising adapters. "[6] and "MAD-X: An Adapter-Based Framework for Multi-Task Cross-Lingual Transfer"[3], which trains adapters on top of a well trained multilingual pretrained model.”
>
> We want to emphasize that our work is rooted in addressing the source language hallucination problem within multilingual pretrained models, particularly in the context of sequence-to-sequence tasks. [X-Mod; 1] empirically demonstrated the concept of models being "cursed by multilinguality" if not pre-trained modularly from the outset. Their experiments revealed that adding adapter modules post-hoc to a non-modular dense model (following the [MAD-X; 3] method) underperforms compared to training a multilingual model modularly from scratch.  Additionally, the findings in [X-Mod; 1] indicated that stacking adapters underperforms fine-tuning the shared parameters. We observed similar trends in our initial experiments with mmT5. This, combined with our compute budget, led us to discontinue pure adapter-based experiments. The decision was driven by a combination of factors, including the modular architecture's effectiveness in tackling hallucination issues and our focus on fine-tuning shared parameters, which we found to yield better results.
>
> Furthermore, we would like to underscore that the setup of machine translation (as in [6]) is significantly different from the context we address in our paper. Our main contribution centers around the source language hallucination problem in natural language generation tasks, where the input and output languages are the same. While the methods we propose may have applicability in machine translation tasks, the distinct task formulations warrant careful considerations and specific experimental setups, which we leave for future work.
>
> > “Line 251-253 "For each dataset, we select the best model checkpoint based on performance on the validation set." The validation set uses examples in the same language as the training dataset or test dataset under the zero-shot cross-lingual transfer setting?”
>
> In our experimental setup, we adopt a strategy similar to previous studies in the field. For example, for the XNLI dataset, we evaluate our models on the validation set of all target languages and select the best checkpoint based on their averaged performance. For the XQuAD dataset, we evaluate using only the available validation data in English, based on which the best checkpoint is selected.
> Our approach to model checkpoint selection is consistent with strategies used in prior multilingual studies, which helps ensure comparability and consistency with existing research. We believe that this approach reflects a reasonable and widely accepted methodology for evaluating model performance across diverse languages.
>
> However, we want to highlight that despite averaging over the validation sets of target languages for some tasks, the checkpoint that consistently yielded the best results across languages was the same as when the checkpoint selection is only conducted on validation sets of the English language. This suggests that, even under the zero-shot cross-lingual transfer setting, the best performance on the English validation set generally aligns with the best performance averaged over all languages.
>
> [1] Lifting the Curse of Multilinguality by Pre-training Modular Transformers (Pfeiffer et al., NAACL 2022)
>
> [3] MAD-X: An Adapter-Based Framework for Multi-Task Cross-Lingual Transfer (Pfeiffer et al., EMNLP 2020)
>
> [6] Multilingual Unsupervised Neural Machine Translation with Denoising Adapters (Üstün et al., EMNLP 2021)

---

### Official Review · Reviewer_QXjY · 2023-08-06

**Soundness:** 3

**Excitement:**

2: Mediocre: This paper makes marginal contributions (vs non-contemporaneous work), so I would rather not see it in the conference.

**Paper Topic And Main Contributions:**

This paper proposes a modular multilingual sequence-to-sequence model named mmT5, with extra language-specific modules for each language. It is expected to address two limitations of current seq-to-seq models: (1) curse of multilinguality; (2) source language hallucination problem. Experiments on both zero-shot cross-lingual transfer and multilingual training scenarios show the effectiveness of the model.

**Questions For The Authors:**

1.	The number of language-specific modules is fixed to 100 in this paper. What if a new language comes? How does the model generalize to the novel language?
2.	It seems that the total parameter of mmT5-base is 580M+14M*100=1980M, which is larger than other baselines (e.g., XLM-R and mT5-base). Do the improvements merely come from the expansion in size?
3.	Compared with encoder-only model XLM-R, mmT5-base is not competitive in both XQuAD and XNLI. Also XLM-R is smaller than mmT5 in size, the authors should explain it more detailedly.
4.	Also, the authors should provide more results on other popular tasks like machine translation.

**Reasons To Accept:**

1.	The paper is well written and the main contributions are clear.
2.	This paper addresses two limitations of current seq-to-seq methods with simple structures.


**Reasons To Reject:**

1.	One of the main contributions of this paper is the extra language-specific modules. However, such method is too straight-forward, by simply expanding the model size. Also, the number of language-specific modules is fixed to 100, it seems that it is difficult to generalize to other novel languages.
2.	In the experiment section, the selected baselines and provided performances are weak: 1) there lack more encoder-decoder models or decoder-only models for comparison, since they are more popular in multilingual tasks. 2) Compared with XLM-R, mmT5 is not competitive (e.g., in XQUAD and XNLI).
3.	Also, the model is not tested on other popular tasks like machine translation, which are also suitable for sequence-to-sequence models.


**Reproducibility:**

4: Could mostly reproduce the results, but there may be some variation because of sample variance or minor variations in their interpretation of the protocol or method.

**Reviewer Confidence:**

3: Pretty sure, but there's a chance I missed something. Although I have a good feel for this area in general, I did not carefully check the paper's details, e.g., the math, experimental design, or novelty.

---

> ### Author Rebuttal · Authors · 2023-08-28
>
> We extend our heartfelt gratitude to the reviewers for their thoughtful engagement with our work and their valuable feedback. We have carefully considered each comment and question.
>
> Before addressing the individual questions, it is essential to emphasize that our primary contribution lies in the freezing strategy applied to modular networks to effectively combat the challenge of generating text in the incorrect language. The pronounced positive impact of this strategy on the modular model, while having a neutral effect on dense models, is a testament to the delicate interplay between these two factors: modularity and training regime of frozen components.
>
> To your questions and concerns:
>
> > ”One of the main contributions of this paper is the extra language-specific modules. However, such method is too straight-forward, by simply expanding the model size”,
>
> The concern regarding the simplicity of our approach and the expansion in model size is noted. While it's true that our method involves increasing the model size by incorporating language-specific modules, we argue that simplicity in design is not inherently a reason for rejection. However, our approach is not solely reliant on model size expansion, instead we allocate model parameters in a "smarter" way by disentangling language and language-agnostic (shared) information in the model.. The incorporation of language-specific modules enables us to tackle the "curse of multilinguality," a challenge prevalent in existing models.
>
>
> > “It seems that the total parameter of mmT5-base is 580M+14M*100=1980M, which is larger than other baselines (e.g., XLM-R and mT5-base). Do the improvements merely come from the expansion in size?”
>
> While it might appear that our model has a larger total number of parameters due to the incorporation of language-specific modules, the crucial aspect lies in the number of per-language trainable parameters, which remains the same between the shared (mT5^s) and modular (mmT5) models. In fact, fine-tuning on a target task results in the same number of trainable parameters as in a dense model, as only the relevant language modules are selected. Moreover, during inference, the modular model maintains efficiency by activating only the necessary language modules, ensuring comparable computational efficiency with a dense model. This architecture choice allows us to strike a balance between language-specific capacity and efficient inference.
>
> Further, our choice of a modular architecture over a simple extension of dense parameters is rooted in several advantages. Firstly, dense models face challenges when extending to new languages due to limitations in the transformer's capacity. In contrast, modular networks inherently support the incorporation of new languages without compromising performance (see Section 5.2 in [1]). Secondly, the absence of language-specific capacity in dense models can lead to an imbalance where high-resource languages dominate the parameter space, disadvantaging low-resource languages. Our modular approach effectively addresses this concern.
>
>
> > “there lack more encoder-decoder models or decoder-only models for comparison, since they are more popular in multilingual tasks.
>
> Our comparison focuses on the encoder-decoder architecture setting. While encoder-only and decoder-only models are indeed popular in multilingual tasks, it's important to note that these models require significant architectural and training adjustments compared to encoder-decoder models. For instance, when transitioning from our encoder-decoder freezing approach to a decoder-only model, it is nontrivial to define what part of the decoder-only model needs to be frozen given that the input and output for encoder-decoder tasks are passed through the same set of parameters, i.e. there is no distinction between the encoder weights and decoder weights.
>
> Given the constraints of compute resources and time, we focused on conducting a comprehensive evaluation within the encoder-decoder model family. This allowed us to deeply explore the potential of our approach while providing meaningful insights into its effectiveness.
> Furthermore, the seq2seq task formulation has become a standard approach to address a wide range of natural language processing tasks, including summarization, and question-answering. The standardized nature of the seq2seq formulation makes it an apt choice for our evaluation, aligning with the established practices in the field.
>
>
> > “ the number of language-specific modules is fixed to 100, it seems that it is difficult to generalize to other novel languages.” “The number of language-specific modules is fixed to 100 in this paper. What if a new language comes? How does the model generalize to the novel language?”
>
> While it's true that the number of language-specific modules in our mmT5 model is fixed to 100, we would like to emphasize that previous state-of-the-art multilingual models are also trained on a limited set of languages, often around 100. This suggests that a fixed set of languages is a common practice in the field, and our choice aligns with established precedents.
>
> Further, we would like to point out that our approach is not limited to the fixed set of 100 languages. Recent works such as [X-Mod; 1],[MAD-X; 3], and [4], propose methods to extend both dense and modular models to more languages. However, [X-Mod; 1] empirically demonstrates that fully dense multilingual models can face catastrophic interference between languages, which our modular approach effectively mitigates. This suggests that the modular architecture can be more robust when extending to more languages, without suffering from degradation in performance. The modularity of our approach enables straightforward integration of new languages and opens up avenues for novel research in continued pretraining for new languages. This is an exciting direction that can lead to advancements in multilingual modeling and further enhance the capabilities of our mmT5 architecture.
>
>
> > “Also, the model is not tested on other popular tasks like machine translation, which are also suitable for sequence-to-sequence models.”, “Also, the authors should provide more results on other popular tasks like machine translation.”
>
> The primary objective of our paper is to address the source language hallucination problem, which is particularly prominent in tasks where the input and output languages are the same. While machine translation is indeed a crucial sequence-to-sequence task, it differs significantly from the tasks we tackle in this paper. Machine translation involves translating text from one language to another, necessitating the handling of different language pairs for input and output. In contrast, our focus is on tasks where the input and output languages are identical, making the hallucination problem more pronounced. While it's plausible that our methods could also be relevant for machine translation tasks, we recognize the need for careful exploration and adaptation of our approach to suit the unique characteristics of machine translation. We acknowledge that machine translation is a valuable research direction and intend to investigate it in future work.
>
>
> > “Compared with encoder-only model XLM-R, mmT5-base is not competitive in both XQuAD and XNLI. Also XLM-R is smaller than mmT5 in size, the authors should explain it more detailedly.”
>
> We want to emphasize that XLM-R and mmT5-base are not directly comparable due to several significant differences in their architecture and training setup (see section 3.2.4 in [7]).
> - Architecture: XLM-R is an encoder-only model, while mmT5-base is a encoder-decoder model. This architectural distinction alone leads to differences in the models' capabilities and performance on different tasks. For instance, for encoder-decoder models classification (e.g. XNLI) and extractive QA (e.g. XQuAD) tasks are framed as generative seq2seq tasks. On the other hand, encoder-only models cannot perform generation at all. Overall, encoder-decoder models are a lot more versatile.
> - Training Regime: XLM-R is trained for a significantly larger number of update steps compared to mmT5-base. This extended training contributes to differences in the models' learned representations and their subsequent performance.
> - Task Setup: The evaluation tasks in the XQuAD and XNLI benchmarks have distinct characteristics. For instance for XQuAD, XLM-R's performance on extractive question-answering and mmT5's performance on generative question-answering tasks may lead to varying performance levels due to the nature of the tasks themselves.
>
> We include the scores of XLM-R and other encoder-only models in our paper merely as a means of completeness.
> Further we want to emphasize that the original (highly impactful) mT5 model does not in fact outperform XLM-R, reasons, among others being the aforementioned points. It's important to note that our proposed approach effectively narrows the performance gap. For instance:
> - In the XQuAD benchmark, XLM-R-large achieves 76.6, mT5-base achieves 67.0, and our mmT5-base achieves 76.3. This indicates that our approach successfully closes the gap by 9.3 points, with only a difference of 0.3 points from XLM-R-Large.
> - Similarly, in the XNLI benchmark, XLM-R-large achieves 79.2, mT5-base achieves 75.4, and our mmT5-base achieves 77.8. This demonstrates a notable narrowing of the performance gap by 2.4 points, with only a difference of 1.4 points from XLM-R.
>
>
> [1] Lifting the Curse of Multilinguality by Pre-training Modular Transformers (Pfeiffer et al., NAACL 2022)
>
> [3] MAD-X: An Adapter-Based Framework for Multi-Task Cross-Lingual Transfer (Pfeiffer et al., EMNLP 2020)
>
> [4] Efficient Test Time Adapter Ensembling for Low-resource Language Varieties (Wang et al., EMNLP Findings 2021)
>
> [7] Exploring the Limits of Transfer Learning with a Unified Text-to-Text Transformer (Raffel et al. Journal of Machine Learning Research 2020)

---

### Meta-Review · Area_Chair_1k5c · 2023-09-19

**Recommendation:** 4

**Metareview:**

This paper explores increasing the language coverage in multi-lingual seq2seq models. The authors propose a modular approach to handle this issue.

Overall, the paper is sound: “The performance of the proposed approaches outperform its most similar precursor mT5” , and “The paper is well written”.  However, the paper could not inspire excitement. All reviewers agree that the paper is not presenting a strong contribution to the research community and mostly incremental. That said, the presented research is worth to be known by the community, as it could be useful for similar experiments.

---

### Decision · Program_Chairs · 2023-10-07

**Decision:**

Accept-Findings

**Comment:**

This paper explores increasing the language coverage in multi-lingual seq2seq models. The authors propose a modular approach to handle this issue.

Overall, the paper is sound: “The performance of the proposed approaches outperform its most similar precursor mT5” , and “The paper is well written”.  However, the paper could not inspire excitement. All reviewers agree that the paper is not presenting a strong contribution to the research community and mostly incremental. That said, the presented research is worth to be known by the community, as it could be useful for similar experiments.